# Unshielded Precipitation Gauge Collection Efficiency with Wind Speed and Hydrometeor Fall Velocity

Jeffery Hoover[1], Michael E. Earle[1], Paul I Joe[1], Pierre E. Sullivan[2]

[1]Environment and Climate Change Canada, Toronto, ON, M3H 5T4, Canada

[2]Department of Mechanical and Industrial Engineering, University of Toronto, Toronto, ON, M5S 3G8, Canada

*Correspondence to*: Jeffery Hoover (jeffery.hoover@canada.ca)

**Abstract.** Collection efficiency transfer functions that compensate for wind-induced collection loss are presented and evaluated for unshielded precipitation gauges. Three novel transfer functions with wind speed and precipitation fall velocity dependence were derived using computational fluid dynamics modelling (CFD function) and measurement data (HE1 function

with fall velocity threshold and HE2 function with linear fall velocity dependence). These functions are evaluated alongside universal ($K_{Universal}$) and climate-specific ($K_{CARE}$) transfer functions with wind speed and temperature dependence. Transfer function performance is assessed using 30-minute precipitation event accumulations reported by unshielded and shielded Geonor T-200B3 precipitation gauges over two winter seasons. The latter gauge was installed in a Double Fence Automated Reference (DFAR) configuration. Estimates of fall velocity were provided by a Precipitation Occurrence Sensor System

(POSS). The CFD function reduced the RMSE (0.08 mm) relative to $K_{Universal}$ (0.20 mm), $K_{CARE}$ (0.13 mm), and the unadjusted measurements (0.24 mm), with a bias error of 0.011 mm. The HE1 function provided a RMSE of 0.09 mm and bias error of 0.006 mm, capturing well the collection efficiency trends for rain and snow. The HE2 function better captured the overall collection efficiency, including mixed precipitation, resulting in a RMSE of 0.07 mm and bias error of 0.006 mm. These functions are assessed across solid and liquid hydrometeor types and for temperatures between -22 °C and 19 °C. The results

demonstrate that transfer functions incorporating hydrometeor fall velocity can dramatically reduce the uncertainty of adjusted precipitation measurements relative to functions based on temperature.

## 1 Introduction

Automated catchment-type precipitation gauge measurements are critical as references for, and input to, weather, climate, hydrology, transportation, and remote sensing applications. The systematic bias and uncertainty of gauge measurements due to wind-induced undercatch is a major challenge, particularly with respect to the measurement of mixed and solid precipitation (Rasmussen et al., 2012;Kochendorfer et al., 2018). For example, an unshielded weighing precipitation gauge can capture less than 50% of the actual amount of solid precipitation falling in air when the wind speed exceeds 5 m s$^{-1}$ (Kochendorfer et al.,

2017b). This measurement challenge has prompted: (1) modelling studies to better understand and visualize the undercatch of hydrometeors by precipitation gauges; and (2) the development of transfer functions to adjust measurements for undercatch effects. Previous work in each of these domains is outlined in Sections 1.1 and 1.2, respectively. The objectives of the present study, which implements numerical modelling and experimental analysis to develop transfer functions with wind speed and hydrometeor fall velocity dependence, are presented in Section 1.3.

## 1.1 Modelling studies

Computational fluid dynamics (CFD) studies have been used to simulate the airflow around precipitation gauges and the associated collection efficiencies for rain and solid precipitation (Nešpor and Sevruk, 1999;Constantinescu et al., 2007;Colli, 2014;Colli et al., 2014;Colli et al., 2015;Colli et al., 2016a;Colli et al., 2016b;Thériault et al., 2012;Thériault et al., 2015;Baghapour and Sullivan, 2017;Baghapour et al., 2017). These studies have demonstrated the influence of wind speed,

turbulence, hydrometeor characteristics (size, density, drag, terminal velocity), and gauge and shield geometry on precipitation gauge undercatch. For rainfall, Nešpor and Sevruk (1999) showed increases in wind-induced error for smaller drop sizes with lower terminal velocities, with errors increasing for higher wind speeds. The conversion factor (inverse of integral collection efficiency) varied with the precipitation intensity and rainfall type, which influenced the distribution of hydrometeor sizes and terminal velocities. Thériault et al. (2012) demonstrated similar trends for snowfall, with collection efficiencies varying

significantly with the type of solid precipitation and size distribution. Simulated collection efficiencies for wet snow and dry snow hydrometeors captured the general upper and lower bounds of experimental observations, respectively, with the lower collection efficiency for dry snow hydrometeors attributed to their lower terminal velocity and interaction with the local airflow around the gauge.

For a Geonor gauge with single-Alter shield, Thériault et al. (2012) used a constant drag coefficient hydrometeor tracking

model to develop a series of transfer functions based on wind speed for different hydrometeor types. Colli et al. (2015) extended this work to show the influence of different hydrometeor drag models on collection efficiency results. Empirical drag model results (Khvorostyanov and Curry, 2005), based on the relative hydrometeor-to-air velocity over the hydrometeor trajectory, were shown to yield higher collection efficiencies compared with constant drag coefficient results that can overestimate drag values. Colli et al. (2015) developed transfer functions based on wind speed for unshielded and single-Alter-shielded gauges

for three specific hydrometeor size distributions. Further studies, using computationally intensive Large Eddy Simulation

models, better resolved the intensity and spatial extent of turbulence around the gauge orifice, which can lead to temporal variations in collection efficiency results (Colli et al., 2016a;Colli et al., 2016b;Baghapour and Sullivan, 2017;Baghapour et al., 2017). The degree of turbulence was found to vary depending on the specific shield configuration and wind speed (Baghapour et al., 2017).

## 1.2 Transfer functions

Intercomparisons of precipitation gauges have served as the primary mechanism for developing transfer functions. In the 1998 World Meteorological Organization (WMO) Solid Precipitation Measurement Intercomparison, transfer functions were determined experimentally by comparing measurements from different gauges (primarily manual) with those from a manual collector with a Tretyakov shield in the WMO Double Fence Intercomparison Reference (DFIR) configuration (Goodison et al., 1998). Precipitation events were monitored by observers, who reported the amount and type of snow, wind speed, and temperature statistics for each event. Events were defined based on the duration of continuous snowfall when the reference DFIR precipitation accumulation was greater than or equal to 3 mm. Adjustment functions for unshielded gauge collection efficiencies were recommended for snow, mixed precipitation, and rain, based on the wind speed at gauge height (Goodison, 1978;Goodison et al., 1998;Yang et al., 1998). While these adjustments could be applied to manual precipitation accumulation measurements, their application to automated measurements at shorter time scales, and where the precipitation type may not be well defined, presents a significant challenge (Colli, 2014;Colli et al., 2014;Colli et al., 2016a;Colli et al., 2016b;Thériault et al., 2015;Thériault et al., 2012)

The WMO commissioned another intercomparison, the Solid Precipitation Intercomparison Experiment (SPICE), to assess various automated technologies for the measurement of precipitation accumulation and snow depth, and to recommend automated field reference systems (Nitu et al., 2018). An automated precipitation gauge configured with a single-Alter shield within a DFIR fence was chosen as the field reference configuration for precipitation accumulation; this was referred to as the Double Fence Automated Reference (DFAR) configuration. Transfer functions for unshielded and shielded gauges were derived as an exponential function of wind speed following the approach of Goodison (1978) and using 30-minute precipitation events from the SPICE data set (Kochendorfer et al., 2017a). Separate functions were developed for solid precipitation and mixed precipitation, as defined by air temperature ranges: less than -2 °C for solid precipitation, and between -2 °C and 2 °C for mixed precipitation.

Using Bayesian analysis of Norwegian measurement data, Wolff et al. (2015) developed a precipitation phase-independent, continuous transfer function with respect to wind speed and air temperature for a single-Alter shielded Geonor precipitation gauge. A similar, but less complex, function was developed by Kochendorfer et al. (2017a;2018) using the SPICE data set, including results from eight measurement sites in Canada, Norway, Finland, Switzerland, and the USA. The application of this "universal" function to precipitation accumulation measurements from unshielded weighing gauges in SPICE was shown to reduce the overall bias relative to the DFAR; however, reductions in the root mean square error (RMSE) were less significant (Kochendorfer et al., 2017a;2017b;2018;Wolff et al., 2015).

When applying universal adjustments with wind speed and air temperature dependence, the errors can vary significantly by
site, presumably driven by differences in climatology (Smith et al., 2020;Kochendorfer et al., 2017a). This has motivated
further work on climate-specific transfer functions (Koltzow et al., 2020;Smith et al., 2020). Other studies have proposed the
use of precipitation intensity for the improved adjustment of solid precipitation (Chubb et al., 2015;Colli et al., 2020). Another
potential avenue for reducing errors in adjusted measurements is by improving the ability of transfer functions to distinguish
among different precipitation types and their aerodynamic properties (Thériault et al., 2012;Wolff et al., 2015;Nešpor and
Sevruk, 1999).

### 1.3 Objectives

In this work, adjustment functions incorporating hydrometeor fall velocity are developed to reduce the uncertainty (RMSE) in
collection efficiency and precipitation accumulation estimates from unshielded Geonor T-200B3 precipitation gauges. The
unshielded gauge configuration allows for the assessment of a broader range of collection efficiencies, as the degree of
undercatch is generally more pronounced for unshielded gauges relative to shielded configurations. Further, by focussing on
the unshielded configuration, no assumptions are required regarding the behaviour of the shield slats and their role in
momentum reduction and turbulence generation around the gauge.

A combined modelling and experimental approach is used in this study. In the modelling component, computational fluid
dynamics and Lagrangian analysis is used to characterize the gauge collection efficiency dependence explicitly in terms of
wind speed and hydrometeor fall velocity, and to derive a corresponding transfer function. In the experimental component, fall
velocity and precipitation type estimates from a Precipitation Occurrence Sensor System (POSS) are used to investigate how
the hydrometeor properties influence the relationships among measured catch efficiency, wind speed, and temperature. Two
additional transfer functions are derived experimentally with wind speed and fall velocity dependence. These new transfer
functions are assessed against transfer functions with dependence on wind speed and air temperature, including one of the
universal functions developed by Kochendorfer et al. (2017a) and a climate-specific function derived herein using a similar
methodology.

## 2 Modelling method

### 2.1 Computational fluid dynamics model

A high-resolution 3-dimensional computer aided design model of the Geonor T-200B3 600 mm capacity gauge (hereafter
Geonor gauge) with 2 m gauge orifice height was developed for the analysis using SolidWorks engineering software (Fig. 1).
The Geonor gauge was modelled with a 200 $cm^2$ orifice, 3.15 mm orifice thickness, and full 360 mm length inlet extending
down into the gauge housing. SolidWorks Flow Simulation software (SolidWorks, 2019) was used to simulate the time-
averaged 3-dimensional flow around the unshielded precipitation gauge. Favre-averaged Navier-Stokes equations were used
to relate the fluid density, velocity components, viscous shear stress, Reynolds (turbulence) stress tensor, and mass distributed

external force per unit mass, with contributions from porous media resistance, gravitational acceleration, and the coordinate systems rotation. The fluid was modelled as isothermal and incompressible. Bulk turbulence through the fluid was captured using the $k$–$\varepsilon$ turbulence model with 5 % turbulence intensity at the inlet (Kato and Launder, 1993). A modified wall functions approach using Van Driest's profile was used to characterize the flow in the near-wall region (SolidWorks, 2013, 2019). The domain width was 7 m and height was 8 m to achieve undisturbed flow at the edges of the domain and ensure uniform

flow near the modeled gauge. The length of the domain was 18 m to allow hydrometeors to be released from a horizontal plane in the free-stream airflow ahead of the gauge (Fig. 1, Table 1). The ground was modelled as an adiabatic frictionless wall, with horizontal wind speeds $U_w$ applied in 1 m s$^{-1}$ increments from 0 to 10 m s$^{-1}$. A finite-volume approach with rectangular parallelepipeds for fluid cells and polyhedrons at fluid solid interfaces was used for mesh generation. A clustered mesh with first- order refinement around the gauge (8 mm cells) and secondary refinement of 2 mm cells around the mounting post,

gauge, and orifice (8.3 million cells in total) was used to resolve the nonlinear updraft velocity profile around the leading edge of the gauge rim and fluid dynamics in the area of the orifice. Simulations for each of the wind speeds were run to convergence for mass, energy, and momentum with model details summarized in Table 1.

**Table 1.** Computational fluid dynamics and Lagrangian hydrometeor tracking model

| Component | Description |
| --- | --- |
| Model | Favre-averaged Navier-Stokes, steady-state, $k$-$\varepsilon$, isothermal, incompressible, gravitational acceleration $a_z$ = -9.81 m s$^{-2}$ |
| Fluid | Air (gas): molecular mass 0.02896 kg mol$^{-1}$ |
| Boundary conditions | Ground: ideal wall (adiabatic, frictionless) |
| Initial and ambient conditions | $P$ = 101325 Pa, $T$ = 293.2 K, $U_w$ = 0,1,2,…,10 m s$^{-1}$ (free-stream wind speed along $x$) |
| Domain | Length x = 18 m, width y = 7 m, height z = 8 m |
| Mesh | 8.3 million cells, $y^+_{mean}$ = 2.5 $y^+_{max}$ =15.1 ($U_w$=1 m s$^{-1}$), $y^+_{mean}$ =17.2 $y^+_{max}$ =90.8 ($U_w$=10 m s$^{-1}$) |
| Hydrometeor injection | Horizontal injection rectangle: length x = 5.5 m, width y = 0.4 m, height z = 2 to 8 m 100000 hydrometeors/rectangle (~4.5 hydrometeors/cm$^2$) |
| Hydrometeor tracking | Lagrangian, uncoupled, spherical hydrometeors, elastic wall reflection inside gauge orifice |

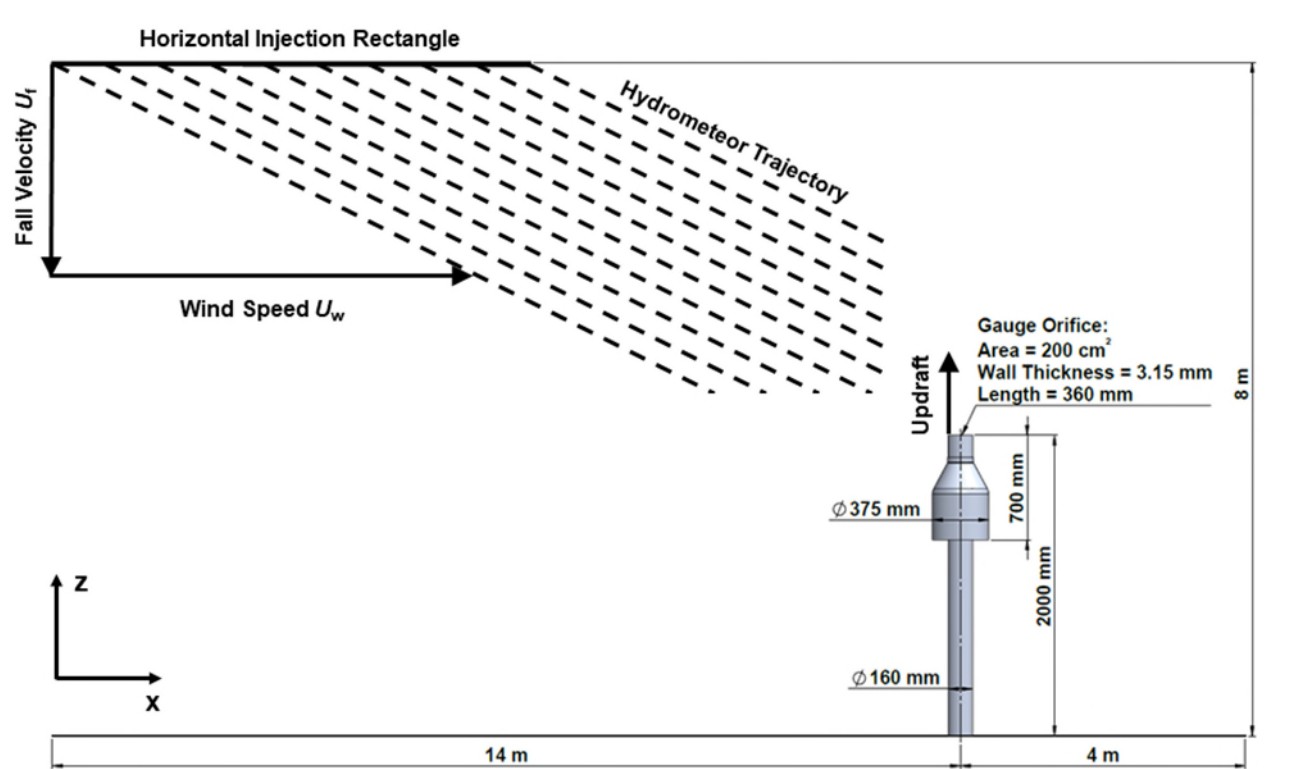

**Figure 1.** Unshielded Geonor T-200B3 600mm capacity gauge and model geometry, including computational domain with ground, horizontal hydrometeor injection rectangle, free-stream hydrometeor trajectories based on wind speed and hydrometeor fall velocity and local updraft around leading edge of gauge orifice.

## 2.2 Collection efficiency based on hydrometeor fall velocity and wind speed

For each wind speed, monodispersed hydrometeors were injected from a horizontal plane upstream and above the gauge orifice (Fig. 1). Hydrometeor types were characterized by their fall velocity, diameter, density, and mass (Table 2). For dry snow and wet snow, the hydrometeor sizes are related to the fall velocity by a general power law (Rasmussen et al., 1999). The fall velocity for dry snow $U_{dry}$ (cm s$^{-1}$) is a function of the size $D_{dry}$ (cm), with a similar relationship for wet snow (Eqs. 1a and b).

$$U_{dry} = 107D_{dry}^{0.2},$$ (1a)

$$U_{wet} = 214D_{wet}^{0.2},$$ (1b)

For dry snow and wet snow, the hydrometeor density was chosen such that the size and fall velocity followed the power law relationship of Rasmussen et al. (1999), with the drag coefficient for spherical hydrometeors given by Henderson (1976) based

on the relative hydrometeor to air velocity. This drag formulation closely matches that of Haider and Levenspiel (1989), and has been used in previous studies (Baghapour and Sullivan, 2017).

**Table 2.** Characteristics for rain, dry snowflake, wet snowflake, and ice pellet hydrometeors.

| Description | Diameter (m) | Density (kg m⁻³) | Mass (kg) | Fall Velocity (m s⁻¹) |
|---|---|---|---|---|
| Rain | 9.989E-05 | 9.982E+02 | 5.209E-10 | 0.25 |
| Rain | 1.571E-04 | 9.982E+02 | 2.028E-09 | 0.50 |
| Rain | 2.121E-04 | 9.982E+02 | 4.986E-09 | 0.75 |
| Rain | 2.671E-04 | 9.982E+02 | 9.956E-09 | 1.00 |
| Rain | 3.227E-04 | 9.982E+02 | 1.756E-08 | 1.25 |
| Rain | 3.793E-04 | 9.982E+02 | 2.851E-08 | 1.50 |
| Rain | 4.370E-04 | 9.982E+02 | 4.363E-08 | 1.75 |
| Rain | 4.962E-04 | 9.982E+02 | 6.385E-08 | 2.00 |
| Rain | 5.569E-04 | 9.982E+02 | 9.029E-08 | 2.25 |
| Rain | 6.195E-04 | 9.982E+02 | 1.242E-07 | 2.50 |
| Rain | 1.378E-03 | 9.982E+02 | 1.369E-06 | 5.00 |
| Rain | 3.956E-03 | 9.982E+02 | 3.236E-05 | 10.00 |
| Dry snow | 2.228E-04 | 5.439E+02 | 3.150E-09 | 0.50 |
| Dry snow | 1.692E-03 | 3.745E+01 | 9.498E-08 | 0.75 |
| Dry snow | 7.130E-03 | 8.837E+00 | 1.677E-06 | 1.00 |
| Wet snow | 2.228E-04 | 1.345E+03 | 7.790E-09 | 1.00 |
| Wet snow | 6.800E-04 | 3.099E+02 | 5.101E-08 | 1.25 |
| Wet snow | 1.692E-03 | 1.062E+02 | 2.693E-07 | 1.50 |
| Wet snow | 3.657E-03 | 4.728E+01 | 1.211E-06 | 1.75 |
| Wet snow | 7.130E-03 | 2.695E+01 | 5.114E-06 | 2.00 |
| Wet snow | 1.285E-02 | 1.731E+01 | 1.922E-05 | 2.25 |
| Wet snow | 2.176E-02 | 1.221E+01 | 6.586E-05 | 2.50 |
| Ice pellet | 1.472E-03 | 9.167E+02 | 1.532E-06 | 5.00 |
| Ice pellet | 4.276E-03 | 9.167E+02 | 3.752E-05 | 10.00 |

For dry snow, hydrometeor fall velocities between 0.5 m s⁻¹ to 1.0 m s⁻¹ were included, representing sizes up to 7 mm and densities below that of ice (Table 2). Fall velocities for wet snow were selected between 1.0 m s⁻¹ to 2.5 m s⁻¹ for sizes between 0.2 to 21.8 mm. Spherical ice pellets with a density of 916.7 kg/m³ were also included in the analysis for fall velocities of 5.0 m s⁻¹ and 10.0 m s⁻¹. Spherical hydrometeors with a density of 998.2 kg/m³, representing rain, were included for fall velocities from 0.25 m s⁻¹ to 10 m s⁻¹ for comparison.

The hydrometeor trajectory was derived from the drag force, gravitational force, and buoyancy forces acting on the hydrometeor as it moves through the flow using Lagrangian analysis. A dilute two-phase flow was assumed, where the influence of the hydrometeors on the fluid flow was negligible and hydrometeor interactions including potential hydrometeor coalescence were ignored. Combining these forces gave the net hydrometeor acceleration $\boldsymbol{a}_p$ as a function of the drag coefficient $C_D$, hydrometeor cross sectional area $A_p$, density of air $\rho_a$, relative hydrometeor to air velocity $\boldsymbol{u}_p - \boldsymbol{u}_a$,

hydrometeor density $\rho_p$, hydrometeor volume $V_p$, and acceleration due to gravity $g$ acting in the negative $\hat{\mathbf{z}}$ direction as shown in Eq. (2).

$$a_p = \frac{d\boldsymbol{u}_p}{dt} = -\frac{C_D \rho_a A_p \left| \boldsymbol{u}_p - \boldsymbol{u}_a \right| \left( \boldsymbol{u}_p - \boldsymbol{u}_a \right)}{2 \rho_p V_p} + \frac{\left( \rho_a - \rho_p \right) g}{\rho_p} \hat{\mathbf{z}} ,$$ (2)

The hydrometeors were injected into the flow uniformly at equilibrium with an initial velocity $\boldsymbol{u}_{p1}$ equal to the free-stream wind speed $U_w$ along the $\hat{\mathbf{x}}$ direction and hydrometeor fall velocity $U_f$ in the negative $\hat{\mathbf{z}}$ direction (down).

$$\boldsymbol{u}_{p1} = U_w \hat{\mathbf{x}} - U_f \hat{\mathbf{z}} ,$$ (3)

In the free-stream region under steady-state conditions, the hydrometeor fall velocity and terminal velocity will be equivalent. Hydrometeor interactions with the inside of the gauge orifice were assumed to be ideal reflections.

The collection efficiency $CE\left(U_w, U_f\right)$ for a given free-stream wind speed $U_w$ and hydrometeor fall velocity $U_f$ corresponds to the ratio of the number of hydrometeors collected $N_C$ over the horizontal gauge orifice area $A_C$ to the number of hydrometeors injected from the horizontal injection plane above $N_I$ over the horizontal injection plane area $A_I$, as shown in Eq. (4).

$$CE\left(U_w, U_f\right) = \frac{\dfrac{N_C}{A_C}}{\dfrac{N_I}{A_I}} = \frac{N_C A_I}{N_I A_C} ,$$ (4)

## 2.3 Integral collection efficiency with wind speed

The collection efficiency presented in Sect. 2.2 was for monodispersed hydrometeors with identical size, mass, density and fall velocity. In this section, we define the integral collection efficiency as that derived over the entire hydrometeor size distribution and associated characteristics (e.g. fall velocity, density and volume).

### 2.3.1 Rainfall

The total precipitation intensity $P_{Total}$ is a function of the hydrometeor size distribution $N_R(D)$, density $\rho_p(D)$, volume $V_p(D)$, fall velocity $U_f(D)$, and density of water $\rho_w$.

$$P_{Total} = \frac{1}{\rho_w} \int_0^\infty N_R(D) \rho_p(D) V_p(D) U_f(D) dD ,$$ (5)

The hydrometeor size distribution (number of hydrometeors per unit size per unit volume) for raindrops $N_R(D)$ can be expressed as a gamma distribution defined by the parameter $N_{0R}$ (m$^{-3}$ cm$^{-1-\mu}$), exponential factor $\Lambda_R$ (cm$^{-1}$), exponent $\mu$ (unitless), and hydrometeor diameter $D$ (cm) as given by Ulbrich (1983).

$$N_R(D) = N_{0R} D^{\mu} e^{-\Lambda_R D}, \tag{6}$$

Assuming the product of the exponential factor and the maximum hydrometeor diameter is large, the exponential factor can be expressed in terms of the exponent and median volume diameter $D_{0R}$ (cm).

$$\Lambda_R = \frac{3.67 + \mu}{D_{0R}}, \tag{7}$$

The median volume diameter is determined based on the rainfall intensity $R$ (mm/hr) and the empirical constants $\varepsilon$ and $\delta$ for the specific rain type.

$$D_{0R} = \varepsilon R^{\delta}, \tag{8}$$

The value of the exponent $\mu$ will be positive or negative depending on the rain type (orographic, thunderstorm, stratiform, or showers), corresponding to a concave up or down distribution when plotted on a $\log(N_R(D))$ versus $D$ plot. Orographic rain with an exponent less than zero corresponds to a concave up distribution with small droplets and low fall speed (Ramana et al., 1959) (Table 3). Thunderstorm rain with a concave down distribution corresponds to large drops and high fall speed (Blanchard, 1953).

The hydrometeor fall velocity for rainfall is given by Beard (1976). At standard air temperature and pressure, the rainfall hydrometeor fall velocity $U_f$ (m s$^{-1}$) is a function of the equivalent hydrometeor diameter, acceleration due to gravity $g$, raindrop hydrometeor density (density of water) $\rho_w = 998.2 kg/m^3$, density of air $\rho_a = 1.23 kg/m^3$, dynamic viscosity of air $\eta = 1.79E-5 N \cdot s/m^2$, and surface tension of water $\sigma_w = 0.07199 N/m$.

**Table 3.** Rainfall parameters for gamma drop size distribution summarized by Ulbrich (1983).

| Description | $\mu$ | $N_0$ | $\varepsilon$ | $\delta$ | Source |
|---|---|---|---|---|---|
| Orographic rain | -1.03 | 9.82 x 10$^3$ | 0.055 | 0.28 | Ramana et al. (1959) |
| Thunderstorm rain | 1.01 | 1.24 x 10$^6$ | 0.101 | 0.18 | Blanchard (1953) |

The integral collection efficiency is the ratio of the precipitation intensity that is captured by the gauge to that which is falling in the free-stream airflow for a given hydrometeor size distribution, following the approach of Nešpor and Sevruk (1999). The collection efficiency is implicitly dependent on the equivalent hydrometeor diameter through the hydrometeor fall velocity.

$$CE_{\text{R,Overall}} = \frac{\int_0^\infty CE(U_{\text{w}}, U_{\text{f}}) D^3 N_R(D) U_{\text{f}}(D) \, \mathrm{d}D}{\int_0^\infty D^3 N_R(D) U_{\text{f}}(D) \, \mathrm{d}D} , \tag{9}$$

### 2.3.2 Snowfall

The total precipitation intensity $P_{\text{S,Total}}$ for snowfall is a function of the hydrometeor size distribution for snowfall $N_{\text{S}}(D)$,

density $\rho_{\text{p}}(D)$, volume $V_{\text{p}}(D)$, fall velocity $U_{\text{f}}(D)$ and density of water $\rho_{\text{w}}$, integrated over the range of equivalent diameters $D$.

$$P_{\text{S,Total}} = \frac{1}{\rho_{\text{w}}} \int_0^\infty N_{\text{S}}(D) \rho_{\text{p}}(D) V_{\text{p}}(D) U_{\text{f}}(D) \, \mathrm{d}D , \tag{10}$$

Taking the equivalent snowfall diameter as that for a spherical water droplet with the density of water gives the total precipitation intensity integral as a function of the snowfall size distribution. Using this approach, both the size distribution

and fall velocities of snowflakes are defined as a function of the equivalent spherical diameter of water droplets.

$$P_{\text{S,Total}} = \frac{\pi}{6} \int_0^\infty D^3 N_{\text{S}}(D) U_{\text{f}}(D) \, \mathrm{d}D , \tag{11}$$

The size distribution for snowflakes $N_{\text{S}}(D)$ can be expressed by the Gunn and Marshall size distribution (Gunn and Marshall, 1957) as a function of the size distribution parameter $N_{0S}$ (m$^{-1}$ mm$^{-1}$), exponential factor $\Lambda_S$ (cm$^{-1}$), and equivalent spherical water drop diameters $D$ (cm) above 0.1 cm.

$$N_{\text{S}}(D) = N_{0S} e^{-\Lambda_S D} , \tag{12}$$

The size distribution parameter $N_{0S}$ (m$^{-1}$ mm$^{-1}$) and exponential factor $\Lambda_S$ (cm$^{-1}$) vary with the precipitation intensity $R$ (mm h$^{-1}$).

$$N_{0S} = 3.8 \times 10^3 R^{-0.87} , \tag{13}$$

$$\Lambda_S = 25.5 R^{-0.48} , \tag{14}$$

The median volume diameter $D_{0S}$ (cm) varies with the size distribution slope parameter or precipitation intensity as shown by Atlas (1953).

$$D_{0S} = \frac{3.67}{\Lambda_S} , \tag{15}$$

The fall velocity $U_{\text{f}}(D)$ (cm s$^{-1}$) for various snowfall types, based on the equivalent spherical diameter of water droplets $D$ (cm), is given by a general power law.

$$U_f(D) = aD^b ,\tag{16}$$

The fall velocity power law coefficients for dendrites, rimed dendrites, and a mixture of dendrites and aggregates of plates is summarized in Table 4 based on the work of Langleben (1954).

**Table 4.** Snowfall fall velocity parameters with power law formulation for equivalent water droplet diameter.

| Description | $a$ | $b$ | Source |
|---|---|---|---|
| Dendrites | 178 | 0.372 | Langleben (1954) |
| Rimed dendrites | 210 | 0.283 | Langleben (1954) |
| Mixture of dendrites and aggregate of plates | 366 | 0.611 | Langleben (1954) |


The integral collection efficiency at a given wind speed is the ratio of the precipitation intensity that is captured by the gauge to that which is falling in the free-stream airflow for a given crystal habit and size distribution. As with that for rainfall, the collection efficiency is implicitly dependent on the equivalent hydrometeor diameter through the hydrometeor fall velocity.

$$CE_{S,Overall} = \frac{\int_0^\infty CE(U_w, U_f) D^3 N_S(D) U_f(D) \, dD}{\int_0^\infty D^3 N_S(D) U_f(D) \, dD} ,\tag{17}$$

**3 Modelling results**

**3.1 Numerical modelling**

Computational fluid dynamics simulations that included the time-averaged effects of flow turbulence were run for free-stream wind speeds between 0 and 10 m s$^{-1}$ for the unshielded Geonor precipitation gauge. Results for the 1 m s$^{-1}$ case are shown in Fig. 2. The flow is diverted upward as it passes over the leading edge of the gauge, with recirculation of the airflow within the

gauge orifice (Fig. 2). This general velocity profile has been well established in previous studies (Thériault et al., 2012;Colli et al., 2016a;Baghapour et al., 2017).

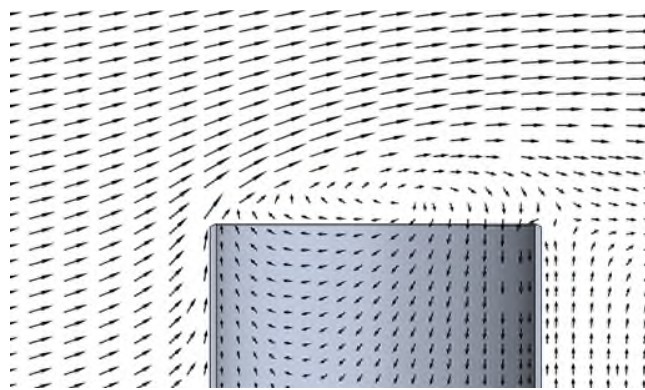

**Figure 2.** Air velocity around Geonor gauge for 1 m s$^{-1}$ free-stream horizontal wind speed represented by velocity magnitude and direction.


To validate the CFD model used in the present study, normalized velocities along the gauge centerline $u*$ (normalized by the free-stream wind speed) as a function of the normalized height above the gauge orifice $z*$ (normalized by the orifice diameter) are compared with results from previous studies for wind speeds of 1 m s$^{-1}$ and 10 m s$^{-1}$ in Fig. 3. The peak normalized velocities are 2 % lower than the Baghapour et al. (2017) $k$-$\omega$ SST model at 1 m s$^{-1}$ wind speed (Fig. 3a) and within 1 % at 10

m s$^{-1}$ wind speed (Fig. 3b). Larger differences in peak normalized velocity, within 9 %, are observed relative to the Colli (2016a) $k$-$\omega$ SST model results at 1 m s$^{-1}$ wind speed (Fig. 3a).

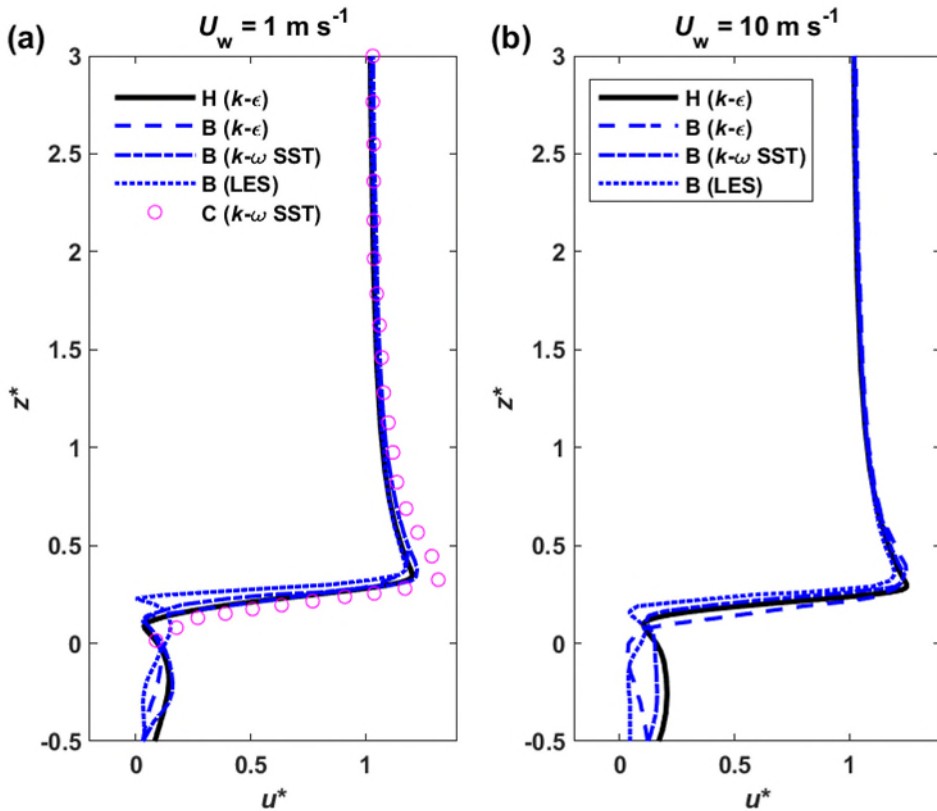

**Figure 3.** Gauge centerline results comparison above gauge orifice for present model (H $k$-$\varepsilon$), Baghapour et. al. (2017) $k$-$\varepsilon$ model (B $k$-$\varepsilon$), Baghapour et. al. (2017) $k$-$\omega$ shear stress tensor model (B $k$-$\omega$ SST), Baghapour et. al. (2017) LES model (B LES), and Colli et. al. (2016a) $k$-$\omega$ shear stress tensor model (C $k$-$\omega$ SST) for normalized free-stream velocity with (a) 1 m s$^{-1}$ wind speed $U_w$ and (b) 10 m s$^{-1}$ wind speed $U_w$.

For visualization purposes, hydrometeor trajectories are illustrated for the 3 m s$^{-1}$ wind speed case with fall velocities of 0.5 m s$^{-1}$, 1.0 m s$^{-1}$, 1.5 m s$^{-1}$ and 2.0 m s$^{-1}$ in Fig. 4. As the fall velocity increases, the hydrometeor approach angle from the horizontal increases, based on the relative magnitudes of the wind speed and fall velocity. For the 2.0 m s$^{-1}$ hydrometeor fall velocity, it is apparent that the hydrometeor trajectories experience little change due to the local airflow around the gauge (Fig. 4d). For lower fall velocities, the deflection due to the updraft around the leading edge of the gauge is more apparent, with the 0.5 m s$^{-1}$ fall velocity hydrometeors closely coupled to the flow upward and over the gauge, with a few hydrometeors drawn in at the back side of the gauge orifice (Fig. 4a). For each of the hydrometeor injections, the same horizontal spacing of hydrometeors is present prior to encountering the local airflow around the gauge. The number of hydrometeors captured is reduced for lower hydrometeor fall velocities at the same wind speed. At 3 m s$^{-1}$ wind speed, rainfall hydrometeors with 0.25 m s$^{-1}$ hydrometeor fall velocity are all carried over the gauge, corresponding to a collection efficiency of zero (not shown).

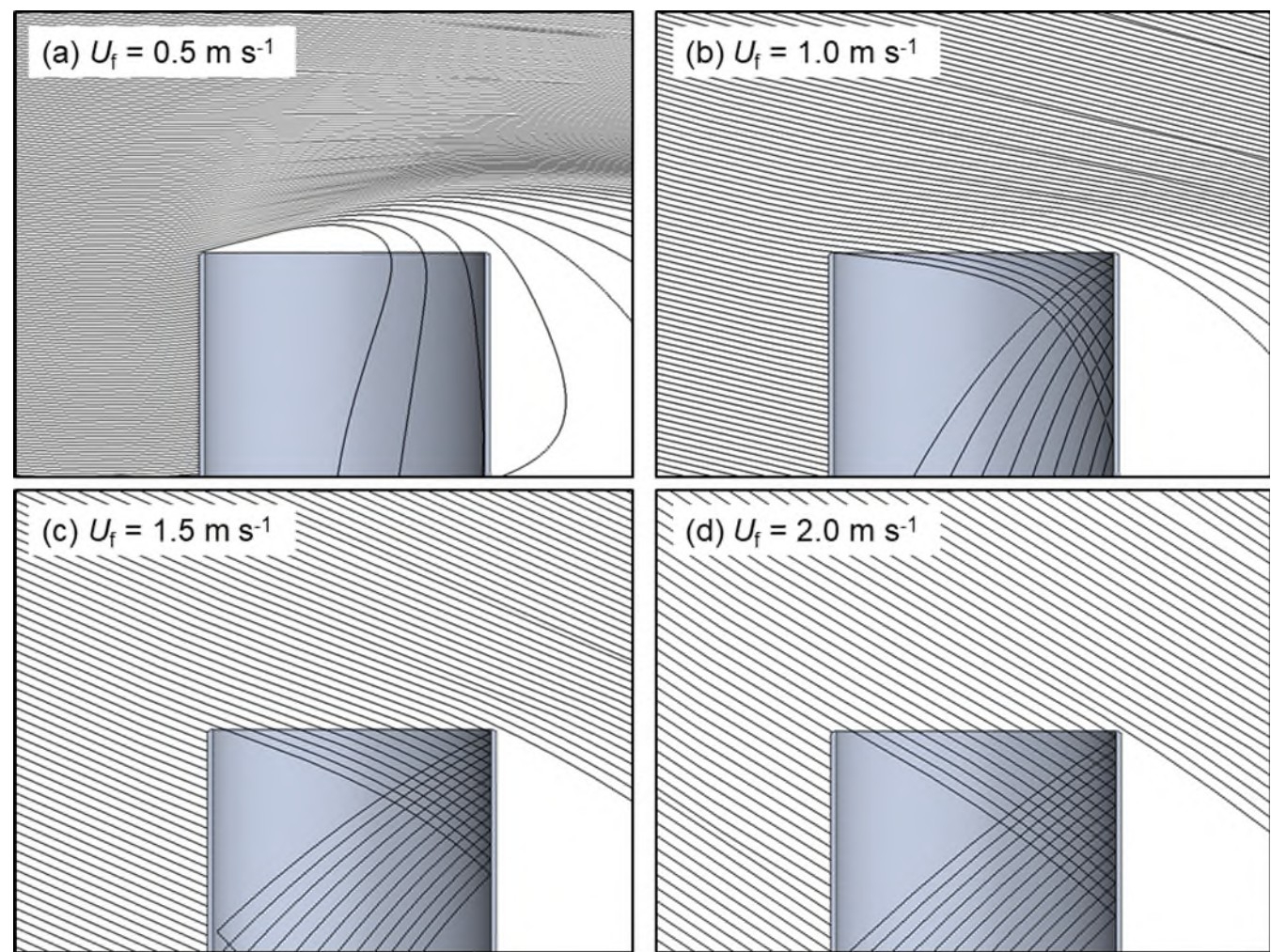

**Figure 4.** Flow simulation results showing hydrometeor trajectories for 3 m s$^{-1}$ free-stream wind velocity with rain hydrometeor fall velocity of (a) 0.5 m s$^{-1}$, (b) 1.0 m s$^{-1}$, (c) 1.5 m s$^{-1}$, and (d) 2.0 m s$^{-1}$.

### 3.2 Collection efficiency dependency on wind speed and hydrometeor fall velocity

The numerical results demonstrate a clear dependence on the hydrometeor fall velocity (Fig. 5). Hydrometeors with higher fall velocities exhibit increased collection efficiency, and the collection efficiency tends to decrease with increasing wind speed. Rain, dry snow, and wet snow hydrometeors with 1.0 m s$^{-1}$ fall velocity exhibit a similar collection efficiency decrease with increasing wind speed, despite differences in diameter, density, and mass. For rain and ice pellet hydrometeors with 5.0 m s$^{-1}$ fall velocities, the results are close to 1 and nearly identical at all wind speeds, irrespective of differences in density. Here, the circles for rain overlap the squares for ice pellets in Fig. 5. Rain and wet snow with identical fall velocities between 1.0 m s$^{-1}$

and 2.5 m s$^{-1}$ also exhibit similar results for wind speeds under 5 m s$^{-1}$. Above 5 m s$^{-1}$ wind speed, the collection efficiency for rain is slightly elevated above that for wet snow. For dry snow hydrometeors with fall velocities between 0.5 m s$^{-1}$ and 1.0 m s$^{-1}$, there is good agreement with the corresponding rain hydrometeors for horizontal wind speeds up to about 3 m s$^{-1}$. Above this wind speed, the 0.5 m s$^{-1}$ dry snow hydrometeors exhibit good agreement with rain hydrometeors, while the collection efficiency for 1.0 m s$^{-1}$ dry snow hydrometeors decreases more rapidly with wind speed relative to rain hydrometeors with the same fall velocity. Collection efficiency differences across all hydrometeor types with identical fall velocities are within 0.18, with root mean square differences of 0.05, over all wind speeds and hydrometeor fall velocities studied.

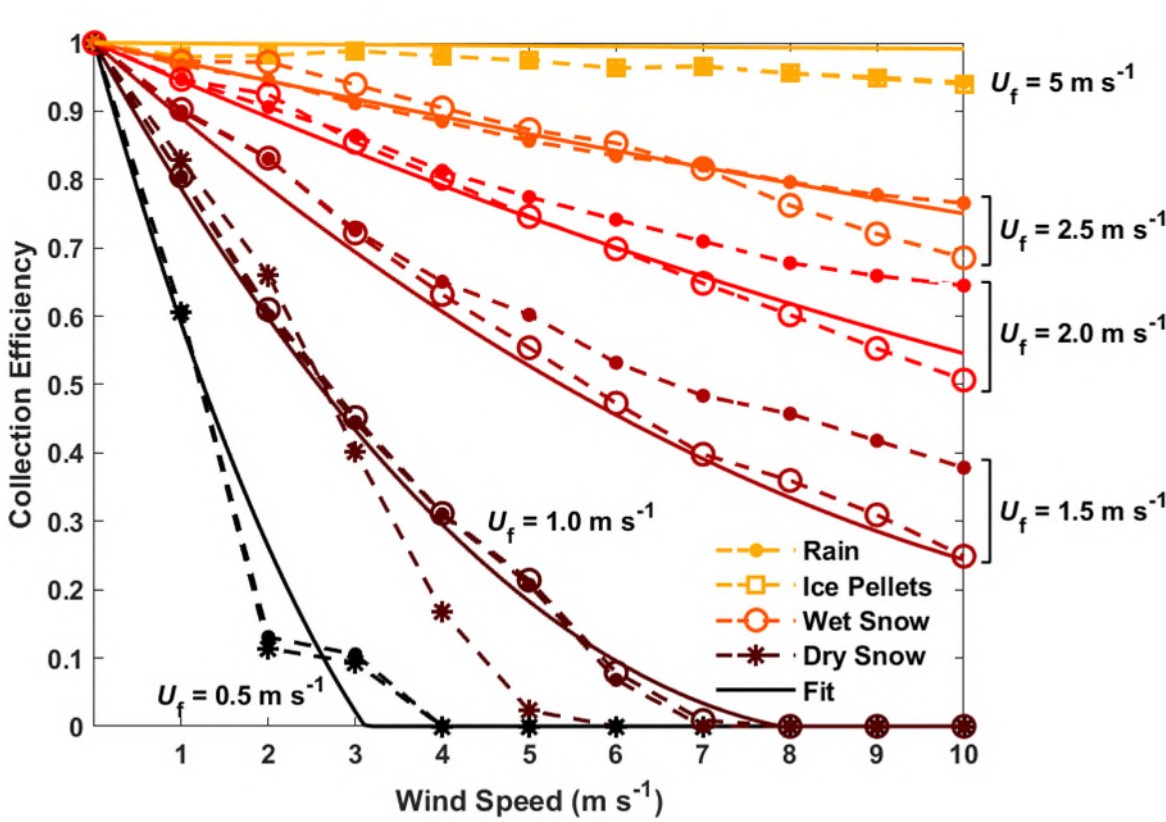

**Figure 5.** Flow simulation results for Geonor unshielded gauge collection efficiency based on wind speed and hydrometeor fall velocity for rain, ice pellets, wet snow, dry snow, and CFD transfer function.

### 3.3 CFD transfer function

The simulation results demonstrate that the collection efficiency is dependent on the free-stream wind speed $U_w$ and hydrometeor fall velocity $U_f$. The CFD transfer function, $CE_{CFD}(U_w, U_f)$, is presented based on a polynomial fit to wind speed and an exponential hydrometeor fall velocity dependence, with both velocities having units of m s$^{-1}$.

$$CE_{CFD}(U_w, U_f) = 1 - b_1 U_w e^{-b_2 U_f} + b_3 U_w^2 e^{-b_4 U_f},$$ (18)

This expression was selected due to its ability to capture the nonlinearity in the collection efficiency up to 10 m s$^{-1}$ wind speed,

as well as the nonlinear fall velocity dependence with collection efficiencies approaching 1 for higher fall velocities. Table 5 shows the best-fit coefficients (RMSE of 0.03) from a combined nonlinear regression for dry snow (0.5 m s$^{-1}$ and 0.75 m s$^{-1}$ fall velocities), wet snow (1.0 m s$^{-1}$, 1.25 m s$^{-1}$, ... , 2.5 m s$^{-1}$ fall velocities), and rain (5 and 10 m s$^{-1}$ fall velocities). A single CFD curve was used for each fall velocity in the fit to ensure that the transfer function was unbiased over the entire range of fall velocities studied.


**Table 5.** Non-linear regression fit parameters, standard errors (SE), and units for the Geonor unshielded gauge collection efficiency as a function of wind speed and hydrometeor fall velocity with RMSE = 0.0302 and $R^2$ = 0.989.

| Coefficient | Value | SE | Units |
|---|---|---|---|
| $b_1$ | 0.908 | 0.048 | s m$^{-1}$ |
| $b_2$ | 1.387 | 0.037 | s m$^{-1}$ |
| $b_3$ | 0.143 | 0.031 | s$^2$ m$^{-2}$ |
| $b_4$ | 2.422 | 0.167 | s m$^{-1}$ |

A wind speed threshold $U_{wc}$ is defined above which the collection efficiency is zero for a given maximum cutoff hydrometeor

fall velocity $U_{fc}$. This was derived from Eq. (18) by solving for the roots where the collection efficiency is zero using the quadratic formula. Conversely, this expression gives the hydrometeor fall velocity below which the collection efficiency will be zero for a given wind speed.

$$U_{wc} = \frac{b_1}{2b_3} \exp\left[(b_4 - b_2)U_{fc}\right] - \frac{\sqrt{b_1^2 \exp(-2b_2 U_{fc}) - 4b_3 \exp(-b_4 U_{fc})}}{2b_3 \exp(-b_4 U_{fc})},$$ (19)

Fig. 5 shows the comparison of the CFD transfer function with the CFD results. For hydrometeor fall velocities above 5.0 m

s$^{-1}$, the collection efficiency expression is within -0.13 and 0.10 of CFD results over all hydrometeor types. For fall velocities between 1.25 to 2.5 m s$^{-1}$, the fit is within ±0.06 over all wind speeds. For fall velocities of 0.25 m s$^{-1}$ to 1.0 m s$^{-1}$, the fit captures the rapid decrease in collection efficiency with wind speed well overall, with a maximum difference of 0.16 for dry snow at 5 m s$^{-1}$ wind speed. The CFD transfer function captures well the collection efficiency trends for the different hydrometeor types, with RMSE values of 0.04 for rain, 0.02 for ice pellets, 0.02 for wet snow, and 0.05 for dry snow.


Fig. 6 shows the CFD transfer function dependence with fall velocity. For a given wind speed, the collection efficiency increases nonlinearly with hydrometeor fall velocity. For fall velocities above 3 m s$^{-1}$ the collection efficiency is close to 1. The collection efficiency rapidly decreases as the fall velocity is reduced, particularly below 2.5 m s$^{-1}$ fall velocity. Increasing the wind speed decreases the collection efficiency. The fall velocity where the collection efficiency is zero illustrates the fall

velocity cutoff given by Eq. (19). Hydrometeors at or below this fall velocity will not be captured by the gauge. As the wind speed increases, the fall velocity cutoff increases, reaching 1.1 m s$^{-1}$ at 9 m s$^{-1}$ wind speed.

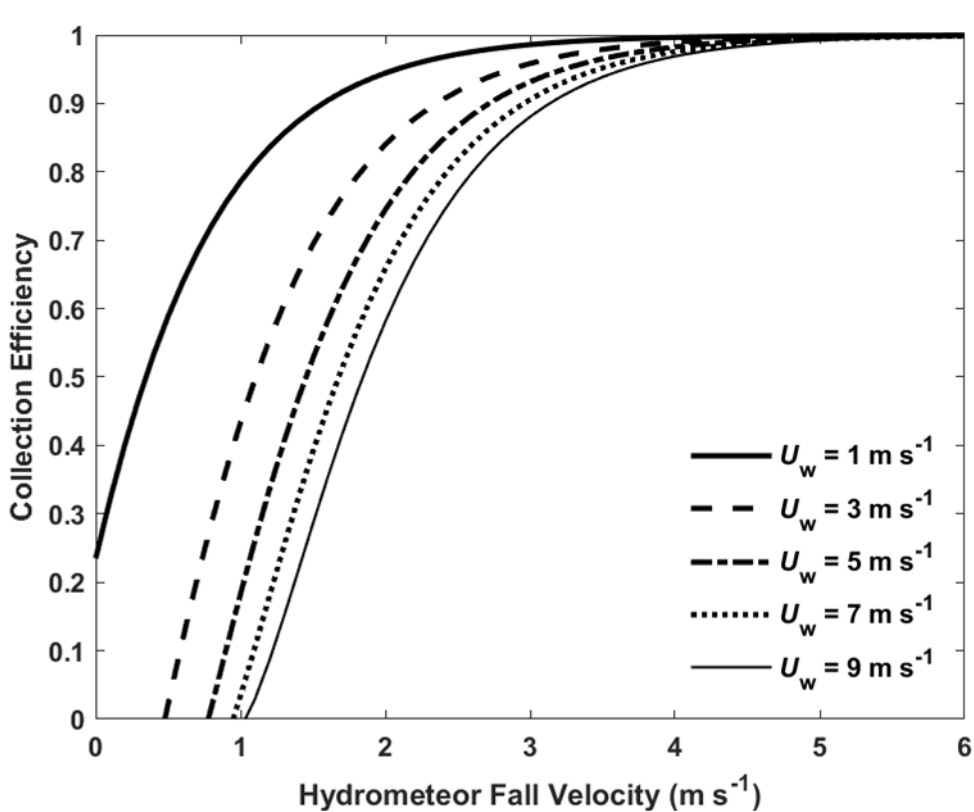

**Figure 6.** Geonor unshielded gauge collection efficiency for exponential fit model with hydrometeor fall velocity and wind speed.


### 3.4 Integral collection efficiency

### 3.4.1 Wind speed dependence

For each hydrometeor type and precipitation intensity, the integral collection efficiency (Eqs. 9 and 17) was derived for wind

speeds from 0 to 10 m s$^{-1}$ using the CFD transfer function (Eq. 18) based solely on wind speed and hydrometeor fall velocity. The collection efficiency, derived using the CFD transfer function, decreases nonlinearly with wind speed, decreasing more rapidly at lower wind speeds and more gradually at higher wind speeds above approximately 5 m s$^{-1}$ (Fig. 7). A wide range of integral collection efficiency results with wind speed is apparent, depending on the hydrometeor type and precipitation intensity.

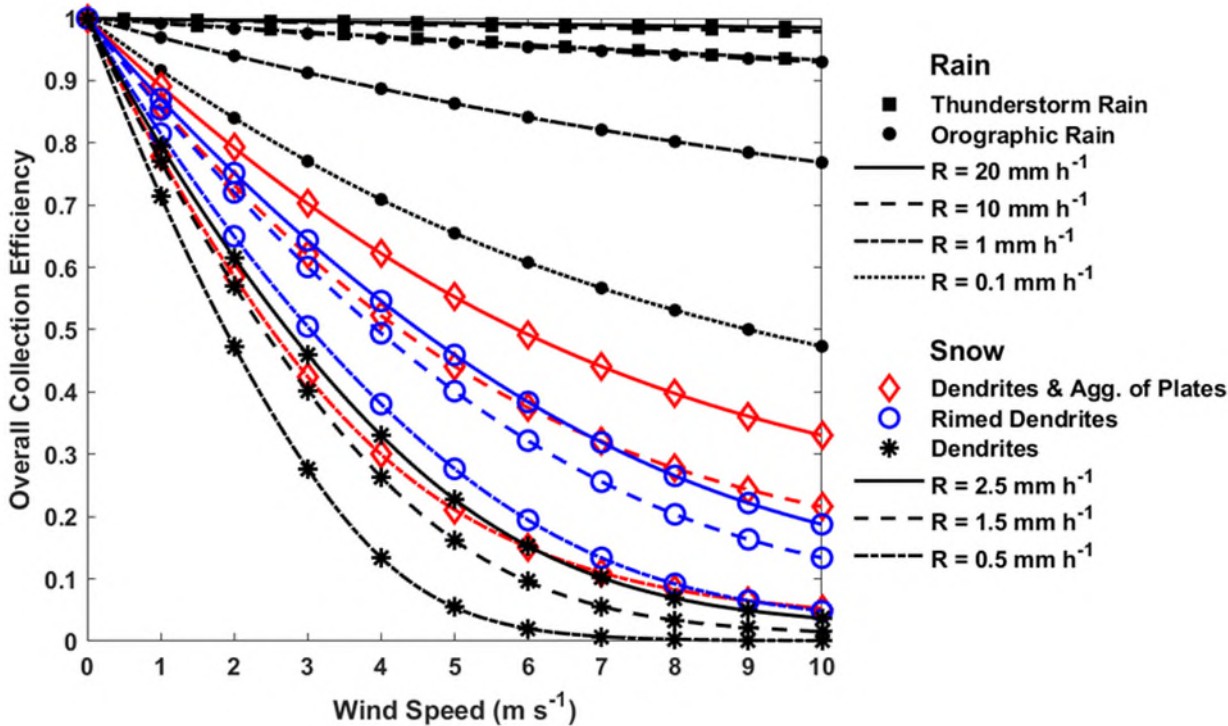


**Figure 7.** Integral Geonor unshielded gauge collection efficiency with wind speed for thunderstorm rain at 20 mm h$^{-1}$, 10 mm h$^{-1}$, and 1 mm h$^{-1}$ precipitation intensities (R); orographic rain at 10 mm h$^{-1}$, 1 mm h$^{-1}$, and 0.1 mm h$^{-1}$ precipitation intensities (R); and aggregates of plates, rimed dendrites, and dendrites at 2.5 mm h$^{-1}$, 1.5 mm h$^{-1}$, and 0.5 mm h$^{-1}$ precipitation intensities (R).

Lower integral collection efficiencies are observed for snowfall relative to orographic and thunderstorm rain. Across snowfall types, a wide range of integral collection efficiency values are apparent. Dendrites show the greatest nonlinearity with wind speed and the lowest integral collection efficiency, with the latter decreasing to 0.01 at 6.7 m s$^{-1}$ wind speed and 0.5 mm h$^{-1}$ precipitation intensity. The integral collection efficiency continues to decrease with increasing wind speed, remaining small, but non-zero, up to 10 m s$^{-1}$ wind speed. For 1.5 mm h$^{-1}$ and 2.5 mm h$^{-1}$ intensities, integral collection efficiencies are higher,

with the magnitude of the increase varying with the wind speed. It is important to note the more gradual decrease in integral collection efficiency for wind speeds above ~6 m s⁻¹ compared with the results in Fig. 5 for a given hydrometeor fall velocity. The integral collection efficiency includes the contribution from a range of hydrometeor sizes, including larger hydrometeors with higher fall velocities that are still able to be captured by the gauge, providing small but non-zero integral collection efficiencies.

For comparison, integral collection efficiency results were computed for dry snow and wet snow using the same hydrometeor size distribution parameters as Colli et al. (2016b) for sizes between 0.25 mm to 20 mm. The results from the different models show good agreement (Fig. 8). For dry snow, the integral collection efficiency decreases more gradually with wind speed in the present study than the Colli et al. (2016b) results, with collection efficiency values up to 0.18 higher at 3 m s⁻¹ wind speed using the present model. For wet snow, the results of Colli et al. (2016b) show a nonlinear decrease in collection efficiency

above 5 m s⁻¹ wind speed that is not apparent in the results from the present study. As a result, the present study predicts up to 0.16 higher collection efficiency at 8 m s⁻¹ wind speed.

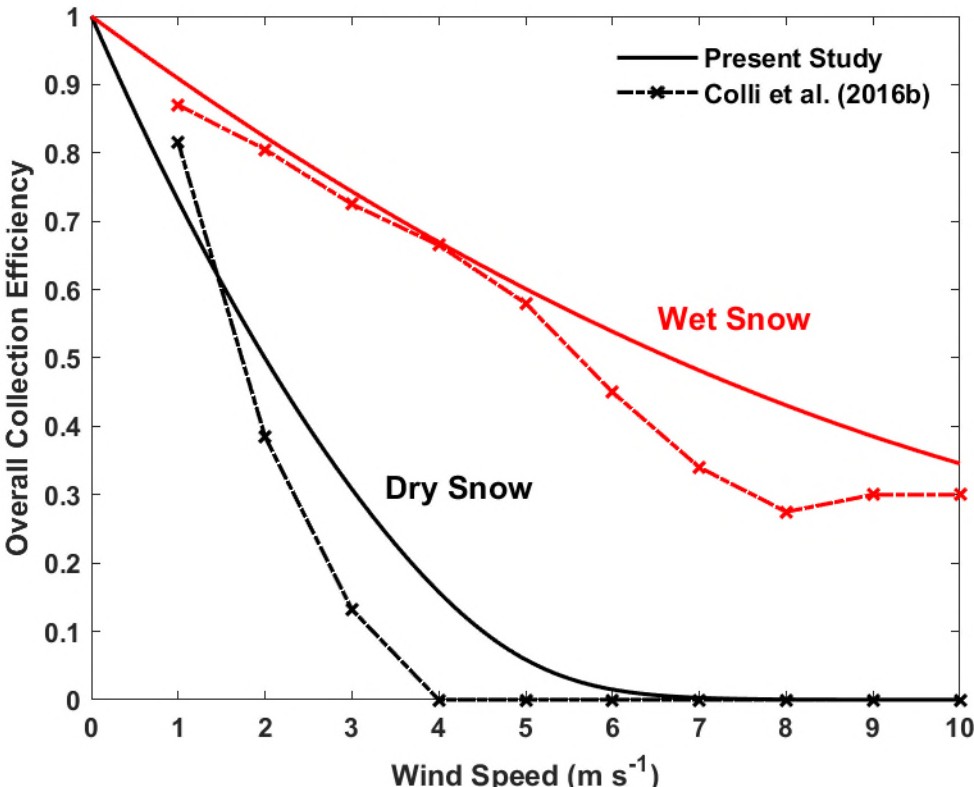

**Figure 8.** Integral Geonor unshielded gauge collection efficiency with wind speed for present study and Colli et al. (2016b) *k-ω* SST model
for dry snow and wet snow with Colli et al. (2016b) size distribution.

### 3.4.2 Precipitation intensity dependence

Using the CFD transfer function, integral collection efficiencies were derived for 0.5 mm h$^{-1}$, 1.5 mm h$^{-1}$, and 2.5 mm h$^{-1}$ precipitation intensities for three different snowfall types, between 0.1 mm h$^{-1}$ and 10 mm h$^{-1}$ for orographic rain, and between 380 1 mm h$^{-1}$ and 10 mm h$^{-1}$ for thunderstorm rain. The results are shown in Fig. 9 for selected wind speeds (1 m s$^{-1}$, 3 m s$^{-1}$, and 6 m s$^{-1}$).

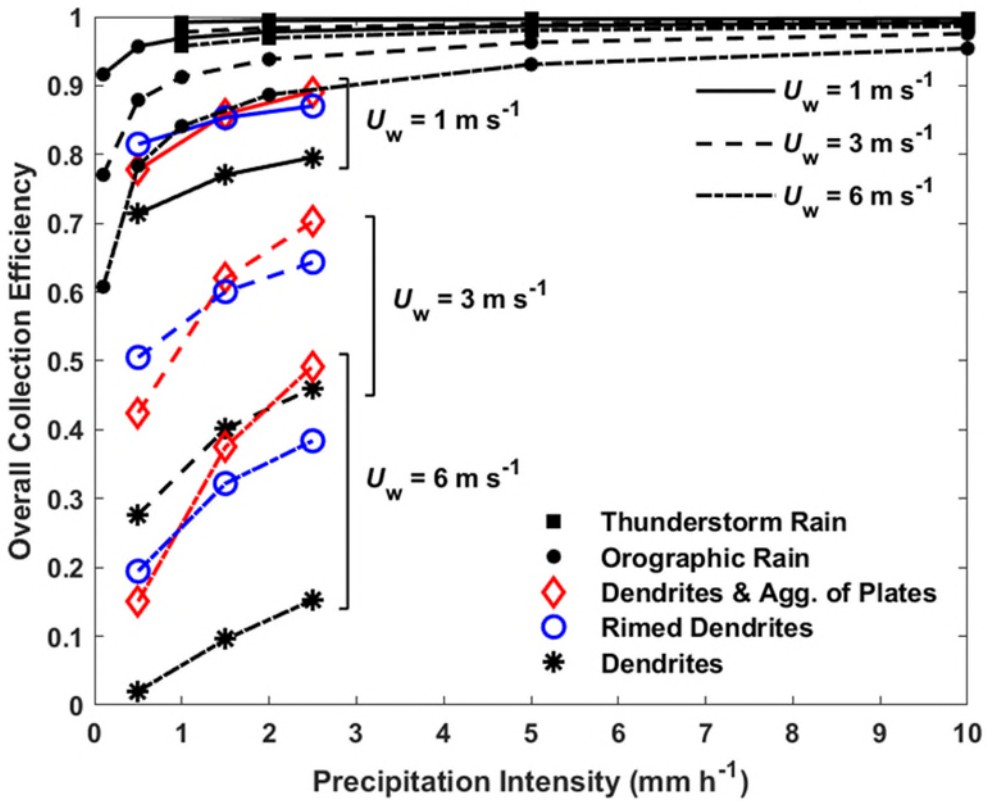

**Figure 9.** Integral Geonor unshielded gauge collection efficiency with precipitation intensity for rainfall and snowfall types at 1 m s$^{-1}$, 3 m s$^{-1}$, and 6 m s$^{-1}$ wind speeds.


Integral collection efficiencies increase with precipitation intensity and decrease with wind speed. For thunderstorm rain at 3 m s$^{-1}$ wind speed, the integral collection efficiency increases from 0.97 to 0.99 when the precipitation intensity increases from 1 mm h$^{-1}$ to 10 mm h$^{-1}$. For orographic rain, a sharp decrease in the integral collection efficiency is apparent with decreasing precipitation intensity below 1 mm h$^{-1}$.

For all snowfall types, the integral collection efficiency is shifted to lower values relative to rain. The integral collection efficiency increases with precipitation intensity from 0.42 at 0.5 mm h$^{-1}$ to 0.70 at 2.5 mm h$^{-1}$ for dendrites and aggregates of

plates at 3 m s$^{-1}$ wind speed. Increasing the wind speed to 6 m s$^{-1}$ further decreases the integral collection efficiency from 0.15 at 0.5 mm h$^{-1}$ to 0.49 at 2.5 mm h$^{-1}$.

For dendrites and aggregates of plates, rimed dendrites, and dendrites, integral collection efficiencies are within 0.09 to 0.10 of one another for 0.5 mm h$^{-1}$ precipitation intensities at 1 m s$^{-1}$ wind speed. This range increases to 0.17 for 0.5 mm h$^{-1}$ precipitation intensity and 0.34 for 2.5 mm h$^{-1}$ precipitation intensity at 6 m s$^{-1}$ wind speed. This provides an estimate of the overall variability in integral collection efficiency due to crystal habit if the precipitation intensity, wind speed, and the occurrence of snowfall are all known, but not the specific snowfall type. In cases where only the precipitation intensity and wind speed is known, and the hydrometeor phase (rainfall or snowfall) and type is uncertain, the range of possible integral collection efficiencies grows dramatically. For example, at 6 m s$^{-1}$ wind speed, the integral collection efficiencies for rain exceed 0.75, while that for dendrites is below 0.16; hence, the catch efficiency in this case can vary by ~ 0.74 depending on the hydrometeor phase and type.

### 3.4.3 Hydrometeor fall velocity dependence

Fig. 10 shows the integral collection efficiency as a function of hydrometeor fall velocity for each precipitation type (thunderstorm rain, orographic rain, dendrites and aggregates of plates, rimed dendrites, and dendrites), precipitation intensity (0.1 to 20 mm h$^{-1}$ for rainfall and 0.5 to 2.5 mm h$^{-1}$ for snowfall), and wind speed (1 m s$^{-1}$, 3 m s$^{-1}$, and 6 m s$^{-1}$) shown previously in Fig. 9. Here, the fall velocity at the median volume diameter is used as an estimate for the fall velocity distribution. The results take a similar form to that of the CFD transfer function shown in Fig. 6, with collection efficiencies increasing nonlinearly with hydrometeor fall velocity for a given wind speed. Dendrites, with the lowest fall velocity, exhibit the lowest integral collection efficiency. Rimed dendrites and dendrites and aggregates of plates with higher fall velocity exhibit higher collection efficiency. In this fall velocity range below 1.5 m s$^{-1}$, the collection efficiency rapidly increases approximately linearly with fall velocity. For orographic rain and thunderstorm rain, with even higher fall velocity, the integral collection efficiency nonlinearly approaches 1. As wind speeds increase from 1 m s$^{-1}$ to 6 m s$^{-1}$, collection efficiencies for all precipitation types are shifted down at the lower end of the fall velocity spectrum below 2 m s$^{-1}$ and still converge to 1 at higher fall velocities, close to 5 m s$^{-1}$.

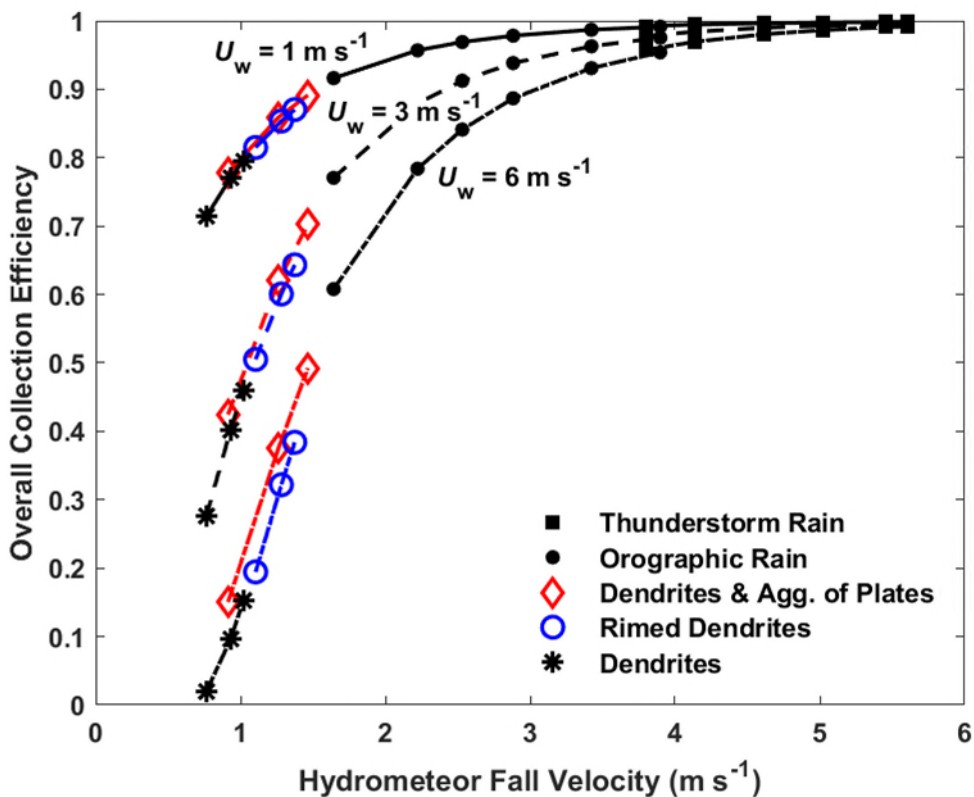

**Figure 10.** Integral Geonor unshielded gauge collection efficiency with hydrometeor fall velocity at median volume diameter for rainfall and snowfall types at 1 m s$^{-1}$, 3 m s$^{-1}$, and 6 m s$^{-1}$ wind speeds.

For snowfall, the integral collection efficiency difference across dendrites, rimed dendrites, and dendrites and aggregates of plates is less than 0.06 for 0.5 mm h$^{-1}$, 1.5 mm h$^{-1}$, and 2.5 mm h$^{-1}$ precipitation intensities at 6 m s$^{-1}$ wind speed, and within 0.03 for the same precipitation intensities at 3 m s$^{-1}$ wind speed. For rainfall, the integral collection efficiency difference is less than 0.01 at 3.8 m s$^{-1}$ fall velocity, where orographic rain and thunderstorm rain overlap. Orographic rain exhibits median volume diameter fall velocities between 1.6 m s$^{-1}$ to 3.9 m s$^{-1}$ for precipitation intensities from 0.1 mm h$^{-1}$ to 10 mm h$^{-1}$.

Thunderstorm rain exhibits median volume diameter fall velocities between 3.8 m s$^{-1}$ to 5.6 m s$^{-1}$ for precipitation intensities from 1 mm h$^{-1}$ to 20 mm h$^{-1}$.

## 4 Experimental method

### 4.1 Instrumentation

Experimental measurements were performed in conjunction with SPICE over the 2013/14 and 2014/15 winter periods (November 1 to April 30) at the Centre for Atmospheric Research Experiments (CARE) site in Egbert, Ontario, Canada. Measurements of precipitation accumulation were performed using 600 mm capacity Geonor T-200B3 gauges in unshielded and reference DFAR configurations. Both gauges were securely mounted on concrete foundations to limit wind-induced vibrations. The performance of these gauges was confirmed by full-scale field verifications at the start and end of testing, with

annual maintenance to inspect, clean, level, and recharge each gauge. The gauges were charged with a mixture of antifreeze (60% methanol and 40% propylene glycol) and oil (Esso Bayol 35 in 2013/14, discontinued; Exxon Mobil Isopar M in 2014/15).

Measurements of precipitation occurrence were obtained using a Thies Laser Precipitation Monitor (LPM) installed inside the inner fence of the DFAR. Wind speed and direction measurements at 2 m gauge height were performed with a Vaisala WS425

ultrasonic wind sensor adjacent to the unshielded gauge. Temperature was measured with a Yellow Springs International model 44212 thermistor in an aspirated Stevenson screen. Further details are available in the SPICE final report (Nitu et al., 2018).

### 4.2 Sampling, quality control, and precipitation event selection

The instruments were sampled using a Campbell Scientific CR3000 data logger. For each Geonor T-200B3 precipitation gauge, the frequency and precipitation accumulation for each of the three transducers was reported at 6-second intervals, the latter

computed from the former using manufacturer-provided calibration coefficients. Minutely measurements of precipitation occurrence from the Thies LPM were recorded. The scalar average wind speed and vector average wind direction were recorded over 1-minute intervals. Based on SPICE procedures, these data were processed using a format check to replace missing data with null values, a range check to identify and remove outliers outside the manufacturer-specified output thresholds, a jump filter to remove spikes exceeding maximum point-to-point variation thresholds, and a Gaussian filter to

smooth out high frequency noise in Geonor precipitation accumulation measurements (Nitu et al., 2018). Periods of instrument maintenance and power outages were removed from the analysis. The Geonor accumulation data were aggregated to 1-minute intervals for subsequent analysis.

Precipitation events were identified during both measurement periods using the SPICE event selection procedure (Nitu et al. 2018). These events were defined as 30-minute periods with at least 0.25 mm of precipitation recorded by the reference DFAR

precipitation gauge and at least 60% precipitation occurrence reported by the Thies LPM. The use of the LPM as a secondary confirmation of precipitation occurrence minimizes the likelihood of events with false precipitation due to dumps of snow or ice into the gauge, wind induced vibrations, or other factors. Following the approach of Kochendorfer (2018), a minimum 0.075 mm accumulation threshold was applied for the unshielded gauge to ensure that measurements exceeded the gauge uncertainty and that derived collection efficiency values were reliable. The 30-minute event duration was chosen to be

sufficiently long to reduce noise and ensure high confidence in measured parameters and sufficiently short to avoid the influence of diurnal temperature variations, while also providing a larger number of events for analysis relative to longer durations. Note that unless otherwise stated, all precipitation events referred to hereafter are 30-minute events derived using the above approach.

## 4.3 POSS fall velocity and precipitation type

The POSS is a small upward-facing bistatic X band radar capable of measuring the precipitation fall velocity based on the Doppler frequency shift of the received signal (Canada, 1995;Sheppard, 1990, 2007;Sheppard et al., 1995;Sheppard and Joe, 1994, 2000, 2008). During periods of precipitation, the POSS outputs both the mean and mode received signal frequency derived from the Doppler frequency spectrum over the previous minute. The mean precipitation fall velocity $U_{f\_mean}$ is estimated from the transmitted wavelength $\lambda$ and the mean frequency $f_{mean}$ of the measured Doppler power density spectrum

for falling precipitation hydrometeors.

$$U_{f\_mean} = \frac{f_{mean}\lambda}{2},$$
(20a)

The mode precipitation fall velocity $U_{f\_mode}$ is described by a similar function, based on the mode frequency $f_{mode}$ of the measured Doppler power density spectrum.

$$U_{f\_mode} = \frac{f_{mode}\lambda}{2},$$
(20b)

For each 30-minute event, the mean and mode event fall velocity correspond to the average of all minutely mean and mode values, respectively. The transfer functions presented in this work were derived using both forms of event fall velocity and assessed in terms of the RMSE and bias error (BE) of adjusted measurements relative to the DFAR. The specific fall velocity indicated for each transfer function corresponds to that which produced the lowest RMSE and BE. The POSS also provides a minutely precipitation type output corresponding to very light, light, moderate, and heavy precipitation for rain, snow, hail,

and undefined precipitation. Each event is classified as 'rain' or 'snow', corresponding to a minimum 70 % occurrence of that precipitation type over the event period (i.e. at least 21 minutes of precipitation occurrence). 'Mixed' precipitation events correspond to the presence of both 'rain' and 'snow' for the remaining events not classified as rain or snow. 'Undefined' precipitation corresponds to events where the precipitation is not captured by the three other classifications.

## 4.4 Transfer functions with wind speed and temperature

Due to the systematic error associated with gauge undercatch, the unshielded gauge can capture less precipitation than the true amount falling in the air. The measured collection efficiency $CE_m$ is defined as the ratio of the precipitation accumulation reported by the unshielded gauge $P_{un}$ relative to that reported by the DFAR $P_{DFAR}$ for each event, and is given by:

$$CE_m = \frac{P_{un}}{P_{DFAR}}, \tag{21}$$

Assuming that the gauge measurement uncertainties are independent and random with equivalent accumulations (corresponding to a collection efficiency equal to 1) and uncertainties, the uncertainty in the collection efficiency $\sigma_{CE}$ scales with the relative magnitude of the gauge uncertainty $\sigma_P$ and the event accumulation value $P$ by error propagation.

$$\sigma_{CE} = \frac{\sqrt{2}\sigma_P}{P}, \tag{22}$$

Collection efficiency transfer functions $CE$ attempt to capture the performance of the unshielded gauge relative to the reference configuration based on wind speed, temperature, or other meteorological parameters. They can then be applied to adjust precipitation accumulations from an unshielded gauge in operational settings where reference measurements are not available.

$$P_{adj} = \frac{P_{un}}{CE}, \tag{23}$$

Kochendorfer et al. (2017a;2018) used SPICE measurement data from eight test sites to develop an exponential and trigonometric transfer function based on wind speed $U_w$ and air temperature $T$. This is referred to as K$_{Universal}$ in this work (Eq. 24a). For wind speeds above a threshold value $U_{wt}$ of 7.2 m s$^{-1}$, the wind speed is fixed at the threshold value (Eq. 24b) to avoid the potential for erroneous catch efficiency values at higher wind speeds that were not well represented in the SPICE measurement dataset. Based on a similar rationale, no adjustment is applied for temperatures above 5 °C. Note that while Kochendorfer et al. (2017b) considered wind speeds at both gauge height and at 10 m, $U_w$ will denote the gauge height wind speed in this work.

$$CE_K\left(U_w \leq U_{wt}, T\right) = \exp\left[-b_1 U_w \left(1 - \tan^{-1}\left(b_2 T\right) + b_3\right)\right], \tag{24a}$$

$$CE_K\left(U_w > U_{wt}, T\right) = \exp\left[-b_1 U_{wt} \left(1 - \tan^{-1}\left(b_2 T\right) + b_3\right)\right], \tag{24b}$$

The coefficients for K$_{Universal}$ are provided in Table 6.

**Table 6.** Unshielded Geonor T-200B3 precipitation gauge collection efficiency transfer function coefficients for solid and mixed precipitation with 30-minute scalar mean wind speed $U_w$ at gauge height for: $K_{Universal}$ function with wind speed and air temperature $T$ dependence, with constant value above wind speed threshold with Kochendorfer et al. (2017a) coefficients; $K_{CARE}$ function with wind speed and air temperature dependence, with constant value above wind speed threshold; HE1 model with dependence on wind speed and mean hydrometeor fall velocity $U_{f\_mean}$ threshold; and HE2 model with wind speed and mode hydrometeor fall velocity dependence and mode hydrometeor fall velocity threshold.

| Description | Eq. | Function | Coefficients | | | | Threshold |
|---|---|---|---|---|---|---|---|
| | | | $b_1$ | $b_2$ | $b_3$ | $b_4$ | |
| $K_{Universal}$ | 5 | $f(U_w,T)$ | 0.0785 | 0.729 | 0.407 | - | $U_{wt} = 7.2$ m s$^{-1}$, $T \leq 5$ °C |
| $K_{CARE}$ | 5 | $f(U_w,T)$ | 0.1651 | 0.186 | -0.757 | - | $U_{wt} = 7.2$ m s$^{-1}$, $T \leq 1.33$ °C |
| HE1 | 7 | $f(U_w,U_{f\_mean})$ | 0.139 | - | - | - | $U_{f\_mean} \leq 1.93$ m s$^{-1}$, $U_w \leq 5.75$ m s$^{-1}$ |
| HE2 | 8 | $f(U_w,U_{f\_mode})$ | 0.244 | 0.0869 | - | - | $U_{f\_mode} \leq 2.81$ m s$^{-1}$, $U_w \leq 0.8/(b_1\text{-}b_2U_f)$ |

Using the same formulation, a site-specific transfer function based on wind speed and temperature was derived using the CARE dataset, for comparison with $K_{Universal}$. Best-fit regression coefficients were determined by varying the temperature threshold below 5 °C with the collection efficiency constrained to 1 above the threshold value. Solving Eq. 24a for the temperature when the collection efficiency equals 1 provides additional constraint on the $b_3$ coefficient as a function of the $b_2$ coefficient and temperature threshold $T_t$.

$$b_3 = \tan^{-1}(b_2 T_t) - 1, \tag{24c}$$

The coefficients for the CARE site-specific transfer function, referred to as $K_{CARE}$ in this work, are provided in Table 6. The temperature threshold was varied over the measurement range in 0.01 °C increments to provide the lowest overall RMSE.

# 5 Experimental results

## 5.1 Precipitation type

Using the minutely POSS precipitation type output, events were classified as 'rain', 'snow', 'mixed', or 'undefined' following the methodology in Sect. 4.3. The relative occurrence of different precipitation types as reported by the POSS for the event dataset is summarized in Table 7. The fall velocities in Table 7 were estimated by the POSS following the methodology in Sect. 4.3; the temperatures were estimated from a YSI44212 thermistor in an aspirated Stevenson screen as described in Sect. 4.1.

**Table 7.** Mean fall velocities and temperatures of precipitation events by type classification.

| Precipitation phase | Fall velocities (m s$^{-1}$) | Temperatures (°C) | Events (#) |
|---|---|---|---|
| Snow | 0.93 to 2.32 | < 0.5 | 233 |
| Mixed | 1.2 to 4.6 | -7.0 to 2.1 | 45 |
| Undefined | 1.0 to 4.3 | -5.4 to 6.6 | 40 |
| Rain | 1.4 to 6.4 | -4.8 to 18.9 | 196 |

Based on the mean fall velocities and temperatures for each precipitation event (Fig. 11, Table 7), snow events occurred at temperatures below 0.5 °C and with fall velocities of 0.93 m s$^{-1}$ to 2.32 m s$^{-1}$. Mixed events were characterized by mean temperatures between -7.0 °C and 2.1 °C and mean fall velocities between 1.2 m s$^{-1}$ and 4.6 m s$^{-1}$, while undefined precipitation events occurred at mean temperatures between -5.4 °C and 6.6 °C and fall velocities between 1.0 m s$^{-1}$ and 4.3 m s$^{-1}$. Rain events were characterized by mean temperatures between -4.8 °C and 18.9 °C and mean fall velocities between 1.4 m s$^{-1}$ and 6.4 m s$^{-1}$. Over the temperature range between -5 °C and 2 °C, rain, snow, mixed, and undefined precipitation types were all present, demonstrating the challenge of estimating precipitation type using temperature alone (e.g. as done for the $K_{Universal}$ and $K_{CARE}$ transfer functions). Within this temperature range, a wide variety of mean fall velocities, between 1 and 6 m s$^{-1}$, is also apparent.

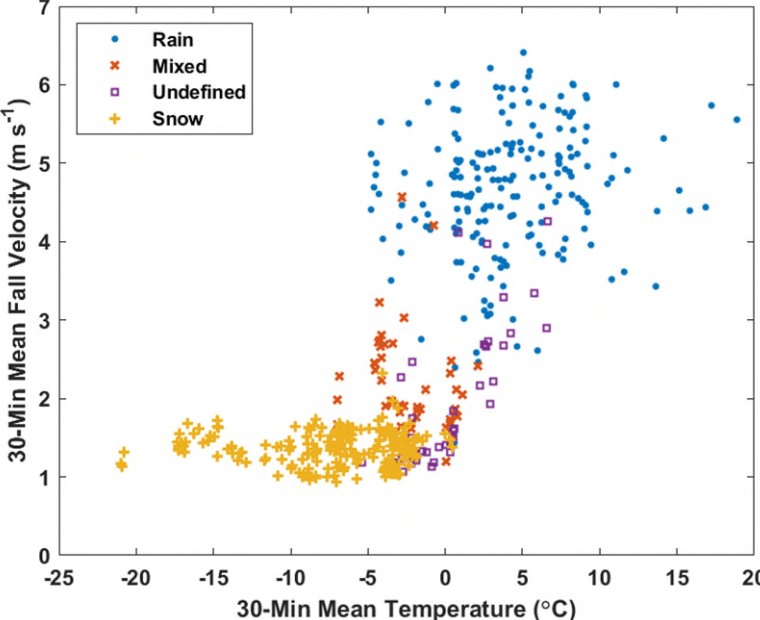

**Figure 11.** Mean air temperature and fall velocity for 30-minute events with rain, snow, mixed, and undefined precipitation (see Table 7 for summary).

## 5.2 Collection efficiency

The unshielded gauge collection efficiency results are shown as a function of the 30-minute DFAR event accumulations in Fig. 12a and stratified by precipitation type classification. The collection efficiency for rain shows less scatter and less uncertainty for higher reference precipitation accumulations. The dashed lines in Fig. 12a show the decrease in the collection efficiency uncertainty with increasing precipitation accumulation for a collection efficiency equal to 1 and a precipitation accumulation uncertainty of 0.1 mm ($k = 2$) given by Eq. 22. These lines appear to capture the overall trend observed for rain events. The snowfall events show a markedly different trend, however, with collection efficiencies as low as 0.3.

The collection efficiency for all events as a function of mean wind speed and precipitation type classification is shown in Fig. 12b. For rain events, the collection efficiencies are close to 1. For snow, an approximately linear decrease in the collection efficiency with mean wind speed is apparent, with the collection efficiency decreasing to 0.3 at a wind speed of 5 m s$^{-1}$. Mixed precipitation collection efficiencies span a range of values between those of rain and snow. For undefined precipitation, some events have collection efficiencies close to 1 at high wind speeds, similar to rain events, while others appear to decrease with increasing wind speed in a similar fashion to that observed for snow events.

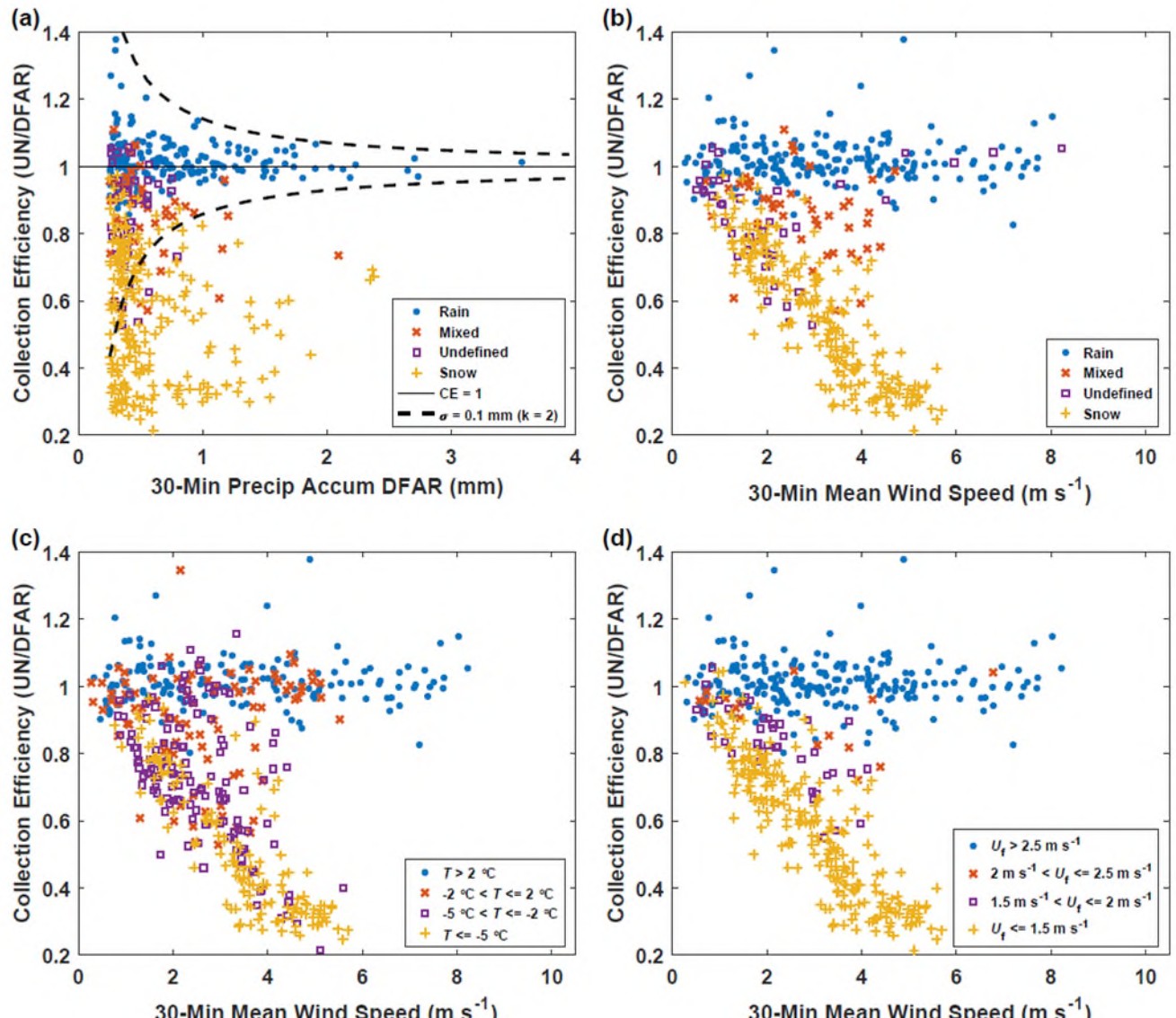

**Figure 12.** Collection efficiency of the unshielded gauge as a function of: (a) precipitation accumulation and event precipitation type (dashed lines illustrate accumulation uncertainty threshold); (b) wind speed and event precipitation type; (c) wind speed and mean air temperature $T$ categories; and (d) wind speed and mode fall velocity $U_{f\_mode}$ categories.


The dependence of collection efficiencies on the mean wind speed over four separate mean temperature ranges is shown in Fig. 12c. For mean event temperatures above 2 °C, the collection efficiencies are generally close to 1, typical of rain. For temperatures between -5 °C and -2 °C and between -2 °C and 2 °C, a range of collection efficiency values are observed, from those typical of snow to those typical of rain. This variation is attributed to the wide range of fall velocities within this

temperature range, which includes snow, rain, and mixed precipitation events (Fig. 12b). At colder temperatures, below -5 °C, collection efficiencies appear to decrease approximately linearly with wind speed, consistent with the trend observed for snow events in Fig. 12b.

Stratifying the collection efficiency results as a function of mean event wind speed by the mode fall velocity shows more distinct trends (Fig. 12d) relative to those observed when stratifying by temperature (Fig. 12c). Collection efficiencies are close

to 1 for fall velocities greater than 2.5 m s$^{-1}$, generally corresponding to rain. Conversely, fall velocities below 1.5 m s$^{-1}$ show an approximately linear decrease in collection efficiency with increasing wind speed up to about 6 m s$^{-1}$. A number of the values with higher collection efficiencies in this low fall velocity range correspond to mixed precipitation, where both snow and rain may be present. Between 1.5 m s$^{-1}$ to 2.5 m s$^{-1}$ fall velocity, intermediate collection efficiency values are evident, with collection efficiencies transitioning from lower to higher values, despite a fewer number of observations in this range.


## 5.3 Derivation of fall velocity transfer functions from CE results

Two additional transfer functions were formulated based on the apparent linear dependence of CE on wind speed for different hydrometeor fall velocity regimes observed in experimental results (Fig. 12d). These functions are applicable to all hydrometeor types, and have different fall velocity thresholds to describe the transition of precipitation phase from the lower

fall velocities characteristic of snow to the higher fall velocities characteristic of rain and mixed precipitation.

The first transfer function, referred to as HE1, is based on the assumption of a linear decrease in collection efficiency $CE_{HE1}$ with wind speed $U_w$ for hydrometeors with mean fall velocity $U_f$ below 1.93 m s$^{-1}$, generally corresponding to snowfall. This linear decrease is extrapolated up to a 5.75 m s$^{-1}$ wind speed threshold (Eq. 25a), above which the collection efficiency for snowfall is 0.2 (Eq. 25b), following the general approach of Kochendorfer et al. (2017a). For hydrometeors with mean fall

velocity greater than 1.93 m s$^{-1}$, corresponding to mixed and liquid precipitation, the collection efficiency is 1 (Eq. 25c). The fall velocity threshold was varied over the measurement fall velocity range in 0.01 m s$^{-1}$ increments, with the threshold of 1.93 m s$^{-1}$ found to provide the lowest overall RMSE.

$$CE_{HE1}\left(U_w \leq 5.75 \text{m s}^{-1}, U_f \leq 1.93 \text{m s}^{-1}\right) = 1 - b_1 U_w , \qquad (25a)$$

$$CE_{HE1}\left(U_w > 5.75 \text{m s}^{-1}, U_f \leq 1.93 \text{m s}^{-1}\right) = 0.2 , \qquad (25b)$$

$$CE_{HE1}\left(U_f > 1.93 \text{m s}^{-1}\right) = 1 , \qquad (25c)$$

The second transfer function, referred to as HE2, adds another dimension to describe the slope of the linear decrease in CE with increasing wind speed: the hydrometeor fall velocity. For mode fall velocity $U_f$ below 2.81 m s$^{-1}$ and wind speed $U_w$ below the threshold value, which is also dependent on the fall velocity, the collection efficiency $CE_{HE2}$ is assumed to decrease linearly with decreasing wind speed for a given hydrometeor fall velocity (Eq. 26a). For mode fall velocity below 2.81 m s$^{-1}$

and wind speed above the threshold value, the collection efficiency is 0.2 (Eq. 26b). For mode fall velocity above 2.81 m s$^{-1}$,

the collection efficiency is equal to 1 (Eq. 26c). The fall velocity threshold was varied over the measurement fall velocity range in 0.01 m s$^{-1}$ increments with the threshold of 2.81 m s$^{-1}$ found to provide the lowest overall RMSE.

$$CE_{\mathrm{HE2}}\left(U_{\mathrm{w}} \leq \frac{0.8}{b_1 - b_2 U_{\mathrm{f}}}, U_{\mathrm{f}} \leq 2.81\mathrm{m\ s}^{-1}\right) = 1 - \left(b_1 - b_2 U_{\mathrm{f}}\right)U_{\mathrm{w}}, \tag{26a}$$

$$CE_{\mathrm{HE2}}\left(U_{\mathrm{w}} > \frac{0.8}{b_1 - b_2 U_{\mathrm{f}}}, U_{\mathrm{f}} \leq 2.81\mathrm{m\ s}^{-1}\right) = 0.2, \tag{26b}$$

$\quad CE_{\mathrm{HE2}}\left(U_{\mathrm{f}} > 2.81\mathrm{m\ s}^{-1}\right) = 1, \tag{26c}$

## 5.4 Assessment of transfer functions: collection efficiency

Observed collection efficiencies were compared with adjusted values using both existing transfer functions from SPICE and those presented in this work. Results are presented in Fig. 13, with relevant transfer function parameters compiled in Tables 5 and 6, and resulting bias errors, root mean square errors, and correlation coefficients (r) presented in Table 8. To further

contextualize the assessment of the different transfer functions, the RMSE results are presented for different precipitation classifications, temperature ranges, and fall velocity ranges in Table 9.

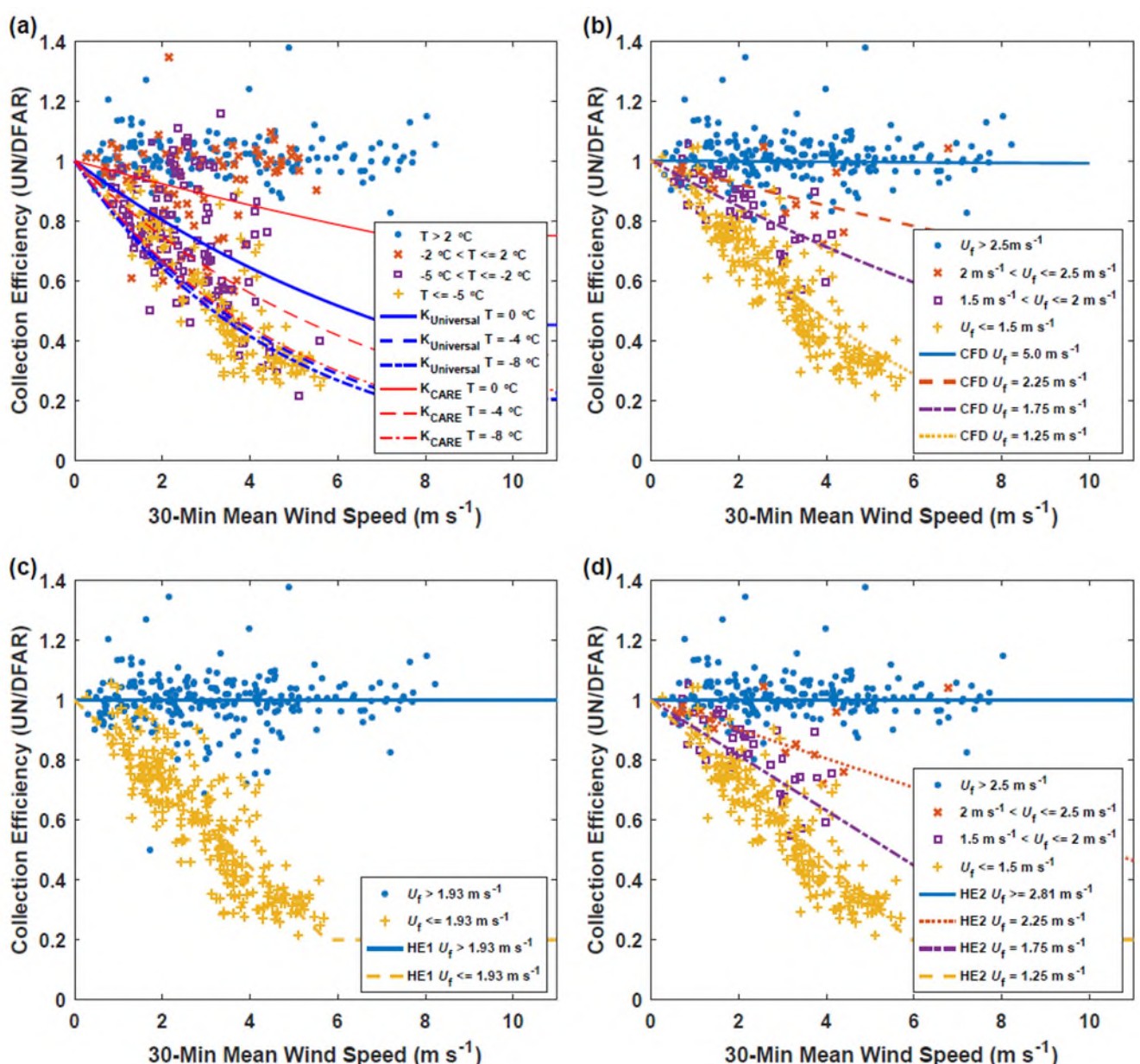

**Figure 13.** Collection efficiency of unshielded gauge as a function of wind speed for: (a) mean air temperature $T$ categories for the $K_{Universal}$ and $K_{CARE}$ transfer functions; (b) mode fall velocity $U_{f\_mode}$ categories with the CFD transfer function; (c) mean fall velocity $U_{f\_mean}$ categories for the HE1 transfer function; and (d) mode fall velocity $U_{f\_mode}$ categories with the HE2 transfer function.


**Table 8.** Unshielded gauge 30-minute event bias error (BE), root mean square error (RMSE), correlation coefficient (r), and number of events (N) for collection efficiency and precipitation accumulation between the unshielded and reference DFAR shielded Geonor T-200B3 gauge for: unadjusted comparison; $K_{Universal}$ transfer function with wind speed and air temperature dependence; $K_{CARE}$ transfer function with wind speed and air temperature dependence; present study CFD transfer function with wind speed and mode fall velocity dependence; HE1 transfer function with wind speed and mean fall velocity dependence; and HE2 transfer function with wind speed and mode fall velocity dependence. Statistics are based on the comparison of experimental results from the CARE site between November 1 and April 30, 2013/14 and 2014/15.

| Description | Collection efficiency | | | Precip accum (mm) | | | N |
|---|---|---|---|---|---|---|---|
| | BE | RMSE | r | BE | RMSE | r | |
| Unadjusted | - | - | - | -0.13 | 0.24 | 0.900 | 514 |
| $K_{Universal}$ | 0.07 | 0.15 | 0.853 | 0.07 | 0.20 | 0.949 | 514 |
| $K_{CARE}$ | -0.005 | 0.12 | 0.878 | 0.002 | 0.13 | 0.963 | 514 |
| CFD | -0.02 | 0.08 | 0.949 | 0.011 | 0.08 | 0.986 | 514 |
| HE1 | 0.0004 | 0.10 | 0.928 | 0.006 | 0.09 | 0.983 | 514 |
| HE2 | -0.009 | 0.08 | 0.950 | 0.006 | 0.07 | 0.988 | 514 |

**Table 9.** Unshielded gauge 30-minute event collection efficiency RMSE results stratified by: (a) POSS precipitation type; (b) temperature; and (c) fall velocity. Results are shown for: $K_{Universal}$ transfer function with wind speed and air temperature dependence; $K_{CARE}$ transfer function with wind speed and air temperature dependence; present study CFD transfer function with wind speed and mode fall velocity dependence; HE1 transfer function with wind speed and mean fall velocity dependence; and HE2 transfer function with wind speed and mode fall velocity dependence. Statistics are based on the comparison of experimental results from the CARE site between November 1 and April 30, 2013/14 and 2014/15.

| | RMSE | | | |
|---|---|---|---|---|
| (a) | Rain | Mixed | Undefined | Snow |
| Description | (N = 196) | (N = 45) | (N = 40) | (N = 233) |
| $K_{Universal}$ | 0.17 | 0.27 | 0.09 | 0.09 |
| $K_{CARE}$ | 0.12 | 0.20 | 0.13 | 0.11 |
| CFD | 0.08 | 0.09 | 0.09 | 0.09 |
| HE1 | 0.07 | 0.16 | 0.08 | 0.10 |
| HE2 | 0.08 | 0.10 | 0.09 | 0.08 |
| (b) | $T > 2\ °C$ | $-2\ °C < T \leq 2\ °C$ | $-5\ °C < T \leq -2\ °C$ | $T \leq -5\ °C$ |
| Description | (N = 150) | (N = 89) | (N = 134) | (N = 141) |
| $K_{Universal}$ | 0.08 | 0.19 | 0.21 | 0.11 |
| $K_{CARE}$ | 0.07 | 0.13 | 0.17 | 0.10 |
| CFD | 0.09 | 0.08 | 0.08 | 0.09 |
| HE1 | 0.07 | 0.10 | 0.11 | 0.10 |
| HE2 | 0.09 | 0.08 | 0.07 | 0.08 |
| (c) | $U_f > 2.5\ m\ s^{-1}$ | $2\ m\ s^{-1} < U_f \leq 2.5\ m\ s^{-1}$ | $1.5\ m\ s^{-1} < U_f \leq 2\ m\ s^{-1}$ | $U_f \leq 1.5\ m\ s^{-1}$ |
| Description | (N = 212) | (N = 15) | (N = 40) | (N = 247) |
| $K_{Universal}$ | 0.19 | 0.23 | 0.16 | 0.09 |
| $K_{CARE}$ | 0.13 | 0.17 | 0.12 | 0.11 |
| CFD | 0.08 | 0.10 | 0.08 | 0.09 |
| HE1 | 0.08 | 0.13 | 0.15 | 0.10 |
| HE2 | 0.08 | 0.12 | 0.08 | 0.08 |

Both $K_{Universal}$ and the climate-specific $K_{CARE}$ transfer function have continuous temperature dependence and display similar profiles at -8 °C, with the collection efficiency for the $K_{CARE}$ transfer function decreasing more gradually with wind speed compared to the $K_{Universal}$ transfer function at -4 °C and 0 °C (Fig. 13a). Using the approach outlined in Sect. 4.4, a temperature threshold $T_t$ of 1.33 °C for the best-fit $K_{CARE}$ transfer function was found to minimize the precipitation accumulation RMSE. The overall collection efficiency root mean square error is reduced from 0.15 for the $K_{Universal}$ transfer function to 0.12 for the $K_{CARE}$ transfer function (Table 8). The bias error is also reduced from 0.07 for the $K_{Universal}$ transfer function to -0.005 for the best-fit $K_{CARE}$ transfer function. For $K_{Universal}$ and $K_{CARE}$, respectively, the RMSE is reduced from 0.17 to 0.12 for rain and from 0.27 to 0.20 for mixed precipitation, with slightly elevated RMSE from 0.09 to 0.13 for undefined precipitation and 0.09 to 0.11 for snow (Table 9a). For mean event temperatures between -2 °C and 2 °C, and between -5 °C and -2 °C, respectively, the RMSE values of 0.19 and 0.21 for the $K_{Universal}$ transfer function are relatively large compared to the 0.13 and 0.17 values for the $K_{CARE}$ transfer function (Table 9b). This results from the more gradual decrease in the $K_{CARE}$ transfer function with wind speed over these temperature ranges (Fig. 13a).

A comparison of the CFD transfer function with observed CE is shown in Fig. 13b. Overall, the measured data have less scatter when stratified by fall velocity than when stratified by temperature (Table 8, Figs. 13a and b). The CFD transfer function provides a lower overall RMSE (0.08) and higher r (0.949) relative to the $K_{Universal}$ and $K_{CARE}$ transfer functions based on temperature. Reductions in the collection efficiency RMSE using the CFD transfer function are most pronounced for rain and mixed precipitation (Table 9a) and for mean event temperatures between -2 °C and 2 °C and between -5 °C and -2 °C (Table 9b) compared with the $K_{Universal}$ and $K_{CARE}$ functions. Collection efficiency RMSE values are between 0.08 and 0.10 over all fall velocity classes, despite fewer numbers of events with fall velocities between 1.5 m s$^{-1}$ and 2.5 m s$^{-1}$ (Table 9c).

The HE1 transfer function provides good agreement with observed data in the mean fall velocity regimes relevant to snow and rain (Fig. 13c), resulting in an overall RMSE of 0.10, BE of 0.0004, and r of 0.928 (Table 8). The RMSE for mixed precipitation is 0.16, which is lower than that of the $K_{CARE}$ transfer function with temperature (0.20) but higher that that of the CFD model (0.09), which varies continuously with fall velocity (Table 9a).

The HE2 function better captures the observed collection efficiencies for mode fall velocities between the snow and rain regimes (Fig. 13d), improving the overall RMSE to 0.08 and r to 0.95, while increasing slightly the BE (-0.009) relative to HE1 (Table 8). Note the distinction between mean fall velocity for HE1 and mode fall velocity for HE2 (and CFD). In general, the Doppler frequency spectrum tends to be skewed such that mode fall velocities are slightly lower than the mean fall velocities, impacting the fits to observed data. The HE2 transfer function provides similar results to that of the CFD transfer function, with slightly higher RMSE values for mixed precipitation and slightly reduced RMSE values for snow (Table 9a) and temperatures below -2 °C (Table 9b). For intermediate fall velocities between 2.0 m s$^{-1}$ and 2.5 m s$^{-1}$, the HE2 transfer function, with a linear change in collection efficiency with fall velocity, has a higher RMSE (0.12) than that for the CFD function (0.10), which exhibits a nonlinear change in collection efficiency with fall velocity (Table 9c). Only 15 events were recorded in this intermediate fall velocity range with higher uncertainty relative to the CFD function. In contrast, 212 events

were recorded at fall velocities above 2.5 m s$^{-1}$ and 247 events at fall velocities below 1.5 m s$^{-1}$, representing a greater proportion of the events with lower RMSE relative to the CFD function.

**5.5 Assessment of transfer functions: precipitation accumulation**

The unadjusted and adjusted accumulated precipitation values are compared with reference DFAR accumulation measurements in Fig. 14. Bias, RMSE, and correlation coefficient results are shown in Table 8. Similar to the approach for assessing transfer functions based on collection efficiency results in Sect. 5.4, the precipitation accumulation RMSE results for each transfer function are assessed by precipitation classification, temperature range, and fall velocity range in Table 10.

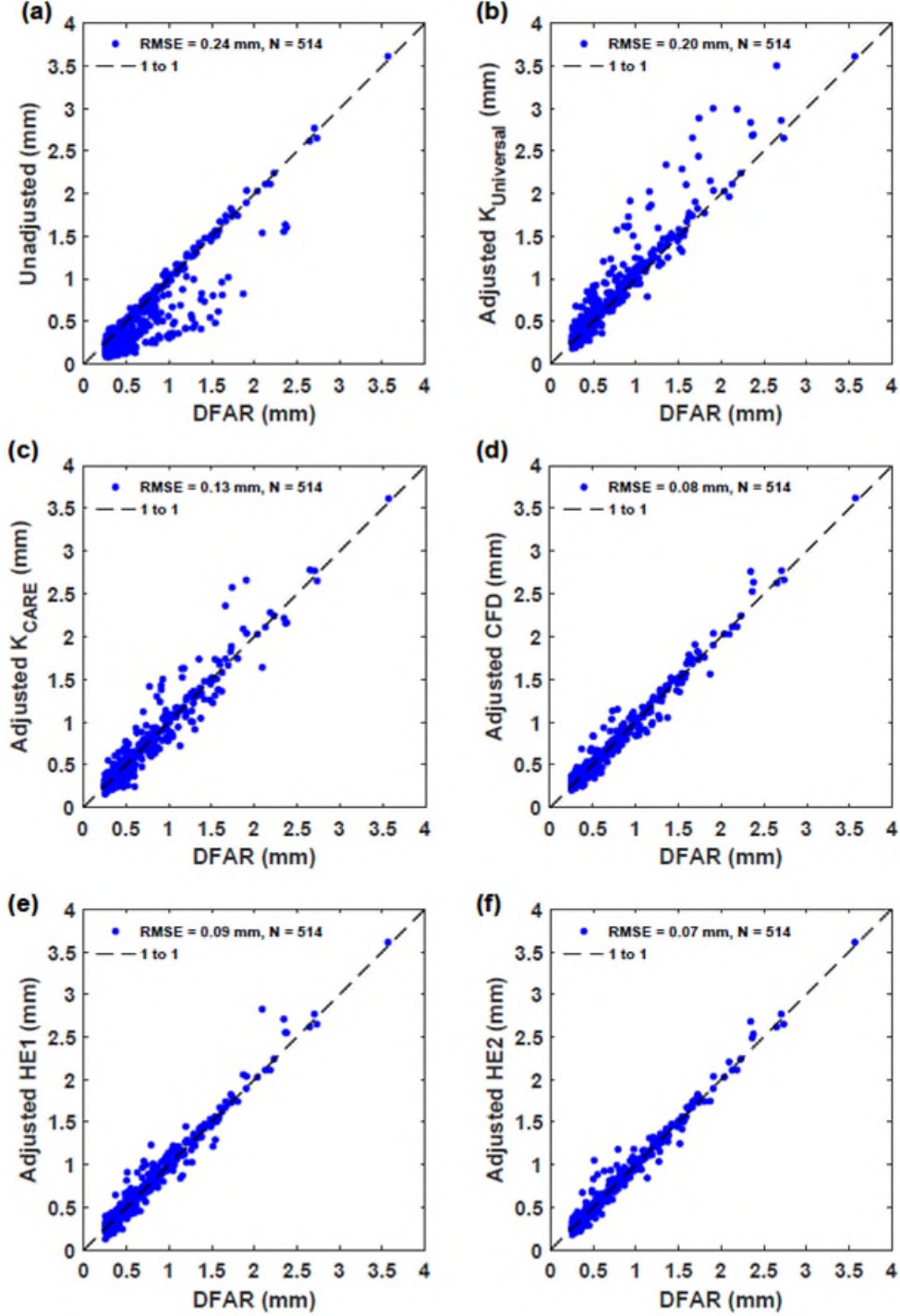

**Figure 14.** Unshielded and reference DFAR 30-minute event precipitation accumulation comparison for: (a) unadjusted precipitation accumulation; (b) $K_{Universal}$ continuous transfer function with wind speed and air temperature dependence; (c) $K_{CARE}$ continuous transfer function with wind speed and air temperature dependence; (d) CFD transfer function with wind speed and fall velocity dependence; (e) HE1 transfer function with wind speed and fall velocity dependence; and (f) HE2 transfer function with wind speed and fall velocity dependence.

**Table 10.** Unshielded gauge 30-minute event RMSE (mm) results stratified by: (a) POSS precipitation type; (b) temperature; and (c) fall velocity. Results are shown for: unadjusted comparison; $K_{Universal}$ transfer function with wind speed and air temperature dependence; $K_{CARE}$ transfer function with wind speed and air temperature dependence; present study CFD transfer function with wind speed and mode fall velocity dependence; HE1 transfer function with wind speed and mean fall velocity dependence; and HE2 transfer function with wind speed and mode fall velocity dependence. Statistics are based on the comparison of experimental results from the CARE site between November 1 and April 30, 2013/14 and 2014/15.

| | RMSE (mm) | | | |
|---|---|---|---|---|
| (a) | Rain | Mixed | Undefined | Snow |
| Description | (N = 196) | (N = 45) | (N = 40) | (N = 233) |
| Unadjusted | 0.04 | 0.15 | 0.09 | 0.35 |
| $K_{Universal}$ | 0.25 | 0.33 | 0.05 | 0.10 |
| $K_{CARE}$ | 0.14 | 0.22 | 0.06 | 0.11 |
| CFD | 0.04 | 0.07 | 0.04 | 0.11 |
| HE1 | 0.04 | 0.17 | 0.04 | 0.10 |
| HE2 | 0.04 | 0.09 | 0.04 | 0.09 |
| (b) | $T > 2$ °C | $-2$ °C $< T \leq 2$ °C | $-5$ °C $< T \leq -2$ °C | $T \leq -5$ °C |
| Description | (N = 150) | (N = 89) | (N = 134) | (N = 141) |
| Unadjusted | 0.04 | 0.14 | 0.23 | 0.39 |
| $K_{Universal}$ | 0.05 | 0.25 | 0.29 | 0.12 |
| $K_{CARE}$ | 0.04 | 0.11 | 0.20 | 0.12 |
| CFD | 0.05 | 0.06 | 0.08 | 0.11 |
| HE1 | 0.04 | 0.12 | 0.09 | 0.10 |
| HE2 | 0.05 | 0.07 | 0.08 | 0.09 |
| (c) | $U_f > 2.5$ m s$^{-1}$ | $2$ m s$^{-1} < U_f \leq 2.5$ m s$^{-1}$ | $1.5$ m s$^{-1} < U_f \leq 2$ m s$^{-1}$ | $U_f \leq 1.5$ m s$^{-1}$ |
| Description | (N = 212) | (N = 15) | (N = 40) | (N = 247) |
| Unadjusted | 0.04 | 0.06 | 0.16 | 0.34 |
| $K_{Universal}$ | 0.26 | 0.22 | 0.22 | 0.10 |
| $K_{CARE}$ | 0.15 | 0.14 | 0.15 | 0.11 |
| CFD | 0.04 | 0.05 | 0.06 | 0.10 |
| HE1 | 0.04 | 0.06 | 0.16 | 0.10 |
| HE2 | 0.04 | 0.06 | 0.07 | 0.09 |

In the comparison of unadjusted accumulation measurements with reference values (Fig. 14a), some values fall along the 1-to-1 line, while others are considerably lower. The values along the 1-to-1 line generally correspond to rain events with high precipitation fall velocity, or to events with low mean wind speeds. The RMSE for the unadjusted unshielded gauge measurements relative to the DFAR is 0.24 mm, with a bias error of -0.13 mm and correlation coefficient of 0.900 (Table 8). Using the $K_{Universal}$ transfer function, with wind and temperature dependence, shifts the adjusted values up to and above the 1-to-1 line (Fig. 14b). This yields a positive bias error of 0.07 mm, reduced RMSE of 0.20 mm, and correlation coefficient of 0.949 (Table 8) relative to the unadjusted measurements (Fig. 14a). While the $K_{Universal}$ transfer function greatly reduces the RMSE for snow from 0.35 mm to 0.10 mm compared with unadjusted values, the RMSE is increased from 0.04 mm to 0.25 mm for rain, and from 0.15 mm to 0.33 mm for mixed precipitation (Table 10a). Compared with the unadjusted results, RMSE

increases for the $K_{Universal}$ function are also apparent for temperatures between -2 °C and 2 °C and between -5 °C and -2 °C (Table 10b), and for fall velocities greater than 1.5 m s$^{-1}$ (Table 10c).

Applying the site-specific $K_{CARE}$ transfer function, based on the best-fit results to the CARE SPICE dataset, results in a reduced bias error of 0.002 mm, lower RMSE of 0.13 mm, and higher correlation coefficient of 0.963 (Table 8) relative to the $K_{Universal}$ results, with the scatter in adjusted accumulations more evenly balanced across the 1-to-1 line (Fig. 14c). The scatter in adjusted values using the $K_{CARE}$ transfer function results primarily from mixed precipitation (Table 10a) at temperatures between -5 °C and -2 °C (Table 10b). Compared to the $K_{Universal}$ transfer function, the $K_{CARE}$ transfer function has lower RMSE values for rain (0.14 mm) and mixed precipitation (0.22 mm), with 0.01 mm higher RMSE for undefined precipitation and snow (Table 10a). The more rapid increase in collection efficiency with temperature for $K_{CARE}$ relative to $K_{Universal}$ reduces the overadjustment of some of the rain and mixed precipitation events at temperatures between -5 °C and -2 °C, at the expense of the underadjustment of some snow events in this temperature range. It is also worth noting that the adjusted precipitation accumulation RMSE for the $K_{CARE}$ transfer function is larger than that for unadjusted results for rain and mixed precipitation, similar to the results for $K_{Universal}$. Both the $K_{Universal}$ and $K_{CARE}$ transfer functions with temperature show signs of heteroscedasticity, with an increased spread of values with increasing magnitude of event precipitation accumulation.

Applying the CFD transfer function results in a greatly reduced spread of values about the 1-to-1 line (Fig. 14d). The spread does not appear to increase with increasing precipitation accumulation. The overall RMSE is reduced to 0.08 mm, 2.5 times lower than that for the $K_{Universal}$ transfer function, with a bias error of 0.011 mm and correlation coefficient of 0.986 (Table 8). The RMSE is reduced from 0.25 mm for the $K_{Universal}$ transfer function to 0.04 mm using the CFD transfer function for rain, and from 0.33 mm to 0.07 mm (4.7 times lower) for mixed precipitation, while RMSE results for undefined precipitation and snow are within 0.01 mm (Table 10a). Reductions in the RMSE using the CFD transfer function compared with the $K_{Universal}$ transfer function are most pronounced for mean event temperatures between -5 °C and 2 °C (Table 10b). Over this temperature range, rain, mixed precipitation, and snow may be present, corresponding to a wide range of fall velocities and collection efficiencies. The CFD transfer function is better able to distinguish among these precipitation types – and their respective collection efficiencies – based on its dependence on hydrometeor fall velocity. Across the fall velocity classifications in Table 10c, the RMSE using the CFD transfer function increases from 0.04 mm for fall velocities greater than 2.5 m s$^{-1}$ to 0.10 mm for fall velocities less than 1.5 m s$^{-1}$. As shown in Table 10c, the RMSE for the CFD transfer function matches the value for unadjusted measurements at fall velocities greater than 2.5 m s$^{-1}$, where collection efficiencies are close to 1. At lower fall velocities, where the bias due to gauge undercatch is more prevalent, the RMSE values for the CFD function are lower than those for the unadjusted measurements.

Using the HE1 transfer function results in similar overall improvement in the agreement between adjusted and DFAR accumulation values as observed for the CFD function (Fig. 14e). The adjusted values appear to be distributed symmetrically about the 1-to-1 line. Furthermore, there is close agreement over the full range of accumulation values; that is, the spread in values does not increase with the magnitude of precipitation accumulation. This results in a lower RMSE of 0.09 mm and a

higher correlation coefficient of 0.983 relative to the $K_{CARE}$ transfer function results. While the RMSE for rain (0.04 mm) using the HE1 transfer function is improved compared with the $K_{CARE}$ transfer function results, the RMSE for mixed precipitation is only marginally better (0.17 mm).

Applying the HE2 transfer function provides further improvement, with adjusted accumulation values more tightly clustered around the 1-to-1 line (Fig. 14f). The overall RMSE is 0.07 mm, which is 3.3 times lower than that for the unadjusted unshielded gauge measurements, and 1.8 times lower than the $K_{CARE}$ transfer function based on mean event temperature and wind speed. The HE2 transfer function exhibits the lowest overall RMSE for snow (0.09 mm), with a RMSE of 0.09 mm for mixed precipitation, which is slightly higher than that for the CFD function (0.07 mm), but much lower than that for the $K_{CARE}$ (0.22 mm) and HE1 (0.17 mm) transfer functions. Further, the correlation coefficient of 0.988 is the highest among the transfer functions assessed.

## 6 Discussion

### 6.1 Modelling discussion

#### 6.1.1 Numerical modelling results

The time-averaged numerical model describes the three-dimensional airflow around the unshielded Geonor gauge, including the updraft above the leading edge of the gauge orifice and downdraft at the back of the gauge orifice shown in previous studies (Thériault et al., 2012;Colli et al., 2016a;Baghapour et al., 2017). The updraft velocity increases sharply with height above the leading edge of the gauge orifice, which appears to play an important role in the horizontal spreading and capture of hydrometeors, particularly for lower fall velocity hydrometeors (Fig. 2 and 4). These velocities scale with the wind speed, as shown previously by Colli et al. (2016a), and the relative magnitudes of the wind speed and hydrometeor fall velocity influence the collection efficiency. The hydrometeor fall velocity influences both the free-stream approach angle of hydrometeors before they encounter the local airflow around the gauge and the degree of coupling between the hydrometeor trajectories and the local airflow. Hydrometeors with fall velocities above 2 m s$^{-1}$ fall more vertically, and their paths shows less deviation with the updraft and local airflow around the gauge orifice (Fig. 4). Hydrometeors with lower fall velocities have a smaller approach angle and are more closely coupled to the local airflow around the gauge orifice.

The model was validated by comparison with existing models (Fig. 3), with differences attributed in part to differences in model geometry. Peak normalized velocities show very good agreement with Baghapour et al. (2017) $k$-$\omega$ SST simulation results, which used a similar refined orifice thickness as that in the present study. Reductions in the peak gauge centerline velocity relative to the results from Colli (2016b) in Fig. 3 may result from refinements in the gauge geometry used in the present study, including the orifice thickness. This reasoning is supported by Sevruk et al. (1994), who demonstrated that reductions in the orifice thickness can reduce the local airflow velocity over the gauge. The sensitivity of simulation results to the model geometry is discussed in greater detail by Baghapour et al. (2017) and is an area for future study.

The numerical results for this study are based on a 5 % inlet turbulence value that acts as a bulk turbulence in the atmosphere (Panofsky and Dutton, 1984) buy may underestimate experimental results (Armitt and Counihan, 1968). A no-slip boundary condition was modelled at the surface following the approach of previous studies (Baghapour and Sullivan, 2017;Colli et al., 2016b). Further study with a no-slip wall condition under different turbulence conditions could lead to further insights into the influence of turbulence intensity on precipitation gauge collection efficiency.

### 6.1.2 Collection efficiency based on wind speed and hydrometeor fall velocity

The numerical model results demonstrate that collection efficiencies are similar for different hydrometeor types with different sizes, densities, masses, and drag values (spherical drag model), but similar fall velocities. This enables the characterization of collection efficiency independent of hydrometeor characteristics other than fall velocity, allowing for the broad application of transfer functions with wind speed and fall velocity dependence to various hydrometeor types.

The numerical model results capture the three-dimensional airflow and hydrometeor kinematics and illustrate the reductions in collection efficiency with increasing wind speed and decreasing hydrometeor fall velocity (Fig. 5). A slight nonlinearity in the collection efficiency relationship with wind speed is apparent, with the collection efficiency decreasing more rapidly at lower wind speeds and more gradually at higher wind speeds. This wind speed dependence has been demonstrated in previous studies (Nešpor and Sevruk, 1999;Thériault et al., 2012;Colli et al., 2016a;Baghapour et al., 2017), and is generally attributed to the three-dimensional velocity profile around the gauge influencing the trajectories and catchment of incoming hydrometeors. A strong nonlinear dependence on the hydrometeor fall velocity is apparent in Figs. 5 and 6. Hydrometeors with fall velocities above 5 m/s exhibit collection efficiencies close to 1, while lower hydrometeor fall velocities influence the rate of decrease of collection efficiency with wind speed. Collection efficiency decreases are most pronounced below 2.0 m/s hydrometeor fall velocity, where a wide range of collection efficiencies are possible. This demonstrates the challenge in adjusting liquid, solid, and mixed precipitation accumulations in situations where different hydrometeor types and sizes – and with very different fall velocities – can occur. These findings support the conclusions of Thériault et al. (2012), who demonstrated large collection efficiency differences across dry snow and wet snow hydrometeors with different terminal velocities. The present findings also support those of Nešpor and Sevruk (1999), who showed that the wind-induced error increases rapidly for smaller raindrop sizes with lower terminal velocities.

Across rain, ice pellets, wet snow, and dry snow, the numerical collection efficiency results are very similar for hydrometeors with the same fall velocity over a wide range of wind speeds, despite differences in characteristics (size, density, and mass), as shown in Fig. 5. Elevated collection efficiencies for rain compared with wet snow above 4 m s$^{-1}$ wind speed may be due to the higher density of rain relative to wet snow, with hydrometeor inertia playing a role at higher wind speeds. For dry snow with 1.0 m s$^{-1}$ fall velocity, the collection efficiency decreases more rapidly relative to that for wet snow and rain hydrometeors with identical fall velocities above 3 m s$^{-1}$ wind speed. A similar rapid decrease in collection efficiency for dry snow has been demonstrated by Colli (2016b). This decrease may be due to the limitations of the spherical hydrometeor model, which can

overestimate hydrometeor volumes and buoyancies, particularly for non-spherical hydrometeors. Further investigation with non-spherical drag models is recommended as an area for future work.

### 6.1.3 CFD transfer function

The CFD transfer function presented in Eq. 18 (coefficients in Table 5) is based on the computational fluid dynamics results

for an unshielded Geonor T-200B3 600 mm capacity precipitation gauge for wind speeds up to 10 m s$^{-1}$. This transfer function provides a straightforward means of estimating the collection efficiency based on the wind speed and hydrometeor fall velocity. In operational monitoring networks, the hydrometeor fall velocity can be provided by disdrometers (Loffler-Mang and Joss, 2000;Sheppard and Joe, 2000;Bloemink and Lanzinger, 2005;Nitu et al., 2018), vertically pointing Doppler radars (Biral, 2019), or multi-frequency radar techniques (Kneifel et al., 2015). Assessment of these techniques for the measurement of

hydrometeor fall velocity is an area for future work.

The CFD transfer function captures well the nonlinear change in collection efficiency with wind speed and hydrometeor fall velocity observed in the numerical model results across rain, ice pellet, wet snow, and dry snow hydrometeor types (Fig. 5). This expression was derived from simulation results up to 10 m s$^{-1}$ wind speed and should be used with caution at higher wind speeds. Further, this transfer function has not been assessed experimentally for snow above 6 m s$^{-1}$ in the present study for the

CARE dataset. Adjusted precipitation accumulation estimates in this regime, where fall velocities are low and wind speeds are high, can be highly uncertain and should be treated with caution (Smith et al., 2020). Assessment of the transfer function at other sites under such conditions is an area for future work. Application to other gauge or shield combinations should also be investigated, as the flow dynamics around the gauge orifice are dependent on the specific gauge and shield geometry.

The fall velocity cutoff, shown in Fig. 6, corresponds to the fall velocity below which no hydrometeors are captured by the

gauge for a given wind speed. In this case, the hydrometeors are unable to pass through the updraft region and local airflow around the gauge orifice to be captured by the gauge. As the wind speed increases, the fall velocity cutoff increases, and it becomes more difficult for hydrometeors to overcome the updraft velocity and local airflow and be captured. This has important consequences for the integral gauge collection efficiency, as hydrometeors below the fall velocity cutoff in the drop size distribution do not contribute to the total catchment. Previous studies have shown similar results with collection

efficiencies decreasing to zero below a given hydrometeor size for liquid (Nešpor and Sevruk, 1999) and solid hydrometeor types (Thériault et al., 2012;Colli et al., 2016).

The present formulation based on the fall velocity can be applied more broadly across rain and snow types for the unshielded Geonor gauge configuration. These results are based on time-averaged simulations, which provide an estimate of the mean velocities through the domain and have been shown to provide good overall agreement with experimental results (Baghapour

et al., 2017). Further study using LES models, which can better resolve the eddy dynamics and temporal variations in the flow, and under different boundary conditions and turbulence scales representing different site conditions is recommended to better understand the collection efficiency under conditions with high wind speeds and low hydrometeor fall velocities.

### 6.1.4 Integral collection efficiency results

#### 6.1.4.1 Wind speed dependence

The integral collection efficiency decreases nonlinearly with wind speed depending on the hydrometeor type and fall velocity. Large differences in the integral collection efficiency dependence with wind speed are apparent across different hydrometeor types and intensities. Previous studies have shown similar differences across liquid (Nešpor and Sevruk, 1999;Jarraud, 2008) and solid hydrometeor types (Colli et al., 2016b;Colli et al., 2020;Thériault et al., 2012). The fall velocities of snowflakes are generally smaller than those of raindrops; accordingly, the collection efficiency for snowfall at a given wind speed is lower

than that for rainfall (Fig. 8). Similarly, dendrites have lower fall velocities than rimed dendrites and columns and plates, and lower collection efficiency.

The integral collection efficiency results decrease continuously with increasing wind speed as the magnitude of the updraft at the leading edge of the gauge increases, free-stream hydrometeor trajectories decrease, and hydrometeors trajectories become more closely coupled with the local airflow around the gauge. For dendrites, the nonlinearity in the integral collection

efficiency is more pronounced, as collection efficiencies decrease to small but finite values at higher wind speeds. This is due to the smaller number of hydrometeors with sufficient fall velocity to be captured by the gauge at higher wind speeds.

The differences in collection efficiency for different precipitation characteristics (type, habit, precipitation intensity) illustrate the large variability that can be expected when the characteristics or fall velocity are not considered. This variability presents a particular challenge for mixed precipitation conditions, in which the precipitation type may not be well defined and can

change rapidly over time. Both the wind speed and hydrometeor characteristics play important roles in determining collection efficiency. The proposed expression for the collection efficiency as a function of the wind speed and hydrometeor fall velocity (Eq. 18) provides a means of estimating the collection efficiency over different hydrometeor types and intensities, even if the precipitation type is not well defined.

The integral collection efficiency results using the CFD transfer function developed in the present study show good overall

agreement with the results of Colli et al. (2016b) for wet snow and dry snow, as shown in Fig. 8. Integral collection efficiency values in the present study are slightly higher than those of Colli et al. (2016b), who demonstrated that their model results for dry snow slightly underpredict experimental results for temperatures below -4 °C. Differences between the two models may be due to differences in the gauge geometry and hydrometeor drag model, among other factors. The gauge geometry in the present study includes a refined orifice wall thickness and full-length orifice extending down into the gauge housing (Fig. 1).

The peak velocities above the gauge in the present study are similar to those observed by Baghapour et al. (2017), who also used a refined orifice wall thickness and observed reduced peak velocities compared to the results of Colli et al. (2016b) as discussed in Sect. 6.1. Increases in the velocity magnitude over the gauge would be expected to decrease the collection efficiency in a manner similar to that for increased wind speed; hence, the higher peak velocities above the gauge in the results of Colli et al. (2016b) provide one explanation for the lower collection efficiency values observed.

Differences in the hydrometeor drag model may also contribute to differences in results among the two studies. The present study uses a spherical drag model similar to that used by Baghapour and Sullivan (2017), in which the drag coefficient varies based on the relative hydrometeor-to-air velocity over the path of the hydrometeor. The results of Colli et al. (2016b) are based on a constant drag model with a fixed drag coefficient over the hydrometeor path, following the approach of Thériault et al. (2012). Both Colli et al. (2015) and Baghapour and Sullivan (2017) showed that the constant drag model can overestimate the

hydrometeor drag relative to empirical models, thereby reducing the collection efficiency.

The injection of hydrometeors from a horizontal plane in the present study ensures identical horizontal particle densities for each wind speed simulation following the approach of Nešpor and Sevruk (1999) for rain and Baghapour and Sullivan (2017) for snow. Injecting hydrometeors from a vertical plane, as done in previous studies (Colli et al., 2016b), could lead to reduced horizontal particle densities and less certainty in numerical collection efficiency estimates for high wind speeds if the injection

densities are not sufficiently high.

The present study defines the integral collection efficiency based on the ratio of precipitation rate or mass flux of precipitation captured by the gauge to that falling in air, following the approach of Nešpor and Sevruk (1999). This provides a consistent formulation for collection efficiency across rainfall and snowfall types. The collection efficiency defined by Colli et al. (2016b) for wet snow and dry snow is based on a 'volumetric' approach, with the fall velocity term in the integrand omitted. Omitting

the fall velocity neglects the contribution of the rate of fall of hydrometeors to the overall precipitation rate, with higher fall velocities providing a greater precipitation rate. For the dry snow and wet snow comparison shown in Fig. 8 the differences between these two approaches are small.

The use of a continuous collection efficiency expression with wind speed and fall velocity dependence enables the derivation of integral collection efficiencies over intermediate sizes and fall velocities in the hydrometeor size distribution. Collection

efficiencies can be computed at intermediate wind speed values using this approach as well, providing the smooth integral collection efficiency curves shown in Fig. 8. Nešpor and Sevruk (1999) used a similar empirical approach for rain by developing an expression for the partial wind-induced error based on free-stream velocity and drop diameter applicable to Mk2, Hellman and ASTA gauges. The integral collection efficiency results of Colli et al. (2016b) were derived directly from numerical CE results for dry snow and wet snow at discrete sizes and wind speeds.

**6.1.4.2 Precipitation intensity dependence**

Knowledge of the precipitation type, intensity, and wind speed can provide a means for adjusting gauge catchment totals. For rainfall, the precipitation intensity has been shown to be an important parameter for the estimation of integral collection efficiency (Nešpor and Sevruk, 1999;Jarraud, 2008). A gradual increase in integral collection efficiency with precipitation intensity is observed for intensity values above 1 mm h$^{-1}$. Below this intensity, the integral collection efficiency decreases more

rapidly, with the rate of decrease depending on the rainfall type and wind speed. This is in general agreement with the results of Nešpor and Sevruk as presented in Jarraud (2008), who showed a sharper increase in the conversion factor (inverse of integral collection efficiency) below 1 mm h$^{-1}$.

Integral collection efficiencies for snowfall also increase with precipitation intensity, as higher intensities correspond with larger numbers of hydrometeors with higher fall velocities and increased collection efficiencies, as shown by Colli et al. (2020).

Integral collection efficiencies for snowfall can be much lower than for rain, depending on the wind speed. Differences are apparent across different snowfall crystal habits (e.g. dendrites vs. dendrites and aggregates of plates), with the magnitude of differences increasing with wind speed. This illustrates the difficulty of adjusting snowfall measurements if the crystal habit is not known. The range of possible integral collection efficiency values is even larger under conditions when solid, liquid, and mixed precipitation can all be present. An additional challenge is presented by the measurement of low precipitation

intensities for snowfall, where accumulations can be small relative to gauge uncertainties due to environmental factors (e.g. wind, temperature).

### 6.1.4.3 Hydrometeor fall velocity dependence

Integral collection efficiency differences across precipitation types are much smaller when stratified by wind speed and hydrometeor fall velocity (Fig. 10) than when stratified by wind speed and precipitation intensity (Fig. 9) or by wind speed

alone (Fig. 7). This results from the ability of the hydrometeor fall velocity to capture differences in the integral collection efficiency across different hydrometeor types and precipitation intensities. The small differences in collection efficiency across different hydrometeor types with the same fall velocity are attributed to the nonlinearity in the relationship between collection efficiency and fall velocity over the mass-weighted distribution of hydrometeor fall velocities. The results in Fig. 10 follow the general nonlinear profile of the CFD transfer function (Eq. 18, Fig. 6), with the hydrometeor fall velocity defining the

integral collection efficiency magnitude for a given wind speed.

Measurements of fall velocity can be obtained using a number of methods (Sect. 6.1.2), and are increasingly available through the deployment of disdrometers in operational networks. These measurements provide an independent assessment of the hydrometeor fall velocity, and together with gauge height wind speed estimates, can enable the adjustment of gauge precipitation accumulation measurements using Eq. (18). Adjustments using this approach can be applied over a range of

hydrometeor types and even when the hydrometeor type may be unknown or uncertain.

### 6.2 Experimental discussion

Transfer functions were derived using accumulated precipitation amounts reported by automatic weighing precipitation gauges over 30 minute periods. This approach is consistent with that used in SPICE (Nitu et al., 2018) and the related derivation of transfer functions (Kochendorfer et al., 2017a). While automatic precipitation gauges can report at a temporal resolution of

one minute, or even higher, the extension of the transfer function derivation and evaluation to other temporal periods, or different accumulation thresholds, is beyond the scope of this work.

The Kochendorfer et al. (2017a) universal transfer function with wind speed and air temperature dependence, $K_{Universal}$, was derived from measurements at eight SPICE sites in the interest of making the transfer function broadly applicable across different climates. This broad applicability is furthered by the widespread availability of air temperature and wind speed

measurements at meteorological stations. Recent studies have demonstrated that the performance of $K_{Universal}$ can vary substantially by site (Smith et al., 2020). Therefore, climate-specific $K_{CARE}$ transfer function coefficients were also derived for comparison in the present study.

The $K_{CARE}$ transfer function has a lower temperature threshold and exhibits larger increases in collection efficiency with increasing temperature relative to $K_{Universal}$ (Fig. 13a). These differences improved the overall RMSE for $K_{CARE}$ by reducing

the over-adjustment of some rain and mixed precipitation events; however, this improvement came at the expense of under-adjusting some snow events at warmer temperatures. The use of this approach warrants further study over longer periods to better understand the performance impacts of seasonal variability and assessment at other sites and climate regions with different precipitation characteristics and proportions.

Both the $K_{Universal}$ and $K_{CARE}$ transfer functions performed well for snow, but were limited by their ability to distinguish among

snow, rain, and mixed precipitation at temperatures between -5 °C and 2 °C. The largest uncertainties in collection efficiency and adjusted accumulation estimates were observed over this temperature range. Adjustments using wind speed and hydrometeor fall velocity, however, addressed this shortcoming and provided improved collection efficiency and adjusted accumulation estimates. The CFD transfer function, derived from time-averaged numerical simulation results over a wide range of wind speeds and hydrometeor fall velocities, resulted in low RMSE values overall and across rain, snow, mixed, and

undefined precipitation types. These results reinforce the fundamental importance of both wind speed and hydrometeor fall velocity on gauge collection efficiency demonstrated by the CFD model results and results from earlier studies (Nešpor and Sevruk, 1999;Thériault et al., 2012).

The CFD transfer function exhibited the lowest RMSE of all transfer functions for mixed precipitation and for intermediate fall velocities between 1.5 m s$^{-1}$ to 2.5 m s$^{-1}$ (Table 9c), which is attributed to its nonlinear increase in collection efficiency

with fall velocity. As this transfer function was derived theoretically, it is applicable across different sites and climate regimes with different types and relative proportions of hydrometeors. The present results also support the methodology for the CFD model, which can be extended to other shield and gauge combinations. For larger shields, it may be important to employ a more realistic vertical wind profile, with a zero-slip boundary condition at the earth's surface.

The HE1 transfer function showed good results for snow, supporting its use for the unshielded gauge. This approach is

straightforward to implement based on its simplicity, and is less reliant on the accuracy of fall velocity estimates beyond the fall velocity threshold. The collection efficiency for the HE1 transfer function decreases to 0.2 at a wind speed of 5.75 m s$^{-1}$. This demonstrates the challenge of adjusting unshielded gauge snow measurements at windy sites, where the captured accumulations may be small relative to gauge uncertainties. This can lead to large uncertainty in adjusted measurements, as demonstrated by other studies applying transfer functions to unshielded gauge measurements at windy sites (Smith et al.,

2020). The CFD transfer function results suggest a gradual decrease in collection efficiency at higher wind speeds compared with the HE1 transfer function, as some hydrometeors with higher fall velocities are still able to be captured by the gauge; however, these accumulations remain small relative to gauge uncertainties, particularly in windy conditions, making them difficult to assess experimentally. Further testing at other sites is recommended to better understand the collection efficiency

for low fall velocity hydrometeors (light snow) under windy conditions above 6 m s$^{-1}$, which were not available in the CARE

dataset.

A limitation of the HE1 transfer function is the minimal improvement in the RMSE for mixed precipitation and fall velocities between 1.5 m s$^{-1}$ to 2.0 m s$^{-1}$ relative to the K$_{CARE}$ function. This is due to the over-adjustment of mixed precipitation events with fall velocities slightly below the cutoff value, and the under-adjustment of mixed precipitation events with fall velocities slightly above the cutoff. While the RMSE for mixed precipitation is still lower than that for adjustments based on temperature

and wind speed (K$_{Universal}$, K$_{CARE}$), further improvements are obtained by using transfer functions with continuous fall velocity dependence; specifically, the CFD and HE2 transfer functions.

The HE2 transfer function, with a linear increase in collection efficiency with fall velocity, yields a greater reduction in the RMSE for mixed precipitation relative to the HE1 transfer function. The HE2 transfer function results show a higher RMSE for mixed precipitation than those for the CFD function, possibly due to the nonlinearity in the latter with fall velocity. The

HE2 transfer function, however, yields the best RMSE results for snow, temperatures below -5 °C, and fall velocities below 1.5 m s$^{-1}$. Adjusted uncertainties for snow are approximately two times higher than those for rain, and show similar trends with increasing temperature and decreasing fall velocity. The former may be due to the lower event accumulations and greater adjustments for snow relative to rain, with measured values in closer proximity to the gauge uncertainty. The present approach of estimating the fall velocity using the POSS appears to perform well, overall; however, further study to better characterize

the fall velocity distribution and changes over 30-minute time periods could lead to further improvements in the model under specific conditions such as mixed precipitation. While this transfer function was derived using the CARE dataset, it is more universally applicable than adjustments based on temperature, for which the relative proportions of rain, snow, and mixed precipitation at warmer temperatures can influence fit results. Further testing at other sites is recommended to assess this in different climate regions, with different hydrometeor types and associated fall velocities.

**6.3 Application to operational networks**

It is evident that the performance of catchment-type precipitation gauges is dependent on wind speed and the aerodynamic properties of both the gauge and incident hydrometeors (Nešpor and Sevruk, 1999;Thériault et al., 2012;Colli et al., 2016b). The modelling results of this study demonstrated this dependence from a theoretical perspective, resulting in a transfer function that incorporates hydrometeor fall velocity. The experimental results validated this approach, which resulted in improved

precipitation estimates from an unshielded gauge relative to those using surface temperature as a proxy for precipitation phase or type. Indeed, the use of surface temperature in this manner can be instructive (Kienzle, 2008;Harder and Pomeroy, 2013), but does not capture the conditions defining hydrometeor initiation and growth aloft (Stewart et al., 2015).

In this study, the fall velocity of hydrometeors reported by the POSS provided direct measurement of a key parameter related to the aerodynamics of the catchment process. In Canada, the POSS was deployed operationally to report present weather as

part of an automatic weather station. Globally, other types of disdrometers (e.g. OTT Parsivel[2], Thies Laser Precipitation Monitor) have been deployed operationally and can also provide hydrometeor vertical velocities. The uncertainty in fall

velocity estimates for different technologies, hydrometeor types, sizes, fall velocities, wind speeds, and wind directions remains to be assessed. These sensors can also be useful for reporting present weather and verifying the occurrence of precipitation based on their high sensitivity (Nitu et al., 2018;Sheppard and Joe, 2000).

The results from this study demonstrate that the combined use of accumulation reports from an unshielded weighing gauge with fall velocities reported by a disdrometer, wind speed measurements, and an appropriate transfer function can greatly reduce the uncertainty of precipitation accumulation measurements. The extension of the approach in the present study to shielded precipitation gauges or gauge designs with higher sensitivity may provide a means of further reducing the measurement uncertainty for automatic gauges in windy environments. Application to light snow events and different event

durations are other areas for future study.

## 7 Conclusions

Hydrometeors exhibit a wide variety of habits, sizes, shapes, and densities, influencing their aerodynamics and, in turn, their ability to be captured by the gauge. Numerical modelling analysis for an unshielded Geonor T-200B3 600 mm precipitation gauge demonstrated that collection efficiencies are similar for different hydrometeor types with different sizes, densities,

masses, and drag values, but similar fall velocities. The model results illustrated that wind speed influences the updraft magnitude and local airflow around the gauge orifice, while fall velocity affects the approach angle and degree of coupling between the hydrometeor trajectories and the local airflow. An empirical collection efficiency transfer function with wind speed and fall velocity dependence was developed from the model results. Two additional transfer functions with similar dependence were derived experimentally for unshielded Geonor T-200B3 precipitation gauges.

These three collection efficiency transfer functions with gauge height wind speed and precipitation fall velocity dependence were assessed experimentally and compared to universal and climate-specific transfer functions with wind speed and temperature dependence. These functions employ different models to adjust precipitation accumulation measurements for wind-induced undercatch, including:

(1) The nonlinear CFD transfer function model presented in Sect. 3, with collection efficiency decreasing nonlinearly with

wind speed and increasing nonlinearly with precipitation fall velocity;

(2) The HE1 transfer function, with a linear decrease in collection efficiency down to 0.2 with wind speed for 30-minute mean fall velocity below 1.93 m s$^{-1}$, and a collection efficiency of 1 above this fall velocity value;

(3) The HE2 transfer function, with the linear wind speed dependence down to 0.2 collection efficiency, transitioning with increasing mode fall velocity to provide a collection efficiency of 1 when the mode fall velocity reaches 2.81 m s$^{-1}$.

These transfer functions were assessed using accumulation measurements from an unshielded precipitation gauge and DFAR gauge over 30-minute precipitation events during two winter seasons at the CARE test site in Egbert, ON, Canada. Estimates of fall velocity were provided by the POSS upward-facing Doppler radar.

The transfer functions with mean wind speed and fall velocity dependence improved the agreement between the 30-minute adjusted precipitation accumulation values and DFAR reference values relative to the $K_{Universal}$ and $K_{CARE}$ transfer functions

with mean wind speed and air temperature dependence. The CFD transfer function agreed well with experimental results over all observed fall velocities, supporting the use of the numerical modelling approach and providing the lowest RMSE for mixed precipitation. The HE1 transfer function captured the collection efficiency trends for rain and snow well, with the collection efficiency for rain close to 1 and the collection efficiency for snow decreasing with wind speed. The HE2 transfer function better captured the collection efficiency for mixed precipitation with fall velocities between 1.2 m s$^{-1}$ to 4.6 m s$^{-1}$.

The results of this study reinforce the important role of fall velocity on collection efficiency shown in previous studies (Nešpor and Sevruk, 1999;Thériault et al., 2012). Adjustment approaches incorporating fall velocity show tremendous value and potential, particularly where DFAR measurements are not feasible, and can be applied where the precipitation type is complex (e.g. snow transitioning to rain), uncertain, or even unknown. These approaches warrant further investigation at different sites with different precipitation characteristics, fall velocities, and wind speeds. Further study to assess the collection efficiency

relationships with wind speed and precipitation fall velocity for different shield configurations, as well as assessing the fall velocity using other means, including disdrometers or remote sensing, is also recommended.

*Disclaimer.* Many of the results presented in this work were obtained as part of the Solid Precipitation Intercomparison Experiment (SPICE) conducted on behalf of the World Meteorological Organization (WMO) Commission for Instruments and

Methods of Observation (CIMO). The POSS was not included as part of the SPICE intercomparison. The analysis and views described herein are those of the authors, and do not represent the official outcome of WMO-SPICE. Mention of commercial companies or products is solely for the purposes of information and assessment within the scope of the present work, and does not constitute a commercial endorsement of any instrument or instrument manufacturer by the authors or the WMO.

*Author contribution.* J.H. was the lead author and was responsible for the CFD analysis, methodology, analysis, visualization, and manuscript preparation and editing. M.E.E. provided guidance for the methodology, analysis, visualization, and writing – review and editing. P.I.J. provided guidance for the analysis, interpretation of results, visualization, and writing – review and editing. P.E.S. provided guidance for the analysis, interpretation of results, and writing – review and editing.

*Acknowledgements.* The authors would like to acknowledge the encouragement and support of Rodica Nitu for this field of study. Thank-you to Christine Best, Pierrette Blanchard, and Sorin Pinzariu for supporting this work and Brian Sheppard for helpful discussions regarding the POSS. Thank-you to Hagop Mouradian, Sorin Pinzariu, and Lillian Yao for the data logger programming, electrical wiring, site maintenance, data ingest, and quality control for the CARE test site. The authors would also like to thank the WMO-SPICE team for their contributions and for discussions inspiring many facets of this work. We

also thank John Kochendorfer and the anonymous reviewers for providing thoughtful reviews of the original version of this manuscript, and greatly improving the quality of this paper.

*Data availability.* The unshielded and reference event accumulations, wind speed, temperature, mean and mode fall velocity, and precipitation type data used in this study will be made available in a suitable online repository. The flow simulation and collection efficiency results from this study shown in Figs. 3, 5, 7, 8, 9, and 10 will be made available in a suitable online repository.

*Competing interests.* The authors declare that they have no conflict of interest.

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
