# Peer review of "Unshielded Precipitation Gauge Collection Efficiency with Wind Speed and Hydrometeor Fall Velocity"

_Hydrology and Earth System Sciences, 2020_

## Referee Comment (RC1) · Anonymous Referee #1 · 4 Jan 2021

This study proposes that fall speed influences the collection efficiency of unshielded gauge using computation fluid dynamics (CFD). The authors claim that they are using a new method to study gauge collection efficiency and, with this method, that they are the first to demonstrate the impact of fall speed on the gauge collection efficiency. In fact, these have already been done with a similar approach:

1) Theriault et al. (2012), Colli et al. (2016a,b) used CFD to study gauge collection efficiency for snow.

2) Colli et al. (2016a) were the first to compute the flow field near an unshielded gauge as performed in this manuscript.

[Figure]

3) Theriault et al. (2012) found a strong dependence between the gauge collection efficiency and fall speed. Indeed, it was conducted with a shielded gauge but the physical reasons are the same. The updraft upstream of the gauge tends to deviate the slow-falling particles to fall in the gauge. For the same horizontal wind speed, slow-falling snowflakes have lower collection efficiency than faster-falling ones.

4) Colli et al. (2020) used the precipitation intensity as done in this manuscript to adjust the collection efficiency.

Colli, M., Stagnaro, M., Lanza, L. G., Rasmussen, R. and Theriault, J. M. (2020). Adjustments for wind-induced undercatch in snowfall measurements based on precipitation intensity, Journal of hydrometeorology, 21, 1039-1050.

The impact of precipitation intensity on the collection efficiency was also suggested by Chubb et al. (2015) using field measurements.

Chubb, T., Manton, M. J., Siems, S. T., Peace, A. D., & Bilish, S. P. (2015). Estimation of Wind-Induced Losses from a Precipitation Gauge Network in the Australian Snowy Mountains, Journal of Hydrometeorology, 16(6), 2619-2638.

In particular: Section 1: The introduction is very long and the goal is not stated clearly. The literature review is incomplete. What are the authors trying to do exactly? If it is showing that CFD can be used to show the dependence of the collection efficiency on the fall speed, it has already been done before.

Section 2: The simulations described in section 2.1 were already done in Colli et al. (2016a). The collection efficiency computed in section 2.3 were first used in Colli et al. (2020).

Sections 3 and 4: Most results/discussion are not new and/or should be improved for clarity. For example: 1) Sections 4.1, 4.2: Same key findings as in previous studies. 2) Section 4.3: The threshold fall speed value is directly related to the minimum diameter of the size distribution discussed in Theriault et al. (2012) and Colli et al. (2016a, b) and

Colli et al. (2020). Small particles falling slower are deflected by the updraft upstream of the gauge. 3) Section 4: Lines 565-569: It should be corrected as previous studies by Theriault et al. and Colli et al. also used a horizontal plan. Lines 573-577: The volumetric approach is what the gauge measures. When using the fall speed, it is the precipitation intensity as proposed in Colli et al. (2020). Sections 4.4.2, 4.4.3 and 4.4.4: Most of the content are not new findings and are repetitive.

Given those, there is not enough novelty in this manuscript to be published. Since some of the results are needed for Part 2, I recommend merging both manuscripts. A methodology section that explains the CFD simulations should be added to Part 2.

---

## Referee Comment (RC2) · Anonymous Referee #2 · 5 Jan 2021

In this work the authors presented "A new method for assessing collection efficiency using wind speed and hydrometeor fall velocity", but this methodology, based on CFD simulations and Lagrangian particle tracking model have been previously used in the recent literature (e.g. Tériault et al. 2012, Colli et al. 2016a,b). The Geonor precipitation gauge has been studied in these works in both shielded and unshielded configuration.

One of the main conclusion of this work is the relation between "Collection Efficiency" (CE) and the particle fall velocity instead of the particle diameter as shown in Colli et al.

[Figure]

2016b. However, for wet and dry snow they use the relation proposed by Rasmussen et al. 1999 to calculate the particle fall velocity as a function of the particle diameter. Furthermore, in equation 9 and 17 the authors reported the formulas for the "overall Collection Efficiency" for rain and snow respectively. In these equations is highlighted that the fall velocity is a function of the particle diameter (D) and therefore the overall CE depends only on the wind speed and D. For this reason, there is no novelty in this approach.

**L 175**: The authors use the relation proposed by Rasmussen et al. 1999 to calculate the terminal velocity, and they stated that "hydrometeor density was chosen to provide the desired hydrometeor fall velocity", but in the work of Rasmussen et al. the density value relations are provided for both wet and dry snow. How did the authors vary the hydrometeor density? Are these density values realistic? or are they used only to obtain the fall velocity the authors desired? The smaller particle of wet snow has a density value greater than water, is it right?

**Fig. 5 and 6** : in figure 5 the authors showed the "collection efficiency" for different precipitation types and fall velocities respect to wind speed. It is clear from the figure that there are differences in the CE values of different precipitation type but with the same fall velocity. Furthermore, the authors used only part of these data to obtain the "empirical collection efficiency expression" showed in figure 6, but this relation has been used to calculate the "overall Collection Efficiency" for all the particle types. How do this affect the obtained results?

**Sections 3.4.3 and 4.43**: in these sections (Results and Discussion sections) the authors highlight the dependency of overall CE with precipitation intensity. This topic is addressed in the recent work of Colli et al. 2020. Do the authors compare their results with that work?

In general, in this work the authors reproduced methodologies used in previous works and there are no significant improvements or novelty. Furthermore, there are a few points the authors need to clarify, like e.g. the choice of the particle density values and the use of an unique empirical CE relation for different precipitation types and they need to evaluate how these impact on the results.

Reference:
Colli, M., Stagnaro, M., Lanza, L. G., Rasmussen, R. and Thériault, J. M. (2020). **Adjustments for wind-induced undercatch in snowfall measurements based on precipitation intensity**, *Journal of hydrometeorology*, 21, 1039-1050.

---

## Referee Comment (RC3) · John Kochendorfer (Referee) · 13 Jan 2021

**General comments**

Part I of "Unshielded precipitation gauge collection efficiency with wind speed and hydrometeor fall velocity" describes a modelling experiment designed to estimate precipitation undercatch in an unshielded precipitation gauge. The work focuses on the use of hydrometeor fall velocity to create improved transfer functions available to adjust unshielded precipitation measurements. The background and importance of the problem are well described in the introduction, which provides an excellent overview of past work in the modeling of precipitation undercatch. The methods and results are well documented, and the manuscript is generally very well written and easy to follow. The topic of undercatch is an important one, and this work is both new and useful, as it addresses the most difficult outstanding questions in precipitation undercatch; the manuscript establishes a valid way to reduce the significant uncertainty that precipitation transfer functions suffer from, and future work may also prove that this new approach can help reduce the site-to-site variability of collection efficiency and the resultant biases and uncertainty.

There are a couple of methodological points which need to be explored or explained more fully. These are described in more detail in the specific comments below, but I find the unrealistic background surface layer atmospheric flow problematic. In addition, the concept of a wind speed threshold above which collection efficiency is equal to zero is both impractical, and in my opinion theoretically unsound. However, I am not proposing that the entire model be redesigned, as it is certainly a valuable study as-is, especially as demonstrated by the accompanying Part II of this manuscript. I would however like to see these shortcomings handled differently within the manuscript.

After completing my review, I read the reviews from Referees #1 and #2, and feel compelled to write that I disagree with their main point, which is that these manuscripts are not novel enough to merit publication. I am ambivalent about whether or not they need to be published as two separate papers; I will leave that up to the editor. However, I maintain that the main point of this work, which is the inclusion of the fall velocity in a transfer function, is indeed both new and useful.

Theriault et al. (2012) includes a transfer function with a snowflake type parameter in it, but not the hydrometeor fall velocity. While Theriault et al. (2012) helped demonstrate the connection between hydrometeor fall speed and catch efficiency, and in general the importance of snowflake type, it did not include an easily applicable method for the improvement of operational precipitation measurements. While crystal type and hydrometeor fall velocity are certainly linked, as both manuscripts demonstrate, the use of the hydrometeor fall velocity, which can be measured relatively reliably and automatically, is important as a characteristic separate from the crystal type. All hydrometeors (not just snowflakes) have a measurable fall velocity, and as demonstrated by the present manuscripts under review, this fall velocity can be used to improve the collection efficiency transfer function. This is new. None of the references offered by Reviewer #1 and Reviewer #2 demonstrate a transfer function that includes the hydrometeor fall velocity. Nor for that matter, in my opinion, do any of those papers offer practical improvements to the currently available transfer functions that can be applied in an operational network. It is also worth noting that most of the important papers that Reviewer #1 and Reviewer #2 cite as evidence of the lack of novelty in the present paper were already cited in the present paper; it is

not as if the authors of the paper under review were hiding the fact that this past work existed, or that it influenced their own work.

It is also worth noting that the use of the fall velocity is very different from the use of precipitation intensity for the improvement of collection efficiency transfer functions. While there may be some general correlation between precipitation intensity and hydrometeor type, precipitation intensity is not a good proxy for hydrometeor type, and in fact has real limitations for use in collection efficiency transfer functions. One of the most significant of these limitations is the fact that both precipitation intensity and collection efficiency are heavily dependent on the same precipitation measurement; they are not independent variables, and in such a case it is easy to demonstrate correlations that have no real or physical relevance.

**Specific comments**

Ln. 53. Explain what is meant by, "a sharper decay and higher intercept of a negative exponential distribution." The decay is with respect to what? This actually does bring to mind an altered curve, although I'm not sure if I am seeing it correctly. Anyway, I wouldn't write something like this and expect my readers to be able to understand it. In addition, I have no idea what are on the x- and y- axes of this imagined curve.

Ln. 147. Why was the ground modeled as a frictionless wall? I am afraid I may be climbing up onto the soapbox here. However, I maintain that is not a 'get off my lawn' comment, because modeling atmospheric flow is not really my specialty. I know others have modeled gauge catch efficiency using the same boundary condition. But it results in an unrealistic vertical wind speed profile, in which the horizontal wind does not decrease with height, and is not zero at the ground. Just because others have done it, does not mean it makes sense. Especially when modeling a large shield (which is admittedly not the case here), a realistic vertical wind speed profile is needed to simulate realistic flow over the shield. But more importantly, without a zero-slip boundary condition at the surface, the model will not generate realistic background turbulence; in neutral atmospheric conditions, turbulence near the surface is generated by wind shear. With a frictionless surface there will presumably be no wind shear, and also no background turbulence. To clarify, I am not talking about the turbulence created by the gauge, but by the surface of the earth. This 'normal' background surface layer turbulence is important because it affects the flow over the gauge and the hydrometeors falling towards the gauge. In real life, the atmospheric flow at the earth's surface is not laminar. The assumption that undercatch can be modeled accurately in laminar background atmospheric flow should at least be discussed, along with the possible shortcomings.

Table 1. $u_w$ hasn't been defined yet. Or if it has, I can't find it. Also, I find this a confusing choice as the symbol for the free stream wind speed. This is because w is often used for the vertical wind speed, and because $u_x$, $u_y$, and $u_z$ are also used to describe different components of the wind velocity; $u_w$ looks to me like another way to describe the vertical wind speed.

Ln. 198. Based on the statement that hydrometeor interactions were ignored (ln. 188), I am guessing that "interactions *within* the gauge orifice" should be changed to, "interactions *with* the gauge orifice."

Ln. 285. The way this is currently written it could be misinterpreted to mean that u* is the free-stream wind speed, not the, "peak velocity along the gauge centerline normalized by the free-stream wind speed." Perhaps the normalization could be moved to the end of the sentence – this sort of normalization is to be expected anyway, so I would argue that it isn't a critical part of the definition. "Peak velocities along the gauge centerline (u*) are compared… in Fig. 3, with the centerline velocities normalized by the free-stream wind speed." Maybe? Also, I find u* a confusing choice, as ustar ($u_*$) is an often-used variable with a completely different and well-established usage.

Figure 3. I believe the y-axis should be labeled u*, not z*. Also include $u_w$ (or its replacement!) in the caption in parenthesis after, "normalized free-stream velocity" to help clarify the meaning of the panel (a) and (b) titles.

Figure 4. This is an excellent figure. I suspect we will see it reference and recycled many times, in future presentations.

Figure 5. Small issue, but the legend shows open yellow squares for ice pellets, and the plot shows closed yellow squares ($u_f$ = 5 m s$^{-1}$).

Ln. 320. Clarify by changing "hydrometeors up to about 3 m s$^{-1}$ wind speed" to, "hydrometeors for horizontal wind speeds up to about 3 m s$^{-1}$". I was confused by all the different speeds in this sentence.

Ln. 311 – 324. Some explanation of why the "dry snow" results are so unrealistic is needed. Experimental collection efficiencies are never this low (or zero). Is your hypothesis that this is because pure "dry snow" rarely occurs? Or is it because the experimental collection curves are derived wrong? I will say more about this elsewhere, but I find the suggestion that collection efficiency drops to zero problematic (and impractical). I suspect that it may be due to the fact that the modeled background flow is not turbulent. In the real world, surface layer flow and particle dispersion are stochastic processes. Given enough time or water, some hydrometeors will always be forced into the gauge by an errant eddy, no matter how slowly they fall or how high the wind speed is. The trajectories in Figure 4 are fine for what they are, but they show how hydrometeors behave in a laminar wind tunnel, not in actual turbulent surface layer flow. Turbulence intensity typically increases faster than the mean wind speed near the land surface, so it actually becomes more important as the wind speed increases. This may be why most experimental results reveal a sigmoid or exponential response of collection efficiency to wind speed, with the sensitivity of collection efficiency to increasing wind speed decreased (with the sigmoid function becoming flat, or unchanging with respect to wind speed) at high wind speeds.

Ln 335, Eq. 18. Would it be possible to derive a collection efficiency equation, or its functional form, from the equations used within the model? I am a little disappointed that a modeling paper relies on an empirical equation.

Ln. 344 – 345. I am again flummoxed by this concept that collection efficiency = zero at some point. What purpose does it serve? Is there any measurement evidence to support it? And how does one correct a precipitation even that occurs when the collection efficiency is defined as zero? I believe that the introduction of this zero-collection-efficiency concept and the emphasis placed on it in this paper

may confuse others and hinder future progress in collection efficiency research. I grant that at low temperatures and high winds, an unshielded gauge can fail to measure any precipitation, but that is in part because most 30-min snowfall 'events' are near the measurement threshold of the gauge, in the 0 – 0.4 mm range. But just because we can't always measure it, doesn't mean it is zero. And if collection efficiency is defined as zero by the transfer function, how to we apply this function when precipitation is measured under these conditions. In a large enough dataset, we will be very hard pressed to find any commonly-occurring environmental conditions under which the reference catches precipitation and the unshielded gauge NEVER catches precipitation. But this is indeed what this theory prescribes, that there are certain conditions under which it is impossible for an unshielded gauge to collect certain hydrometeor types. That is very tall claim. The existence of such conditions in the real world should be demonstrated before making zero collection efficiency a central part of the theory. At a minimum, the discrepancies between past experimental results and the modeled results should be discussed.

Figure 6 and ln 349 – ln. 352. If I understand correctly, these results were produced using Equation 9 and the fall velocity, not the more complex precipitation characteristics. So why was only wet snow shown (or discussed) at $u_f$ = 1.5 m s$^{-1}$? In theory, the same transfer function would be used for different precipitation types, given the same fall velocity. But not all the precipitation types are shown or discussed. Why aren't all the collection efficiency curves shown in Figure 5 shown here? Was the figure too busy? In all honestly, initially I was confused, and thought that only wet snow was modeled at $u_f$ = 1.5 m s$^{-1}$, but I believe I understand now that these results should be equally valid for all precipitation types, as they are purely a function of fall velocity.

Ln. 389. Clarify that the dependence of collection efficiency on hydrometeor type and precipitation intensity was modeled solely based on differences in hydrometeor fall velocity.

Figure 9 and its discussion. Explain why none these curves look like the 'dry snow' curves in Figure 6. I believe it is because of the distribution of different hydrometeor sizes (and fall velocities), but it is still worth pointing out.

Ln. 507. Delete "with" in, "results with over…"

Ln. 515. Rephrase to clarify that 1.0 m s$^{-1}$ refers to the fall velocity.

Ln. 525. Delete, "considered to be."

Ln. 535. Delete, "that is."

Ln. 573 – 577. Interesting. I had no idea.

Ln. 588. The phrase, "have reduced ability to be collected" is awkward as written.

]Ln. 613, 614, 615, 619, 620, 624. I find the use of "overall" confusing. It has too many other common meanings. For example, my first read of, "Overall collection efficiencies with precipitation intensity…" on ln. 613 made me think that a comma after "overall," had been omitted. Looking back, I see that the term "overall" is nicely defined in Section 2.3, and again on ln. 370, but the use of a term that is less

commonly used in normal English would make it clearer that it has a specific meaning. Perhaps, "integrated catch efficiency?"

Ln. 624, Clarify that, "conditions when solid, liquid, or mixed precipitation can be present" refers to conditions when all of these types may be occurring, such as near-zero degrees C. As-is, 30 deg C in a thunderstorm qualifies as a time when, "solid, liquid, or mixed precipitation can be present," as does very cold conditions, when only solid precipitation can occur. I am sure there are better ways to write it, but one suggestion that remains fairly close to what is written is, "conditions when solid, liquid, *and* mixed precipitation can *all* be present." Or, "conditions when it is difficult to know the phase of the precipitation, "or near-zero degrees…"

Ln. 644 – 645. In my opinion the sentence beginning with, "The results from the ability of the hydrometeor…" can be removed. It is redundant; the previous sentence makes this point.

---

## Author Response (AR1)

**Unshielded precipitation gauge collection efficiency with wind speed and hydrometeor fall velocity.**

**Authors' response:** Thank-you to John Kochendorfer and the anonymous reviewers for providing thoughtful reviews of the original version of this manuscript and greatly improving the quality of this paper. Based on the recommendations of the editor we have merged the Part I modelling results & Part II experimental results papers and condensed the content where possible. The list of all relevant changes and point-by-point reviewer responses are included below.

**List of all relevant changes with reference to tracked changes document:**

- Pierre E. Sullivan included as coauthor from Part I

- Major revision to add relevant parts from Part I modelling method, results, and discussion to Part II manuscript

- "site-specific" changed to "climate-specific"

- "empirical transfer function" changed to "CFD transfer function"

- "Overall collection efficiency" replaced with "Integral collection efficiency"

- Wind speed $u_w$ changed to $U_w$ and fall velocity $u_f$ changed to $U_f$

- Precipitation $h$ changed to $P$

- Introduction updated with condensed Sect. 1.1 modelling studies, Sect. 1.2 transfer functions, and Sect. 1.3 Objectives

- Ln. 50 updated with reference to solid precipitation for Thériault et al. (2012) work

- Ln. 111-114 removed reference to comparison of replicate configurations of weighing gauges

- Ln. 117-119 reference to Colli et al. (2020) and Chubb et al. (2015) added

- Introduction Sect. 1.3 Objectives updated to better highlight goal of study

- Ln. 173 updated to "using a similar methodology"

- Sect. 2 Modelling method added from Part I. Fig. 1b, c and d details removed

- Ln. 217-218 added reference to size and fall velocity relationship of Rasmussen et al. (1999): "… the hydrometeor density was chosen such that the size and fall velocity followed the power law relationship of Rasmussen et al. (1999),…"

- Sect. 3 Modelling results added from Part II.

- Added references to Thériault et al., 2012;Colli et al.,Baghapour et al., 2017 for general velocity profile around gauge

- Fig. 2b velocity contour plot removed

- Ln. 336-337 revised reference to 'normalized velocities;

- Fig. 3 $u_w$ changed to $U_w$, $u*$ added to caption

- Fig. 4 $u_w$ changed to $U_w$

- Ln. 372-373 highlighted circles for rain overlap squares for ice pellets in Fig. 5

-    Ln. 376 clarified by changing "hydrometeors up to about 3 m s$^{-1}$" to "hydrometeors for horizontal wind speeds up to about

3 m s$^{-1}$"

-    Fig. 5 updated with CFD results and CFD transfer function (combined Figs. 5 and 6 from Part I)

-    Ln. 396-398 added "A single CFD curve was used for each fall velocity in the fit to ensure that the transfer function was unbiased over the entire range of fall velocities studied."

-    Ln. 413-415 added "The CFD transfer function captures well the collection efficiency trends for the different hydrometeor types, with RMSE values of 0.04 for rain, 0.02 for ice pellets, 0.02 for wet snow, and 0.05 for dry snow."

-    Fig. 6 $u_w$ changed to $U_w$

-    Sect. 3.4.1 Wind speed dependence updated to include Sect. 3.4.1 Comparison with previous studies and Sect. 3.4.2 Wind speed dependence from Part I. Section condensed.

-    Ln. 447-449 'solely' added to describe derivation of integral collection efficiency with the CFD transfer function based solely on wind speed and hydrometeor fall velocity.

-    Ln. 465-475 highlighted more gradual decrease in collection efficiency for integral collection efficiency compared with results in Fig. 5 for a given hydrometeor size.

-    Sect. 3.4.1 discussion of wind speed results condensed

-    Fig. 9 $u_w$ changed to $U_w$

-    Fig. 10 $u_w$ changed to $U_w$

-    Sect. 4.4 Transfer functions with wind speed and temperature reference to CFD transfer function removed. This information is included in Sect. 3.3

-    Ln. 669-671 added description for estimation of Table 7 fall velocity and temperature ranges shown

-    Fig. 12 $u_w$ changed to $U_w$

-    Removed Part II Fig. 3 and results description

-    Ln. 744-745 added description of how fall velocity threshold was determined. "The fall velocity threshold was varied over the measurement fall velocity range in 0.01 m s$^{-1}$ increments, with the threshold of 1.93 m s$^{-1}$ found to provide the lowest overall RMSE."

-    Eqs. 25a and b updated with collection efficiency of 0.2 above 5.75 m s$^{-1}$ wind speed

-    Ln. 753 clarified that HE2 decreases linearly with wind speed for a given hydrometeor fall velocity

-    Ln. 754-756 added description of how fall velocity threshold was determined. "The fall velocity threshold was varied over the measurement fall velocity range in 0.01 m s$^{-1}$ increments, with the threshold of 2.81 m s$^{-1}$ found to provide the lowest overall RMSE."

-    Eqs. 26a and b updated with collection efficiency of 0.2 above wind speed threshold which varies with fall velocity

-    Fig. 13 caption $u_f$ changed to $U_f$ and Figs. 13c and d updated with 0.2 collection efficiency threshold for HE1 and HE2

transfer functions

-    Table 9 combines results from Part II Tables 4, 5 and 6

- Table 10 combines results from Part II Tables 7, 8, and 9

- Sect. 6.1 Modelling discussion added from Part I

- Ln. 956-960 model limitations added, "The numerical results for this study are based on a 5 % inlet turbulence value that acts as a bulk turbulence in the atmosphere (Panofsky and Dutton, 1984) buy may underestimate experimental results (Armitt and Counihan, 1968). A no-slip boundary condition was modelled at the surface following the approach of previous studies (Baghapour and Sullivan, 2017;Colli et al., 2016). Further study with a no-slip wall condition under different turbulence conditions could lead to further insights into the influence of turbulence intensity on precipitation gauge collection efficiency."

- Sect. 6.1.2 condensed

- Ln. 1000-1004 added description of limitations of dry snow spherical hydrometeor model

- Ln. 1038-1044 turbulence discussion and future work added, "These results are based on time-averaged simulations, which provide an estimate of the mean velocities through the domain and have been shown to provide good overall agreement with experimental results (Baghapour et al., 2017). Further study using LES models, which can better resolve the eddy dynamics and temporal variations in the flow, and under different boundary conditions and turbulence scales representing different site conditions is recommended to better understand the collection efficiency under conditions with high wind speeds and low hydrometeor fall velocities."

- Sect. 6.1.4.1 condensed

- Ln. 1108 reference to Thériault et al. (2012) removed

- Ln. 1135-1136 added reference to Colli et al. (2020) for collection efficiency dependence on precipitation intensity

- Ln. 1139-1140 updated with and and all to clarify meaning "The range of possible integral collection efficiency values is even larger under conditions when solid, liquid, and mixed precipitation can all be present."

- Sect. 6.1.4.3 condensed

- Sect. 6.2 updated reference to HE1 and HE2 collection efficiency of 0.2 above wind speed threshold

- Ln. 1208-1209 added, "For larger shields, it may be important to employ a more realistic vertical wind profile, with a zero- slip boundary condition at the earth's surface."

- Ln. 1220-1221 added, "Further testing at other sites is recommended to better understand the collection efficiency for low fall velocity hydrometeors (light snow) under windy conditions above 6 m s$^{-1}$, which were not available in the CARE

dataset."

- Ln. 1235-1238 added, "The present approach of estimating the fall velocity using the POSS appears to perform well, overall; however, further study to better characterize the fall velocity distribution and changes over 30-minute time periods could lead to further improvements in the model under specific conditions such as mixed precipitation."

- Sect. 6.3 heading "Application to operational networks" added

- Ln. 1247-1249 moved (Kienzle, 2008;Harder and Pomeroy, 2013) reference after "instructive" for clarity

-   Ln. 1258-1259 removed, "At high wind speeds, the unshielded gauge catch may be insufficient for adjustment due to the
low measured quantities."
-   Sect. 7 Conclusions updated to include modelling and experimental results

**Unshielded precipitation gauge collection efficiency with wind speed and hydrometeor fall velocity. Part I: modelling results**

**Author Response to Anonymous (Referee #1)**

**Authors' response:** We respect the reviewer's perspective and the candid nature of their responses, the central theme of which is the perceived unoriginality of the method used in this study the results presented. In the responses that follow, we will articulate how the present work builds upon previous studies in new ways, with novel and impactful results.

This study proposes that fall speed influences the collection efficiency of unshielded gauge using computation fluid dynamics (CFD). The authors claim that they are using a new method to study gauge collection efficiency and, with this method, that they are the first to demonstrate the impact of fall speed on the gauge collection efficiency.

**Authors' response:** This is the first CFD study to develop a universal collection efficiency transfer function based on wind speed and hydrometeor fall velocity, which is broadly applicable, novel, and important. The hydrometeor fall velocity can be measured by a variety of instruments for both rain and snow and the use of this transfer function can dramatically improve experimental collection efficiency estimates as shown in Part II. Previous studies have certainly used CFD to study gauge collection efficiency for shielded and unshielded gauges, and have attributed differences in results to differences in hydrometeor characteristics, including fall velocities. These studies are described and referenced clearly in the manuscript. This manuscript presents a new method in that it is the first CFD modelling study to: (1) characterize precipitation gauge collection efficiency explicitly in terms of wind speed and fall velocity; (2) show that collection efficiencies are similar for different hydrometeor types with identical fall velocities; and (3) develop a universal transfer function based on wind speed and hydrometeor fall velocity. We will revise the manuscript to better highlight the innovations.

In fact, these have already been done with a similar approach:
1) Thériault et al. (2012), Colli et al. (2016a,b) used CFD to study gauge collection efficiency for snow.

**Authors' response:** Neither of these studies develops a collection efficiency transfer function with wind speed and hydrometeor fall velocity.

Thériault et al. (2012) developed transfer functions with wind speed for specific hydrometeor types (radiating assemblage of plates, dendrite, heavily rimed dendrites, hexagonal plates, lump graupel, dry snow, and wet snow), with a different transfer function for each hydrometeor type. The contribution of Thériault et al. (2012) is captured in the current manuscript in a number of places (ln. 69-79, 91-93, 469-471, 492-495, 501-503, 562-564, 568-570, and 596-597). Colli et al. (2016a,b) did not develop a collection efficiency transfer function.

There are also other studies that have used CFD to study gauge collection efficiency for rain and snow, including the work of

Nešpor and Sevruk (1999), Colli et al. (2015), Baghapour et al. (2017), and Baghapour and Sullivan (2017), as described in the manuscript. These studies also did not develop a collection efficiency transfer function with wind speed and hydrometeor fall velocity.

Further, the details of the modelling approach used in the present study differ from those used in the previous studies identified by the reviewer. These differences are discussed in the manuscript (Sect. 4.1, 4.2, and 4.4.1) and are summarized in Table A1, below.

**Table A1:** Summary of Thériault et al. (2012), Colli et al. (2016a,b), and present study numerical collection efficiency models.

|  | Thériault et al. (2012) | Colli et al. (2016a,b) | Present study |
|---|---|---|---|
| Numerical model | Reynolds-averaged Navier-Stokes (RANS) k-ε | Reynolds-averaged Navier-Stokes (RANS) k-ω, Large-eddy simulation (LES) | Favre-averaged Navier-Stokes (FANS) k-ε |
| Gauge | Geonor T-200B | Geonor T-200B | Geonor T-200B |
| Gauge geometry (orifice) | Not specified ~1 cm orifice thickness | Not specified ~1 cm orifice thickness | Refined orifice thickness (3.15mm) and length (360 mm) to match actual gauge |
| Shield | Single-Alter | Unshielded, Single-Alter | Unshielded |
| Mesh | 0.35 M cells | Tetrahedra and prisms, 1.5 M – 29.5 M cells | Structured, 8.3 M cells |
| Inlet turbulence intensity | Not specified | 0 % | 5 % |
| Precipitation type | Dry snow, wet snow, radiating assemblage of plates, hexagonal plates, dendrite, graupel, and heavily rimed dendrite | Dry snow, wet snow | Orographic rain, thunderstorm rain, dry snow, wet snow, ice pellet, snow, dendrites, rimed dendrites, columns and plates, dendrites and aggregates of plates |
| Hydrometeor model | Lagrangian uncoupled | Lagrangian uncoupled | Lagrangian uncoupled |

| | | | |
|---|---|---|---|
| Drag coefficient | Constant over hydrometeor trajectory | Constant over hydrometeor trajectory | Drag varies with relative hydrometeor to air velocity over trajectory |
| Injection plane | Not specified | Vertical | Horizontal |
| Model parameters studied | Wind speed, precip type, hydrometeor size distribution, turbulence | Numerical model, wind speed, precip type, hydrometeor size, shielding | Wind speed, fall velocity, precip type, precip intensity |
| Collection efficiency definition | $CE = \dfrac{\int_0^{D_{max}} A_{inside}(D)N(D)}{\int_0^{D_{max}} A_{gauge}(D)N(D)}$ | $CE = \dfrac{\int_0^{d_{P\max}} V_w(d_p) A_{inside}(d_p, U_w) N(d_p)d_p}{\int_0^{d_{P\max}} V_w(d_p) A_{gauge} N(d_p)d_p}$ | $CE_{R,Overall} = \dfrac{\int_0^{\infty} CE(u_w, u_f) D^3 N_R(D) u_f(D)\,dD}{\int_0^{\infty} D^3 N_R(D) u_f(D)\,dD}$ $CE_{S,Overall} = \dfrac{\int_0^{\infty} CE(u_w, u_f) D^3 N_S(D) u_f(D)\,dD}{\int_0^{\infty} D^3 N_S(D) u_f(D)\,dD}$ |
| Derived transfer function | $CE_{snow}$ = f(wind speed) with unique transfer function for specific solid precip types | None | $CE_{rain\&snow}$ = f(wind speed, hydrometeor fall velocity) universal transfer function across rain and snow precip types Derived fall velocity cutoff for zero collection efficiency |

2) Colli et al. (2016a) were the first to compute the flow field near an unshielded gauge as performed in this manuscript.

**Authors' response:** The Colli et al. (2016a) study is clearly referenced in the Part I manuscript (ln. 96-102). There is no claim that the present study is the first to compute the flow field near an unshielded gauge. Results using the approach from Colli et al. (2016a) are compared with those from the model used in the present study (see Figure 8 and discussion in Sect. 3.4.1), with differences in results between the studies attributed to differences in the models and approaches used.

3) Thériault et al. (2012) found a strong dependence between the gauge collection efficiency and fall speed. Indeed, it was conducted with a shielded gauge but the physical reasons are the same. The updraft upstream of the gauge tends to deviate the slow-falling particles to fall in the gauge. For the same horizontal wind speed, slow-falling snowflakes have lower collection efficiency than faster-falling ones.

**Authors' response:** The authors recognize that the theoretical basis for how hydrometeor fall velocity can influence collection
efficiency has been established using CFD simulations in Thériault et al. (2012) and other studies, and have discussed and
referenced these studies in the present manuscript.

As noted above, the work of Thériault et al. (2012) is captured in the current manuscript in a number of places (ln. 69-79, 91-
93, 469-471, 492-495, 501-503, 562-564, 568-570, and 596-597). This work concluded that "snowflakes fall at different
terminal velocities and therefore interact differently with the deflected flow around the snow gauge," and discussed the
importance of hydrometeor terminal velocity on the collection efficiency results for different crystal types. The results from
the present study reinforce those findings, and build upon them by considering hydrometeor fall velocity more globally, and
not within the limitations of prescribed snowflake/ice crystal types (dry snow and wet snow in Thériault et al., 2012).

The present study shows that collection efficiencies are similar for different hydrometeors with the same fall velocity, despite
differences in size, density, mass etc. using a spherical drag model. It is not apparent from the work of Thériault et al. (2012)
that a raindrop with the same fall velocity as a spherical wet snow or dry snow hydrometeor would have a similar collection
efficiency, despite the large differences in the density. This is an important finding, as it enables the collection efficiency to be
well characterized by the wind speed and fall velocity alone, and enables the development of an explicit CFD transfer function
with wind speed and hydrometeor fall velocity dependence, which was not done in Thériault et al. (2012).

4) Colli et al. (2020) used the precipitation intensity as done in this manuscript to adjust the collection efficiency.

Colli, M., Stagnaro, M., Lanza, L. G., Rasmussen, R. and Thériault, J. M. (2020). Adjustments for wind-induced undercatch
in snowfall measurements based on precipitation intensity, Journal of hydrometeorology, 21, 1039-1050.

The impact of precipitation intensity on the collection efficiency was also suggested by Chubb et al. (2015) using field
measurements.

Chubb, T., Manton, M. J., Siems, S. T., Peace, A. D., & Bilish, S. P. (2015). Estimation of Wind-Induced Losses from a
Precipitation Gauge Network in the Australian Snowy Mountains, Journal of Hydrometeorology, 16(6), 2619-2638.

**Authors' response:** The transfer function developed in the present study uses the wind speed and hydrometeor fall velocity
to adjust the collection efficiency. This is fundamentally different from adjustments based on the wind speed and precipitation
intensity (determined from the measured gauge accumulation) used by Colli et al. (2020) and Chubb et al. (2015). The transfer
function developed in the present study can be used to estimate collection efficiencies for different hydrometeor types and
intensities, representing the hydrometeor properties in terms of the corresponding fall velocity. Results using this approach are
shown in Fig. 10 and discussed in Sections 3.4.3 and 4.4.3 of the present manuscript. It is important to note that both of the
studies identified by the reviewer develop explicit transfer functions with wind speed and precipitation intensity based on
experimental results and not directly based on modeling results as in the present study. In the case of Colli et al. (2020),
different fit coefficients are determined for each of the Marshall, CARE, and Haukeliseter field test sites with collection
efficiency results studied at temperatures below -4 °C.

Using the precipitation intensity approach enables adjustments to be performed at sites where only precipitation gauge and
wind speed measurements are available. However, the degree of the gauge adjustment (obtained from the precipitation intensity
and gauge measurement) is not independent from the measured value to be adjusted. This could be problematic for adjusted
precipitation accumulation estimates, as gauge measurement uncertainties can be propagated through both the measured gauge
accumulation and the collection efficiency transfer function. It is also difficult to apply this approach across different
hydrometeor types (e.g. rain and snow), as different types can have different fall velocities and associated collection
efficiencies, even for the same precipitation intensity. As shown in Fig. 10, the range of collection efficiencies that can be
obtained across rain and snow increases with increasing wind speed for a given precipitation intensity. This makes it difficult
to apply an intensity-based approach over temperature ranges where liquid and/or solid precipitation types can be present. The
fall velocity transfer function developed in the present study can be applied more broadly across different precipitation types,
and the collection efficiency adjustment is determined independently from the gauge accumulation, with separate instruments
for measuring the wind speed, precipitation fall velocity (e.g. a Precipitation Occurrence Sensor System (POSS), as shown in
Part II), and precipitation accumulation.

In particular: Section 1: The introduction is very long and the goal is not stated clearly. The literature review is incomplete.
What are the authors trying to do exactly? If it is showing that CFD can be used to show the dependence of the collection
efficiency on the fall speed, it has already been done before.
**Authors' response:** The goal of this work is to develop a computationally cost-effective, universally applicable, and
quantitative transfer function for adjusting unshielded precipitation gauge measurements with wind speed and hydrometeor
fall velocity. We agree that this can be stated more clearly, and will do so in the revised version of the Part I manuscript.
The introduction describes previous studies that have established the practical and theoretical basis for the present study. The
contributions of Thériault et al. (2012) and Colli et al. (2016a, b) are described in the introduction, among other studies. The
introduction length was required to ensure that previous studies related to the present work were clearly described. It is
important to note that none of the previous studies above develop a CFD transfer function based on wind speed and
hydrometeor fall velocity as is done in the present study. If the reviewer is suggesting that the literature review is incomplete
because the studies by Colli et al. (2020) and Chubb et al. (2015) are not discussed, inclusion of these references could be
considered for the revised version of Part I. Recognizing both the length of the introduction (as noted by the reviewer), and the
fact that these studies are fundamentally different than the present study (as noted in the previous response), these studies were
not included in the original submission.

Section 2: The simulations described in section 2.1 were already done in Colli et al. (2016a).

**Authors' response:** The work of Colli et al. (2016a) is described in the introduction (ln. 91-102) and the numerical modelling results from the present study are compared with those from both Colli et al. (2016a) and Baghapour et al. (2017) in Sect. 3.1

and shown in Fig. 3. The reductions in the peak normalized velocity above the gauge with the present model compared with

Colli et al. (2016a) are attributed to refinement of the gauge geometry, including the orifice thickness, among other factors as discussed in Sect. 4.1. The comparison of results from different models provides a useful benchmark as models are refined and improved over time, and allows the impacts of model changes to be assessed.

The collection efficiency computed in section 2.3 were first used in Colli et al. (2020).

**Authors' response:** The authors disagree that the collection efficiency formulation in the present study was first shown by

Colli et al. (2020). The Sect. 2.3 methodology for calculating the true precipitation intensity falling in air from the hydrometeor size distribution, mass, and fall velocity over hydrometeor sizes in this study follows the approach of Nešpor and Sevruk (1999), as described in ln. 236-237. This is a common definition for precipitation intensity, and the same approach is used for rain and snow in this manuscript, where the size is based on the equivalent water drop diameter. Sect. 2.3 describes the methodology for deriving the overall collection efficiency with wind speed for a given hydrometeor size distribution for both rain and snow, using the collection efficiency transfer function with wind speed and hydrometeor fall velocity developed in

Sect. 3.3.

Sections 3 and 4: Most results/discussion are not new and/or should be improved for clarity. For example:

1) Sections 4.1, 4.2: Same key findings as in previous studies.

**Authors' response:** The results from this study are compared with those from previous studies in Sections 4.1 and 4.2. The results in the present study were determined independently from those in previous studies, with differences in the specific approaches used. Identifying similarities among the results from the present and previous studies is valuable to the scientific community. Accordingly, the fact that some key findings were the same as in previous studies serves to reinforce and support those findings. That said, there are also differences in the key findings of the present study relative to previous studies, which adds new knowledge to the field.

Sect. 4.1 discusses the differences in the numerical modelling results presented in Sect. 3.1 relative to previous studies; specifically, Colli et al. (2016a) and Baghapour et al. (2017). The reductions in the peak normalized velocity above the gauge with the present model compared with Colli et al. (2016a) are attributed to refinement of the gauge geometry, including the orifice thickness, among other factors as discussed in Sect. 4.1. Sect. 4.2 discusses the collection efficiency results with wind speed and hydrometeor fall velocity for rain, ice pellet, wet snow, and dry snow hydrometeors. Fall velocity was not considered explicitly in the modelling approaches used in previous studies. The numerical results show collection efficiency results are similar for hydrometeors with the same fall velocity, despite differences in characteristics (size, density, and mass). This is a new and important finding that has not been demonstrated in previous studies. This finding supports the development of a universal transfer function with wind speed and hydrometeor fall velocity that is applicable across liquid and solid hydrometeor types.

2) Section 4.3: The threshold fall speed value is directly related to the minimum diameter of the size distribution discussed in Thériault et al. (2012) and Colli et al. (2016a, b) and Colli et al. (2020). Small particles falling slower are deflected by the updraft upstream of the gauge.

**Authors' response:** As described in the introduction (ln. 62-63), Nešpor and Sevruk (1999) demonstrated a hydrometeor size limit below which the collection efficiency was zero for smaller size hydrometeors for rain. This defines, not the minimum diameter of the drop size distribution, but the minimum size of hydrometeor with sufficient fall velocity to be captured by the gauge for a given wind speed. This threshold will change with wind speed, as for higher wind speeds, larger drop sizes with higher fall velocities are required to overcome the updraft and local airflow to be captured by the gauge. Colli et al. (2016b) shows that the hydrometeor size at which the collection efficiency is zero increases for 8 m s$^{-1}$ wind speed relative to 4 m s$^{-1}$ wind speeds for dry snow with an unshielded gauge, but does not develop an explicit formula relating wind speed to the minimum drop diameter, which will also be different for different hydrometeor types. The present study develops an explicit expression for this threshold based on the wind speed and hydrometeor fall velocity, based on the numerical model results, which is broadly applicable across hydrometeor types (Eq. 19). None of the three publications listed above derive an explicit expression for the hydrometeor fall velocity threshold (below which the collection efficiency will be zero) based on the wind speed and that is applicable across different hydrometeor types.

3) Section 4: Lines 565-569: It should be corrected as previous studies by Thériault et al. and Colli et al. also used a horizontal plan.

**Authors' response:** The Colli et al. (2016b) publication states the hydrometeors were injected from a vertical plane as described in ln. 568-569 in the present manuscript. The following description is provided in Colli et al. (2016b), "The initial positions of the simulated trajectories lay on an ideal vertical plane located upwind of the windshield and the orifice level. Figure 1 shows the selected seeding window and its location relative to the shield–gauge assembly." The Thériault et al. (2012) publication does not state whether hydrometeors were injected from a horizontal or vertical plane for the modelling analysis and the authors recommend that the reference to Thériault et al. (2012) in ln. 568 is removed.

Lines 573-577: The volumetric approach is what the gauge measures. When using the fall speed, it is the precipitation intensity as proposed in Colli et al. (2020).

**Authors' response:** The Colli et al. (2020) reference is not relevant to this discussion. This section is referring to the
comparison of the Colli et al. (2016) unshielded model results for wet snow and dry snow with those of the present study. Ln.
573-577 in the present manuscript refer to the approach of Colli et al. (2016b), which calculates the ratio of that captured inside
the gauge to the true value falling in air. The fall velocity term is omitted in Eq. (12) in Colli et al. (2016b) in both the numerator
and denominator integrals, which differs from the formulation used by Nešpor and Sevruk (1999) and that used in the present
study. As stated in the manuscript, for the dry snow and wet snow comparison shown in Fig. 8, the difference between these
two approaches is small.

Sections 4.4.2, 4.4.3 and 4.4.4: Most of the content are not new findings and are repetitive.

**Authors' response:** New findings are presented in Sections 4.2.2, 4.4.3, and 4.4.4, as well as relevant comparisons with
previous studies. These sections discuss and contextualize the results from the universal transfer function with wind speed and
hydrometeor fall velocity dependence that are presented in Sections 3.4.2, 3.4.3, and 3.4.4, respectively. Results from the new
transfer function developed herein are studied over a range of hydrometeor types (both liquid and solid), precipitation
intensities, and wind speeds. These sections highlight and discuss the variability in collection efficiency results due to these
different factors for both snow and rain. This provides valuable context to help understand the limitations associated with
performing adjustments based on the wind speed alone (Sect. 4.4.2), with wind speed and precipitation intensity (Sect. 4.4.3),
and with wind speed and hydrometeor fall velocity (Sect. 4.4.4) across different hydrometeor types.
The relationship between collection efficiency and wind speed is discussed in Section 4.4.2 with respect to the approach of
Thériault et al. (2012) for snowfall. Section 4.4.3 discusses the relationship between collection efficiency and precipitation
intensity, comparing with findings from Jarraud (2008), and illustrating the new and important point that an intensity-based
approach can lead to a range of collection efficiency values when multiple snowfall crystal habits are present or when both
solid and liquid precipitation are present. Section 4.4.4 discusses how the spread in collection efficiency results across different
precipitation types at a given wind speed is minimized by representing the hydrometeor properties in terms of fall velocity.
This is a novel and significant finding, demonstrating how the new transfer function can be applied broadly across all
hydrometeor types with no knowledge of their properties other than the fall velocity.

Given those, there is not enough novelty in this manuscript to be published. Since some of the results are needed for Part 2, I
recommend merging both manuscripts. A methodology section that explains the CFD simulations should be added to Part 2.

**Authors' response:** The authors strongly disagree with the reviewer's claims that the work is not novel and will revise the
manuscript to better highlight these points. To the authors' knowledge, this is the first CFD modelling study to: (1) characterize
precipitation gauge collection efficiency with wind speed and fall velocity; (2) show that collection efficiencies are similar for
different hydrometeor types with identical fall velocities; and (3) develop a universal transfer function based on wind speed
and hydrometeor fall velocity. This work demonstrates the significant utility of such a transfer function for reducing
uncertainties in adjusted precipitation accumulation estimates, and provides a foundation for future studies with non-spherical
hydrometeor models and different gauge and shielding configurations used operationally.

The authors strongly recommend that the modelling results are maintained as a standalone paper to enable the modelling
results, transfer function development, comparison with previous modelling studies, and discussion of the results to be clearly
and fully described. As previous studies have shown, and as discussed in this study, the numerical modelling results are
sensitive to a wide range of factors (e.g. gauge and orifice geometry, mesh, boundary conditions, turbulence model,
hydrometeor drag model, hydrometeor type and characteristics…), and it is important that they are discussed in the context of
the model and transfer function development.

The Part II manuscript uses the CFD transfer function and assesses it experimentally alongside existing transfer functions with
wind speed and temperature dependence, as well as two new transfer functions with wind speed and fall velocity dependence.
For Part II, the goal is not the justification of the approach from a modelling and fundamental perspective, but the experimental
evaluation of transfer functions with wind speed and hydrometeor fall velocity alongside existing approaches with wind speed
and temperature. Including the CFD model methodology, results, discussion and conclusions from Part I in the Part II
manuscript would detract from the clarity of the Part II paper, as duplicate methodology, results, discussion and conclusion
aspects would be required from the numerical modelling work. While the numerical modelling, analysis and transfer function
development in Part I is fundamental to the transfer functions and results in Part II, this work is best-suited in its present form
as a standalone paper.

**Unshielded precipitation gauge collection efficiency with wind speed and hydrometeor fall velocity. Part I: modelling results**

**Author Response to Anonymous (Referee #2)**

In this work the authors presented "A new method for assessing collection efficiency using wind speed and hydrometeor fall velocity", but this methodology, based on CFD simulations and Lagrangian particle tracking model have been previously used in the recent literature (e.g. Thériault et al. 2012, Colli et al. 2016a,b). The Geonor precipitation gauge has been studied in these works in both shielded and unshielded configuration.

**Authors' response:** This is the first CFD study to develop a universal collection efficiency transfer function based on wind speed and hydrometeor fall velocity, which is broadly applicable, novel, and important. Previous studies, including those of Thériault et al. (2012) and Colli et al. (2016a,b), have certainly used CFD simulations and Lagrangian particle tracking to study collection efficiency for Geonor gauges in shielded and unshielded configurations. These studies, as well as others that used CFD to study gauge collection efficiency for rain and snow (e.g. Nešpor and Sevruk, 1999; Colli et al., 2015; Baghapour et al., 2017; and Baghapour and Sullivan, 2017), are described and referenced clearly in the manuscript. None of these studies developed a collection efficiency transfer function based on the wind speed and hydrometeor fall velocity, as in the present study; hence, this is a new method. The hydrometeor fall velocity can be measured by a variety of instruments for both rain and snow and the use of this transfer function can dramatically improve experimental collection efficiency estimates, as shown in Part II.

Thériault et al. (2012) developed transfer functions with wind speed for specific hydrometeor types (radiating assemblage of plates, dendrite, heavily rimed dendrites, hexagonal plates, lump graupel, dry snow, and wet snow), with a different transfer function for each hydrometeor type. The contribution of Thériault et al. (2012) is captured in the current manuscript in a number of places (ln. 69-79, 91-93, 469-471, 492-495, 501-503, 562-564, 568-570, and 596-597). The contributions of Colli et al. (2016a,b) do not develop a collection efficiency transfer function. There are also modelling differences between the present study and each of these two earlier studies, as shown in Table A1 (below) and discussed in the manuscript.

**Table A1:** Summary of Thériault et al. (2012), Colli et al. (2016a,b), and present study numerical collection efficiency models.

| | Thériault et al. (2012) | Colli et al. (2016a,b) | Present study |
|---|---|---|---|
| Numerical model | Reynolds-averaged Navier-Stokes (RANS) k-ε | Reynolds-averaged Navier-Stokes (RANS) k-ω, Large-eddy simulation (LES) | Favre-averaged Navier-Stokes (FANS) k-e |

| Gauge | Single-Alter shielded Geonor T-200B | Unshielded and single-Alter shielded Geonor T-200B | Unshielded Geonor T-200B |
|---|---|---|---|
| Gauge geometry (orifice) | Not specified ~1 cm orifice thickness | Not specified ~1 cm orifice thickness | Refined orifice thickness (3.15mm) and length (360 mm) to match actual gauge |
| Shield | Single-Alter | Unshielded, Single-Alter | Unshielded |
| Mesh | 0.35 M cells | Tetrahedra and prisms, 1.5 M – 29.5 M cells | Structured, 8.3 M cells |
| Inlet turbulence intensity | Not specified | 0 % | 5 % |
| Precipitation type | Dry snow, wet snow, radiating assemblage of plates, hexagonal plates, dendrite, graupel, and heavily rimed dendrite | Dry snow, wet snow | Orographic rain, thunderstorm rain, dry snow, wet snow, ice pellet, snow, dendrites, rimed dendrites, columns and plates, dendrites and aggregates of plates |
| Hydrometeor model | Lagrangian uncoupled | Lagrangian uncoupled | Lagrangian uncoupled |
| Drag coefficient | Constant over hydrometeor trajectory | Constant over hydrometeor trajectory | Drag varies with relative hydrometeor to air velocity over trajectory |
| Injection plane | Not specified | Vertical | Horizontal |
| Model parameters studied | Wind speed, precip type, hydrometeor size distribution, turbulence | Numerical model, wind speed, precip type, hydrometeor size, shielding | Wind speed, fall velocity, precip type, precip intensity |
| Collection efficiency definition | $CE = \dfrac{\int_0^{D_{\max}} A_{inside}(D)N(D)}{\int_0^{D_{\max}} A_{gauge}(D)N(D)}$ | $CE = \dfrac{\int_0^{d_{P\max}} V_w(d_p)A_{inside}(d_p,U_w)N(d_p)d_p}{\int_0^{d_{P\max}} V_w(d_p)A_{gauge}N(d_p)d_p}$ | $CE_{\text{R,Overall}} = \dfrac{\int_0^{\infty} CE(u_w,u_f)D^3 N_R(D)u_f(D)\,\mathrm{d}D}{\int_0^{\infty} D^3 N_R(D)u_f(D)\,\mathrm{d}D}$ $CE_{\text{S,Overall}} = \dfrac{\int_0^{\infty} CE(u_w,u_f)D^3 N_S(D)u_f(D)\,\mathrm{d}D}{\int_0^{\infty} D^3 N_S(D)u_f(D)\,\mathrm{d}D}$ |

| Derived transfer function | $CE_{snow}$ = f(wind speed) with unique transfer function for specific solid precip types | None | $CE_{rain\&snow}$ = f(wind speed, hydrometeor fall velocity) universal transfer function across rain and snow precip types
Derived fall velocity cutoff for zero collection efficiency |
| --- | --- | --- | --- |

One of the main conclusion of this work is the relation between "Collection Efficiency" (CE) and the particle fall velocity instead of the particle diameter as shown in Colli et al. 2016b. However, for wet and dry snow they use the relation proposed by Rasmussen et al. 1999 to calculate the particle fall velocity as a function of the particle diameter. Furthermore, in equation 9 and 17 the authors reported the formulas for the "overall Collection Efficiency" for rain and snow respectively. In these equations is highlighted that the fall velocity is a function of the particle diameter (D) and therefore the overall CE depends only on the wind speed and D. For this reason, there is no novelty in this approach.

**Authors' response:** The statement by the reviewer that "the fall velocity is a function of the particle diameter (D) and therefore the overall CE depends only on the wind speed and D" would certainly apply to situations where the hydrometeor type and fall velocity dependence with size is known, only one hydrometeor type is present, and the size distribution of hydrometeors is known. This is essentially what has been shown in the previous studies by Thériault et al. (2012) and Colli et al. (2016b). This is **not** what is shown in the present study. Further, neither of these previous studies considered fall velocity explicitly in the collection efficiency formulation, as it is in the present study.

The present study shows that a single transfer function based on wind speed and hydrometeor fall velocity, developed herein, can accurately capture collection efficiencies across a wide range of wind speeds without explicit knowledge of the hydrometeor type, size distribution, or intensity, and in situations where multiple hydrometeor types are present. By using the fall velocity, which is a singular, observable parameter, it is possible to describe the collection efficiency without any further knowledge of the hydrometeors. It is not apparent from the work of Colli et al. 2016b that the collection efficiency for different hydrometeors with the same fall velocity (rain, wet snow, and dry snow) would be similar despite large differences in the hydrometeor diameter, density and mass. For example, using the present approach, a small raindrop with a fall velocity of 0.5 m s$^{-1}$ is assigned the same collection efficiency as a spherical dry snow hydrometeor with the same fall velocity. There is significant novelty in the approach developed in this study, as adjustments based on fall velocity are more broadly applicable than those developed in previous studies that require knowledge of the hydrometeor size and type.

With respect to the specific formulations used, Eqs. 9 and 17 derive the overall collection efficiency by integrating over the hydrometeor size distribution. The sizes here correspond to the equivalent diameters of water droplets as described in ln. 245-

246, which differ from the values of Rasmussen et al. (1999) based on the hydrometeor size. For snowfall, the power law values in this study are given by Langleben (1954) as described in ln. 206-264 and shown in Table 4. Substituting the fall velocity expression with equivalent drop diameter (Eq. 16) into the overall collection efficiency expression (Eq. 17) would indeed provide different collection efficiency curves for different hydrometeor types, as the relationship between the hydrometeor size and fall velocity is different. These differences are shown in Fig. 9 for different liquid and solid precipitation types and intensities.

**L 175**: a) The authors use the relation proposed by Rasmussen et al. 1999 to calculate the terminal velocity, and they stated that "hydrometeor density was chosen to provide the desired hydrometeor fall velocity", but in the work of Rasmussen et al. the density value relations are provided for both wet and dry snow. How did the authors vary the hydrometeor density?

**Authors' response:** The hydrometeor density was not varied in this study. As described in Sect. 2.2, the hydrometeor density for wet snow and dry snow was determined from the hydrometeor diameter and fall velocity using a spherical drag model (ln. 175-178). The size and fall velocity relationship for spherical wet snow and dry snow hydrometeors follows that of Rasmussen et al. (1999), which was used in previous studies (Thériault et al., 2012;Colli et al. 2016b). The drag coefficient for spherical hydrometeors is given by Henderson (1976) based on the relative hydrometeor to air velocity. This drag formulation closely matches that of Haider and Levenspiel (1989) used in previous studies (Baghapour and Sullivan, 2017). Fig. 5 shows collection efficiencies are similar for hydrometeors with the same fall velocity despite differences in type, size, density, and mass (Table 2). This enables the development of a collection efficiency transfer function based on wind speed and hydrometeor fall velocity, independent of hydrometeor type, size, density, and mass.

**L 175**: b) Are these density values realistic? or are they used only to obtain the fall velocity the authors desired?

**Authors' response:** The density values provided in Table 2 are realistic for spherical hydrometeors with the diameter and fall velocity relationship provided by Rasmussen et al. (1999) and used in previous studies (Thériault et al., 2012; Colli et al. 2016b). The results of this study show that across rain, ice pellet, wet snow, and dry snow hydrometeors, collection efficiency results are highly sensitive to the wind speed and hydrometeor fall velocity, and relatively insensitive to differences in hydrometeor density across hydrometeor types.

**L 175**: c) The smaller particle of wet snow has a density value greater than water, is it right?

**Authors' response:** For wet snow, the density increases rapidly with decreasing size below approximately 3 mm, as shown in Table 2 in the manuscript. Comparing between the 1.0 m s$^{-1}$ fall velocity hydrometeor (with 0.22 mm diameter and 1.35 kg m$^{-3}$ density) and 1.25 m s$^{-1}$ fall velocity hydrometeor (with 1.7 mm diameter and 0.3 kg m$^{-3}$ density) in Table 2 shows the rapid decrease in the diameter and increase in the density as the fall velocity is reduced. While a density above 1 kg m$^{-3}$ is unrealistic, it was included to show the results at the edge of this low fall velocity range, despite slightly over-shooting the density of water. It is worth noting that higher densities for wet snow (2.88 kg m$^{-3}$ and 1.44 kg m$^{-3}$) were included in the modelling analysis of Colli et al. (2016b) for 0.25 mm and 0.5 mm diameter hydrometeors. This overestimation of the density at small hydrometeor sizes may be due to errors in the power law relationship between hydrometeor diameter and fall velocity at small hydrometeor diameters, which is beyond the scope of the present study to assess further.

**Fig. 5 and 6** : in figure 5 the authors showed the "collection efficiency" for different precipitation types and fall velocities respect to wind speed.

**Authors' response:** We agree with this statement.

It is clear from the figure that there are differences in the CE values of different precipitation type but with the same fall velocity.

**Authors' response:** We agree there are differences, and these are already discussed in Sect. 3.2 in the paper.

Furthermore, the authors used only part of these data to obtain the "empirical collection efficiency expression" showed in figure 6, but this relation has been used to calculate the "overall Collection Efficiency" for all the particle types. How do this affect the obtained results?

**Authors' response:** Selected curves were chosen representative of each hydrometeor type to span the range of possibilities in a uniform and unbiased way. This does not significantly affect the obtained results. This is described in Sect. 3.3.

We will modify the discussion in Sect 3.3 to further highlight this point, including adding the RMSE results for rain (0.04), ice pellets (0.02), wet snow (0.02), and dry snow (0.05) compared with the collection efficiency transfer function to further describe the agreement of the transfer function with the CFD results. We also recommend including all the CFD results from

Fig. 5 with the transfer function fits in Fig. 6 to better show the performance of the transfer function relative to the entire CFD

modelling dataset.

Results from this transfer function using the "overall collection efficiency" are compared with Colli et al. (2016b) dry snow and wet snow results and show good overall agreement (Fig. 8) despite modelling differences as discussed in Sect. 4.4.1. The transfer function is also directly assessed with experimental results for rain, snow and mixed precipitation over a wide range of environmental conditions in the Part II manuscript, showing good agreement (Tables 4 and 7).

**Sections 3.4.3 and 4.43**: in these sections (Results and Discussion sections) the authors highlight the dependency of overall CE with precipitation intensity. This topic is addressed in the recent work of Colli et al. 2020. Do the authors compare their results with that work?

**Authors' response:** The authors recommend that the work of Colli et al. (2020) is added to the introduction and referenced in the discussion (Sect. 4.4.3) with respect to the dependence of overall collection efficiency on precipitation intensity for a given hydrometeor type. Their findings support the results of the present study using the transfer function based on wind speed and hydrometeor fall velocity developed herein. It is important to note that the work of Colli et al. (2020) is for a single-Alter shielded Geonor gauge, and is not directly comparable to this work using an unshielded Geonor gauge. The work of Colli et al. (2020) also does not develop an explicit transfer function equation based on CFD modelling results to be directly compared with experimental results, as is developed in the present study. Instead, transfer function fit coefficients are derived experimentally, with different coefficients for each test site at temperatures below -4 °C.

The present approach is fundamentally different than that of Colli et al. (2020), in which wind speed and precipitation intensity (determined from the measured gauge accumulation) are used to adjust the measured gauge accumulation. While this enables adjustments to be performed at sites where only precipitation gauge and wind speed measurements are available, the collection efficiency adjustment (obtained from the precipitation intensity and gauge measurement) is not independent from the measured value to be adjusted. This could be problematic for adjusted precipitation accumulation estimates, as gauge measurement uncertainties can be propagated through both the measured gauge accumulation and the collection efficiency transfer function. It is also difficult to apply this approach across different hydrometeor types (e.g. rain and snow), as different types can have different fall velocities and associated collection efficiencies, even for the same precipitation intensity, as shown in Fig. 10 of the present manuscript. The fall velocity transfer function developed in the present study can be applied more broadly across different precipitation types, and the collection efficiency adjustment is determined independently from the gauge accumulation, with separate instruments for measuring the wind speed, precipitation fall velocity (e.g. a Precipitation Occurrence Sensor System (POSS) as shown in Part II), and precipitation accumulation.

In general, in this work the authors reproduced methodologies used in previous works and there are no significant improvements or novelty.

**Authors' response:** The authors disagree strongly that there are no significant improvements or novelty in this work. This is the first CFD modelling study to: (1) characterize precipitation gauge collection efficiency with respect to wind speed and fall velocity; (2) show that collection efficiencies are similar for different hydrometeor types with identical fall velocities; and (3) develop a universal transfer function based on wind speed and hydrometeor fall velocity that is broadly applicable across both liquid and solid precipitation types.

Previous studies have developed different transfer functions with wind speed for different snowfall crystal types (Thériault et al., 2012) or based on the wind speed and precipitation intensity for snowfall (Colli et al., 2020). The approach in Thériault et al. (2012) requires specific knowledge of the hydrometeor type, and has not been demonstrated to be viable for situations in which more than one precipitation type is present (e.g. both liquid and solid precipitation, different snowflake types) or the precipitation type is unknown or different from the specific crystal types considered; this makes it difficult to implement operationally. The approach in Colli et al. (2020) requires knowledge of the precipitation intensity, which is not independent from the gauge accumulation, considers temperatures below -4 °C only, and does not address the challenge of accurately adjusting liquid and/or solid precipitation at temperatures where either or both of these types may be present. The collection efficiency transfer function using fall velocity developed in the present study addresses these limitations and is broadly applicable. That stated, the authors acknowledge the limitations of the spherical hydrometeor model and recommend the study of non-spherical hydrometeors for future work (ln. 524-525).

With respect to the Reviewer's statement that "the authors reproduced methodologies used in previous works," the authors do not claim to be the first to use a CFD model and Lagrangian particle tracking to study the collection efficiency of hydrometeors for Geonor gauges. Previous studies using these approaches have been discussed and referenced in the manuscript. Similarities and differences in the approaches and results are discussed in the manuscript and in the responses above.

Furthermore, there are a few points the authors need to clarify, like e.g. the choice of the particle density values and the use of an unique empirical CE relation for different precipitation types and they need to evaluate how these impact on the results.

**Authors' response:** As discussed above (response to comment regarding L175), the density values used in this study are for spherical hydrometeors matching the diameter and fall velocity relationship provided by Rasmussen et al. (1999) and used in previous studies. The method by which the density values were determined is important to clarify, but it should be reiterated that the collection efficiency results in this study are highly sensitive to the wind speed and hydrometeor fall velocity and relatively insensitive to differences in hydrometeor density across hydrometeor types.

With respect to evaluating how a "unique empirical CE relation for different precipitation types" impacts results, the transfer function was developed using a uniform and unbiased approach, providing low RMSE values when fit to all modelled datasets including rain (0.04), ice pellets (0.02), wet snow (0.02), and dry snow (0.05). It is also assessed against wet snow and dry snow modelling results from Colli et al. (2016b) and demonstrates good overall agreement (Fig. 8). Further, this transfer function is directly applied to experimental results in the Part II manuscript and shows very good agreement over a wide range of conditions, supporting the modelling methodology and establishing further the fundamental role of hydrometeor fall velocity on gauge collection efficiency. The agreement between the modelling and experimental results suggests that this approach may be universally applicable across different climate regions and sites, demonstrating its potential for improving estimates of precipitation accumulation globally.

Reference:

Colli, M., Stagnaro, M., Lanza, L. G., Rasmussen, R. and Thériault, J. M. (2020). **Adjustments for wind-induced undercatch**

**in snowfall measurements based on precipitation intensity**, Journal of hydrometeorology, 21, 1039-1050.

**Unshielded precipitation gauge collection efficiency with wind speed and hydrometeor fall velocity. Part I: modelling results**

**Author Response to J. Kochendorfer (Referee #3)**

General comments

 Part I of "Unshielded precipitation gauge collection efficiency with wind speed and hydrometeor fall velocity" describes a modelling experiment designed to estimate precipitation undercatch in an unshielded precipitation gauge. The work focuses on the use of hydrometeor fall velocity to create improved transfer functions available to adjust unshielded precipitation measurements. The background and importance of the problem are well described in the introduction, which provides an excellent overview of past work in the modeling of precipitation undercatch. The methods and results are well documented, and the manuscript is generally very well written and easy to follow. The topic of undercatch is an important one, and this work is both new and useful, as it addresses the most difficult outstanding questions in precipitation undercatch; the manuscript establishes a valid way to reduce the significant uncertainty that precipitation transfer functions suffer from, and future work may also prove that this new approach can help reduce the site-to-site variability of collection efficiency and the resultant biases and uncertainty.

There are a couple of methodological points which need to be explored or explained more fully. These are described in more detail in the specific comments below, but I find the unrealistic background surface layer atmospheric flow problematic. In addition, the concept of a wind speed threshold above which collection efficiency is equal to zero is both impractical, and in my opinion theoretically unsound. However, I am not proposing that the entire model be redesigned, as it is certainly a valuable study as-is, especially as demonstrated by the accompanying Part II of this manuscript. I would however like to see these shortcomings handled differently within the manuscript.

After completing my review, I read the reviews from Referees #1 and #2, and feel compelled to write that I disagree with their main point, which is that these manuscripts are not novel enough to merit publication. I am ambivalent about whether or not they need to be published as two separate papers; I will leave that up to the editor. However, I maintain that the main point of this work, which is the inclusion of the fall velocity in a transfer function, is indeed both new and useful.

Theriault et al. (2012) includes a transfer function with a snowflake type parameter in it, but not the hydrometeor fall velocity. While Theriault et al. (2012) helped demonstrate the connection between hydrometeor fall speed and catch efficiency, and in general the importance of snowflake type, it did not include an easily applicable method for the improvement of operational precipitation measurements. While crystal type and hydrometeor fall velocity are certainly linked, as both manuscripts demonstrate, the use of the hydrometeor fall velocity, which can be measured relatively reliably and automatically, is important as a characteristic separate from the crystal type. All hydrometeors (not just snowflakes) have a measurable fall velocity, and as demonstrated by the present manuscripts under review, this fall velocity can be used to improve the collection efficiency transfer function. This is new. None of the references offered by Reviewer #1 and Reviewer #2 demonstrate a transfer function that includes the hydrometeor fall velocity. Nor for that matter, in my opinion, do any of those papers offer practical improvements to the currently available transfer functions that can be applied in an operational network. It is also worth noting that most of the important papers that Reviewer #1 and Reviewer #2 cite as evidence of the lack of novelty in the present paper were already cited in the present paper; it is not as if the authors of the paper under review were hiding the fact that this past work existed, or that it influenced their own work.

It is also worth noting that the use of the fall velocity is very different from the use of precipitation intensity for the improvement of collection efficiency transfer functions. While there may be some general correlation between precipitation intensity and hydrometeor type, precipitation intensity is not a good proxy for hydrometeor type, and in fact has real limitations for use in collection efficiency transfer functions. One of the most significant of these limitations is the fact that both precipitation intensity and collection efficiency are heavily dependent on the same precipitation measurement; they are not independent variables, and in such a case it is easy to demonstrate correlations that have no real or physical relevance.

**Authors' response:** The authors thank Dr. Kochendorfer for his detailed and constructive feedback and his support of the importance and novelty of this work.

Specific comments

Ln. 53. Explain what is meant by, "a sharper decay and higher intercept of a negative exponential distribution." The decay is with respect to what? This actually does bring to mind an altered curve, although I'm not sure if I am seeing it correctly. Anyway, I wouldn't write something like this and expect my readers to be able to understand it. In addition, I have no idea what are on the x- and y- axes of this imagined curve.

**Authors' response:** We will revise the manuscript to clarify this point. The negative exponential distribution defines the number of hydrometeors per unit volume per unit size as a function of the equivalent melted diameter of a water droplet. Plotting the log of the number of hydrometeors per unit volume per unit size on the y-axis against the equivalent melted diameter on the x-axis gives a straight line for the negative exponential distribution. Both the slope and intercept of the line change with precipitation intensity based on the Gunn and Marshall (1957) results, with reduced numbers of larger melted diameters with lower intensities.

Ln. 147. Why was the ground modeled as a frictionless wall? I am afraid I may be climbing up onto the soapbox here. However, I maintain that is not a 'get off my lawn' comment, because modeling atmospheric flow is not really my specialty. I know others have modeled gauge catch efficiency using the same boundary condition. But it results in an unrealistic vertical wind speed profile, in which the horizontal wind does not decrease with height, and is not zero at the ground. Just because others have done it, does not mean it makes sense. Especially when modeling a large shield (which is admittedly not the case here), a realistic vertical wind speed profile is needed to simulate realistic flow over the shield. But more importantly, without a zero-slip boundary condition at the surface, the model will not generate realistic background turbulence; in neutral atmospheric conditions, turbulence near the surface is generated by wind shear. With a frictionless surface there will presumably be no wind shear, and also no background turbulence. To clarify, I am not talking about the turbulence created by the gauge, but by the surface of the earth. This 'normal' background surface layer turbulence is important because it affects the flow over the gauge and the hydrometeors falling towards the gauge. In real life, the atmospheric flow at the earth's surface is not laminar. The assumption that undercatch can be modeled accurately in laminar background atmospheric flow should at least be discussed, along with the possible shortcomings.

**Authors' response:** This is an important point and an area for future work. The authors recommend that a brief discussion is added to Sect. 4.1 to clarify the approach used in the present study and its limitations. This study uses a 5% inlet turbulence value that acts as a bulk turbulence in the atmosphere (Panofsky and Dutton, 1984) but may underestimate experimental results (Armitt and Counihan, 1968). A no-slip boundary condition was modelled at the surface following the approach of previous studies (Baghapour et al., 2017; Baghapour and Sullivan, 2017; Colli et al. 2016a; Colli et al. 2016b). Further study with a no-slip boundary condition under different turbulence conditions could lead to further insights into the influence of turbulence intensity on precipitation gauge collection efficiency.

Table 1. $uw$ hasn't been defined yet. Or if it has, I can't find it. Also, I find this a confusing choice as the symbol for the free stream wind speed. This is because w is often used for the vertical wind speed, and because $ux$, $uy$, and $uz$ are also used to describe different components of the wind velocity; $uw$ looks to me like another way to describe the vertical wind speed.

**Authors' response:** Good point. The authors suggest changing $u_w$ to $U_w$ and $u_f$ to $U_f$ and adding the $U_w$ reference in the updated manuscript.

Ln. 198. Based on the statement that hydrometeor interactions were ignored (ln. 188), I am guessing that "interactions *within* the gauge orifice" should be changed to, "interactions *with* the gauge orifice."

**Authors' response:** This is referring to the potential hydrometeor interactions as they move through the fluid domain in the case where their paths cross near to one another. The potential for coalescence of two hydrometeors, for example, is ignored in this study. The authors will clarify this point in the manuscript.

Ln. 285. The way this is currently written it could be misinterpreted to mean that u* is the free-stream wind speed, not the, "peak velocity along the gauge centerline normalized by the free-stream wind speed." Perhaps the normalization could be moved to the end of the sentence – this sort of normalization is to be expected anyway, so I would argue that it isn't a critical part of the definition. "Peak velocities along the gauge centerline (u\*) are compared… in Fig. 3, with the centerline velocities normalized by the free-stream wind speed." Maybe? Also, I find u\* a confusing choice, as ustar (u\*) is an often-used variable with a completely different and well-established usage.

**Authors' response:** Thank-you. The authors will update the manuscript with the proposed wording change. We recommend maintaining the use of u\* for the normalized velocity, as it follows the convention used by Baghapour et al. (2017).

Figure 3. I believe the y-axis should be labeled u\*, not z\*. Also include *uw* (or its replacement!) in the caption in parenthesis after, "normalized free-stream velocity" to help clarify the meaning of the panel (a) and (b) titles.

**Authors' response:** Figure 3 shows the normalized free-stream velocity along the gauge centerline with normalized height above the gauge orifice z\*. The height above the gauge orifice is normalized by the orifice diameter. The location in the domain is given by x, y, and z coordinates, with the z-axis directed upward. We appreciate Dr. Kochendorfer's perspective here, but recommend maintaining the use of z\* for the description of the normalized position above the gauge orifice, as it follows the convention used by Baghapour et al. (2017). The authors agree that $U_w$ should be added in the caption, as recommended.

Figure 4. This is an excellent figure. I suspect we will see it reference and recycled many times, in future presentations.

**Authors' response:** Thank-you!

Figure 5. Small issue, but the legend shows open yellow squares for ice pellets, and the plot shows closed yellow squares (*uf* $= 5$ m s$^{-1}$).

**Authors' response:** For 5 m s$^{-1}$ fall velocities, rain and ice pellets yield collection efficiencies close to 1 and are nearly identical. In this case, the circle for rain is inside the square for ice pellets. Ln. 315 explains that these results are nearly identical, but we will note how this impacts the markers shown in the figure to help mitigate any confusion.

Ln. 320. Clarify by changing "hydrometeors up to about 3 m s-1 wind speed" to, "hydrometeors for horizontal wind speeds up to about 3 m s-1". I was confused by all the different speeds in this sentence.

**Authors' response:** Good point, thank-you. This has been updated.

Ln. 311 – 324. Some explanation of why the "dry snow" results are so unrealistic is needed. Experimental collection efficiencies are never this low (or zero). Is your hypothesis that this is because pure "dry snow" rarely occurs? Or is it because the experimental collection curves are derived wrong? I will say more about this elsewhere, but I find the suggestion that collection efficiency drops to zero problematic (and impractical). I suspect that it may be due to the fact that the modeled background flow is not turbulent. In the real world, surface layer flow and particle dispersion are stochastic processes. Given enough time or water, some hydrometeors will always be forced into the gauge by an errant eddy, no matter how slowly they fall or how high the wind speed is. The trajectories in Figure 4 are fine for what they are, but they show how hydrometeors behave in a laminar wind tunnel, not in actual turbulent surface layer flow. Turbulence intensity typically increases faster than the mean wind speed near the land surface, so it actually becomes more important as the wind speed increases. This may be why most experimental results reveal a sigmoid or exponential response of collection efficiency to wind speed, with the sensitivity of collection efficiency to increasing wind speed decreased (with the sigmoid function becoming flat, or unchanging with respect to wind speed) at high wind speeds.

**Authors' response:** Dr. Kochendorfer raises some excellent questions here. The authors recommend that a brief discussion is added to Sect. 4.3 to describe the potential limitations of the time-averaged model for estimating small collection efficiencies, highlighting that the transfer function has not been assessed experimentally for snow above 6 m s$^{-1}$ wind speeds, and cautioning users about performing large experimental adjustments with large associated uncertainties. Potential explanations for the unrealistic collection efficiencies for dry snow (values decreasing to zero) are explored below, and present several avenues for future work.

It is important to note that the results to this point, and the transfer function, refer to a given hydrometeor with a specific fall velocity, while in practice, a range of hydrometeor sizes and fall velocities are encountered. In this case, the collection efficiency tends to descend to small (but non-zero) collection efficiency values even at 10 m s$^{-1}$ wind speeds, as a small number of larger hydrometeors, with higher fall velocities, are still able to be captured by the gauge. This is shown in Fig. 9 and discussed in ln. 395-399.

The spherical hydrometeor approximation for dry snow is another area that could contribute to reduced collection efficiency for dry snow. For spherical dry snow hydrometeors, the hydrometeor volume and associated buoyancy can be greatly overestimated relative to that for non-spherical hydrometeors such as dendrites, particularly for large hydrometeor diameters.

The increased buoyancy force could reduce the collection efficiency relative to flat dendrites with much lower volume and associated buoyancy. Further investigation of dry snow with non-spherical hydrometeor models is recommended in the manuscript as an area for future work (ln. 518-519).

The time-averaged numerical model is another area that could play a role. The present time-averaged model results show that collection efficiencies, for a given hydrometeor, can decrease to zero depending on the hydrometeor fall velocity and wind speed. Previous studies have shown similar results with collection efficiencies decreasing to zero below a given hydrometeor size for liquid (Nešpor and Sevruk, 1999) and solid hydrometeor types (Thériault et al., 2012;Colli et al., 2016).

Time-averaged simulations provide an estimate of the mean velocities through the domain and have been shown to provide good overall agreement with experimental results despite underestimating the magnitude of the turbulent intensity above the gauge orifice (Baghapour et al., 2017). Large-eddy simulation (LES) models, which are computationally intensive, can better resolve the eddy dynamics and temporal variations in the flow influencing the collection efficiency values over time. Baghapour et al. (2017) showed that for an unshielded gauge, this temporal variability in collection efficiency increases with wind speed (collection efficiency standard deviation of 0.061 for 3 m s$^{-1}$ wind speed and 0.181 for 7 m s$^{-1}$ wind speed for 5 mm snow size). Time-averaged LES values were 6 % and 2 % lower than RANS results at these wind speeds for this snow size. In this case, the turbulent fluctuations in the flow are contributing to variations in collection efficiency over time and are slightly decreasing the overall ability of the gauge to capture precipitation over time. Under conditions where the collection efficiency is small, the temporal variability in collection efficiency could allow for small but non-zero collection during some periods of time even if nothing is captured most of the time, depending on the turbulence intensity. In addition to the turbulence intensity, local wind direction changes may be more important for collection. From Baghapour and Sullivan (2017), it was found that the forward edge of the gauge causes a local flow layer preventing snow collection – and the corresponding falling snow momentum must be greater to be collected. Wind direction changes would act to temporarily break up these layers. This would suggest a difference between dry and wet snow might be expected. As well, wind tunnel and CFD assume steady wind directions and speed, which are not likely in the field. These local acceleration/decelerations would enhance dry snow collection and would not be captured using current experimental and numerical approaches. Further study using LES models under different boundary conditions and turbulence scales representing different site conditions (roughness, length, topography…) could help to better understand the collection efficiency under conditions where RANS results yield zero collection efficiency.

It is also important to consider the measurement uncertainties associated with small experimental collection efficiencies obtained at high wind speeds. Under these conditions, the measured accumulations can be very small and close to the gauge uncertainty due to environmental factors (e.g. wind noise, temperature change), making small collection efficiencies difficult to assess with certainty experimentally (e.g. Smith et al., 2020). The higher uncertainty in experimental collection efficiency estimates where measured accumulations are small is discussed in Part II (ln. 241-244 and 508-511). The reference DFAR configuration could also be capturing less than the true amount falling in air, particularly for higher wind speeds and low fall velocity hydrometeors. Experimental comparison of the DFAR configuration with the bush gauge suggests this difference is small (Yang, 2014); however, it could contribute to a small systematic increase in the experimental collection efficiency if the reference was catching slightly less than the true value. These are additional areas for future work that are beyond the scope of the present study.

Ln 335, Eq. 18. Would it be possible to derive a collection efficiency equation, or its functional form, from the equations used within the model? I am a little disappointed that a modeling paper relies on an empirical equation.

**Authors' response:** The complex 3-dimensional flow profile varies with the free-stream wind speed, and would be difficult to derive explicitly over the fluid domain due to the non-linear nature of the results. If this velocity profile could be derived explicitly, then integration of hydrometeor trajectories over the domain based on the drag and hydrometeor characteristics would be required to determine the collection efficiency, presenting an additional obstacle for deriving the collection efficiency explicitly from the governing equations.

Ln. 344 – 345. I am again flummoxed by this concept that collection efficiency = zero at some point. What purpose does it serve? Is there any measurement evidence to support it? And how does one correct a precipitation even that occurs when the collection efficiency is defined as zero? I believe that the introduction of this zero-collection-efficiency concept and the emphasis placed on it in this paper may confuse others and hinder future progress in collection efficiency research. I grant that at low temperatures and high winds, an unshielded gauge can fail to measure any precipitation, but that is in part because most

30-min snowfall 'events' are near the measurement threshold of the gauge, in the 0 – 0.4 mm range. But just because we can't always measure it, doesn't mean it is zero. And if collection efficiency is defined as zero by the transfer function, how to we apply this function when precipitation is measured under these conditions. In a large enough dataset, we will be very hard pressed to find any commonly-occurring environmental conditions under which the reference catches precipitation and the unshielded gauge NEVER catches precipitation. But this is indeed what this theory prescribes, that there are certain conditions under which it is impossible for an unshielded gauge to collect certain hydrometeor types. That is very tall claim. The existence of such conditions in the real world should be demonstrated before making zero collection efficiency a central part of the theory. At a minimum, the discrepancies between past experimental results and the modeled results should be discussed.

**Authors' response:** This is an important point, and is discussed in detail above (ln. 311-324 comment). The authors recommend that a brief discussion is added to Sect. 4.3 to describe the potential limitations of the model for estimating small collection efficiencies, highlighting that the transfer function has not been assessed experimentally for snow above 6 m s$^{-1}$

wind speeds, and cautioning users about performing large experimental adjustments with large associated uncertainties.

Figure 6 and ln 349 – ln. 352. If I understand correctly, these results were produced using Equation 9 and the fall velocity, not the more complex precipitation characteristics. So why was only wet snow shown (or discussed) at uf = 1.5 m s$^{-1}$? In theory, the same transfer function would be used for different precipitation types, given the same fall velocity. But not all the precipitation types are shown or discussed. Why aren't all the collection efficiency curves shown in Figure 5 shown here? Was the figure too busy? In all honestly, initially I was confused, and thought that only wet snow was modeled at uf = 1.5 m s-1, but I believe I understand now that these results should be equally valid for all precipitation types, as they are purely a function of fall velocity.

**Authors' response:** These points will be clarified in the manuscript. Currently, Fig. 6 shows the transfer function relative to the specific CFD curves used for the fit as described in ln. 337-339. A single CFD curve was used for each fall velocity in the fit to ensure that the transfer function was unbiased over the entire range of fall velocities studied. The authors recommend adding all of the CFD results from Fig. 5 to Fig. 6 to better demonstrate the results for all hydrometeor types relative to the transfer function. The authors also recommend that the RMSE results for rain (0.04), ice pellets (0.02), wet snow (0.02), and dry snow (0.05) compared with the collection efficiency transfer function are added to Sect 3.3 to better describe the specific

CFD results with each hydrometeor type relative to the transfer function.

Ln. 389. Clarify that the dependence of collection efficiency on hydrometeor type and precipitation intensity was modeled solely based on differences in hydrometeor fall velocity.

**Authors' response:** In lines 385-386, it is stated that "For each hydrometeor type and precipitation intensity, the overall collection efficiency was derived for wind speeds from 0 to 10 m s$^{-1}$ using the empirical expression for collection efficiency (Eq. 18) based on wind speed and hydrometeor fall velocity." We will revise this statement to indicate the point raised by the reviewer more explicitly.

Figure 9 and its discussion. Explain why none these curves look like the 'dry snow' curves in Figure 6. I believe it is because of the distribution of different hydrometeor sizes (and fall velocities), but it is still worth pointing out.

**Authors' response:** Good point. The curves in Fig. 9 are integrated over the hydrometeor size distribution, which includes a range of hydrometeor sizes and fall velocities, as noted. This leads to a more gradual decrease in collection efficiency with wind speed at higher wind speeds than that shown in Fig. 6 (for a given fall velocity) because even at these higher wind speeds there is still a proportion of hydrometeors with sufficiently high fall velocities to be captured by the gauge. The authors recommend this comparison is noted in Sect. 3.4.2.

Ln. 507. Delete "with" in, "results with over…"

**Authors' response:** Removed. Thank-you.

Ln. 515. Rephrase to clarify that 1.0 m s-1 refers to the fall velocity.

**Authors' response:** "fall velocity added". Thank-you.

Ln. 525. Delete, "considered to be."

**Authors' response:** Deleted. Thank-you.

Ln. 535. Delete, "that is."

**Authors' response:** Deleted. Thank-you.

Ln. 573 – 577. Interesting. I had no idea.

**Authors' response:** Thank-you.

Ln. 588. The phrase, "have reduced ability to be collected" is awkward as written.

**Authors' response:** Reworded. Thank-you.

Ln. 613, 614, 615, 619, 620, 624. I find the use of "overall" confusing. It has too many other common meanings. For example, my first read of, "Overall collection efficiencies with precipitation intensity…" on ln. 613 made me think that a comma after

"overall," had been omitted. Looking back, I see that the term "overall" is nicely defined in Section 2.3, and again on ln. 370, but the use of a term that is less commonly used in normal English would make it clearer that it has a specific meaning. Perhaps,

"integrated catch efficiency?"

**Authors' response:** This is an interesting point. The authors recommend replacing "Overall collection efficiency" with

"Integrated collection efficiency" to describe the collection efficiency derived over a range of hydrometeor sizes and fall velocities and distinguish it from collection efficiency results for a specific fall velocity.

Ln. 624, Clarify that, "conditions when solid, liquid, or mixed precipitation can be present" refers to conditions when all of these types may be occurring, such as near-zero degrees C. As-is, 30 deg C in a thunderstorm qualifies as a time when, "solid, liquid, or mixed precipitation can be present," as does very cold conditions, when only solid precipitation can occur. I am sure there are better ways to write it, but one suggestion that remains fairly close to what is written is, "conditions when solid, liquid, *and* mixed precipitation can *all* be present." Or, "conditions when it is difficult to know the phase of the precipitation,

"or near-zero degrees…"

**Authors' response:** Reworded for clarity. Thank-you.

Ln. 644 – 645. In my opinion the sentence beginning with, "The results from the ability of the hydrometeor…" can be removed.

It is redundant; the previous sentence makes this point.

**Authors' response:** The authors agree that this point is somewhat redundant, but recommend that this sentence is retained in the manuscript in order to make this point clearly and explicitly.

**References**

Armitt, J., and Counihan, J.: The simulation of the atmospheric boundary layer in a wind tunnel, Atmospheric Environment, 2, 49-71,
https://doi.org/10.1016/0004-6981(68)90019-x, 1968.
Baghapour, B., and Sullivan, P. E.: A CFD study of the influence of turbulence on undercatch of precipitation gauges, Atmospheric Research,
197, 265-276, https://doi.org/10.1016/j.atmosres.2017.07.008, 2017.
Colli, M., Lanza, L. G., Rasmussen, R., and Thériault, J. M.: The collection efficiency of shielded and unshielded precipitation gauges. Part
II: Modeling particle trajectories., J. Hydromet., 17, 245-255, https://doi.org/10.1175/JHM-D-15-0011.1, 2016.
Nešpor, V., and Sevruk, B.: Estimation of wind-induced error of rainfall gauge measurements using a numerical simulation, Journal of
Atmospheric & Oceanic Technology, 16, 450-464, https://doi.org/10.1175/1520-0426(1999)016<0450:EOWIEO>2.0.CO;2, 1999.
Panofsky, H. A., and Dutton, J. A.: Atmospheric turbulence: models and methods for engineering applications, Wiley-Interscience, 1984.
Thériault, J. M., Rasmussen, R., Ikeda, K., and Landolt, S.: Dependence of Snow Gauge Collection Efficiency on Snowflake Characteristics,
Journal of Applied Meteorology & Climatology, 51, https://doi.org/10.1175/JAMC-D-11-0116.1, 2012.
Yang, D.: Double fence intercomparison reference (DFIR) vs. bush gauge for 'true' snowfall measurement, Journal of Hydrology, 509, 94-
100, https://doi.org/10.1016/j.jhydrol.2013.08.052, 2014.

**Unshielded precipitation gauge collection efficiency with wind speed and hydrometeor fall velocity. Part II: experimental results**

**Author Response to Anonymous Referee #1**

This manuscript shows that the RMSE of the collection efficiency can be significantly reduced if the fall speed derived from the Precipitation Occurrence Sensor System (POSS) is used. The paper is well written and shows new findings as the POSS can be used to improve the adjustment of solid precipitation. Nevertheless, I think that the text could be more concise for clarity and key information are missing. They are listed below. I recommend major revisions.

Major comments:

1. Introduction:

i) A few references are missing. 1) Colli et al. (2020) should be added to the paragraph discussing methods to improve the adjustment of solid precipitation. Colli et al. (2020) showed that the precipitation intensity improvements the adjustment of solid precipitation at given wind speed. 2) Chubb et al. (2015) also proposed that the precipitation rate as could be used to adjust solid precipitation measurements.

Colli, M., Stagnaro, M., Lanza, L. G., Rasmussen, R. and Thériault, J. M. (2020). Adjustments for wind-induced undercatch in snowfall measurements based on precipitation intensity, Journal of hydrometeorology, 21, 1039-1050.

Chubb, T., Manton, M. J., Siems, S. T., Peace, A. D., & Bilish, S. P. (2015). Estimation of Wind-Induced Losses from a Precipitation Gauge Network in the Australian Snowy Mountains, Journal of Hydrometeorology, 16(6), 2619-2638.

**Authors' response:** We thank the reviewer for identifying these references, and will add them to the introduction.

ii) What is the goal of the study? A summary of the methodology is given in the last few paragraphs but it never stated the goal clearly.

**Authors' response:** We will state the goal of the study more clearly in the introduction: "In this work, transfer functions incorporating hydrometeor fall velocity are developed to reduce the uncertainty (RMSE) in collection efficiency and precipitation accumulation estimates from unshielded Geonor T-200B3 precipitation gauges." The authors also propose stating the goal earlier in the introduction, instead of only in the last paragraph.

2. The methodology section is incomplete.

i) a description of the CFD simulations is missing. The relevant information from Part 1 should be added to the methodology of this manuscript.

**Authors' response:** We recommend that a brief description of the CFD model and simulations is added to the methodology introducing the CFD transfer function (Sect. 2.4). We are wary of too much overlap with the Part I manuscript, which includes a detailed description of the CFD model and simulations. Within the present manuscript (Part II), the CFD transfer function is presented in the introduction (ln. 96-101) and methodology (ln. 208-216), with reference to the Part I manuscript.

ii) A description of the method used to develop the transfer functions, in particular, the fall speed threshold values given in

Section 3.1 should be added.

**Authors' response:** We will clarify this in the manuscript. The fall velocity and temperature ranges presented by precipitation phase in Section 3.1 (Table 2) summarize the event-based experimental observations from the POSS and a temperature sensor in an aspirated shield, respectively, and are independent from the methodology used to develop the transfer functions. The descriptions of the methods used to develop the HE1 and HE2 transfer functions in Section 3.3 should be expanded to include more detail regarding the fall velocity threshold values. For the HE1 function, the fall velocity threshold was varied over the measured fall velocity range in 0.01 m s$^{-1}$ increments, with the threshold of 1.93 m s$^{-1}$ found to provide the lowest overall

RMSE for the experimental dataset. For the HE2 transfer function, the fall velocity threshold was varied over the measurement fall velocity range in 0.01 m s$^{-1}$ increments, with the threshold of 2.81 m s$^{-1}$ found to provide the lowest overall RMSE. Details regarding the wind speed threshold for the CFD transfer function are provided in the Part I manuscript (Sect. 3.3), but can be reiterated in Section 2.4 of the present manuscript for clarity. For the KCARE transfer function, ln. 202-205 in the manuscript describes the methodology for determining the temperature threshold $T_t$.

3. Section 3.1: How are the air temperature and fall speed threshold values determined in the study?

**Authors' response:** The derivation of the air temperature and fall velocity thresholds used in the study are addressed in the response to comment 2ii above.

In Table 2, the fall speed values for the precipitation type categories overlap. For example, snow events could also be mixed events if the temperature is <0.5_C and the precipitation falls at < 2.32 m/s. It should be clarified in the text.

**Authors' response:**  We agree to clarify this in the text. In Table 2, the temperature and fall velocity values are stratified by the 30-minute precipitation type classification determined from the minutely POSS precipitation type output following the methodology outlined in Sect. 2.3. As noted in the above response (comment 2ii), the experimental results summarized in

Table 2 and plotted in Figure 1 are not used to determine threshold values for transfer functions. These results are presented in Section 3.1 to illustrate how multiple precipitation types, with different fall velocities, can be present within a given temperature range, presenting a challenge for transfer function methods distinguishing different precipitation types by temperature. The fall velocity thresholds for HE1 and HE2 were determined empirically to best capture the trends in experimental results by minimizing the RMSE.

4. Why not using the temperature thresholds used in Kochendorfer et al. 2017b, which are -2_C to +2_C, to discriminate the precipitation types? Those are the threshold commonly used in the literature.

**Authors' response:** The results in this study illustrate the challenges of using ambient temperature as a proxy for precipitation type, as multiple precipitation types – with different fall velocities – can be present within a given temperature range. Precipitation types and fall velocities in this study were determined from the POSS instrument as described in Sect. 2.3. Fig. 1 shows the event-based results with 30-minute mean surface air temperature and fall velocity by POSS precipitation type classification. It is apparent that in this -2 °C to +2 °C temperature range, a wide range of fall velocities and precipitation types can be present. Accordingly, there is significant scatter in the collection efficiency results with respect to wind speed for this temperature range, as shown in Fig. 2c.

The results in Tables 5 and 7 demonstrate that collection efficiencies and adjusted precipitation accumulation can be determined with greater certainty (lower RMSE) at these temperatures using adjustments based on wind speed and fall velocity relative to adjustments based on wind speed and temperature. The use of fall velocity provides a quantitative means for adjustments to be performed across precipitation types (for example, mixed precipitation with a range of fall velocities) and enables adjustments to be performed even under conditions where the precipitation type may be unknown or difficult to determine (e.g. 'undefined' events).

Minor comments:

1. Lines 81-83: Change hydrometeor type for "type of solid precipitation" or "type of snow" because the study was done for solid precipitation. Add "fall speed" to the sentence because that is a key parameter of the study. The revised sentence could be: "Theriault et al. (2012) demonstrated similar trends for snowfall, with collection
efficiencies varying significantly with the type of solid precipitation, fall speed and size distribution."

**Authors' response:** We apologize for any confusion – this statement was made within the context of previous work involving CFD simulations. The simulations presented in Theriault et al. (2012) investigated how collection efficiency varies with wind speed depending on the specific snowflake type and selected slope size distribution value. Here, we can change "hydrometeor type" to "type of solid precipitation," as proposed. The linkage of the simulation results to theoretical terminal velocities computed for snowflakes that were collected and photographed is captured in lines 82 to 84 of the present manuscript.

2. Lines 171-173: The transfer function uses the accumulated precipitation while the CFD simulations uses the precipitation intensity. Clarify this possible inconsistency.

**Authors' response:** The CFD simulations are based on time-averaged simulation results and the collection efficiency is derived from the ratio of the precipitation intensity captured by the gauge to the true precipitation intensity falling in air.

Integrating over a period of time (in this case 30-minutes) gives the collection efficiency as a function of the ratio of the precipitation accumulation captured by the gauge to the true amount.

3. Equation 3: Could you explain why this equation is relevant? If not, remove it.

**Authors' response:** Equation 3 shows how the uncertainty in the experimental collection estimate scales with the magnitude of precipitation accumulation for rain, as shown in Fig. 2a and discussed in Section 3.2. It is apparent from Eq. 3 and the results in Fig. 2a that as the measured precipitation accumulations become smaller and approach the precipitation gauge measurement uncertainty, the uncertainty in the measured collection efficiency estimates can become quite large. This is an important point for understanding a component of the scatter in the collection efficiency results in Figs. 2b, 2c, and 2d, which is not readily apparent when collection efficiency results are plotted as a function of wind speed.

4. Lines 287-292: Why using 1.93 m/s as a threshold? It should be explained.

**Authors' response:** We will update Sect. 3.3 with this explanation. The threshold of 1.93 m s$^{-1}$ was determined by varying the fall velocity threshold in 0.01 m s$^{-1}$ increments over the measurement range of fall velocities (Table 2). This mean fall velocity threshold provided the lowest RMSE for the HE1 transfer function.

5. Lines 296-301: Why using 2.81 m/s as a threshold? It should be explained.

**Authors' response:** We will update Sect. 3.3 with this explanation. The threshold of 2.81 m s$^{-1}$ was determined by varying the fall velocity threshold in 0.01 m s$^{-1}$ increments over the measurement range of fall velocities (Table 2). This mean fall velocity threshold provided the lowest RMSE for the HE2 transfer function.

6. Figure 4: Did you try using boxplots instead of a scatter plot to show the collection efficiency? It could give an idea of the scatter in the collection efficiency with wind speed.

**Authors' response:** Yes, this approach was considered. While the use of boxplots is useful for summarizing the distribution of collection efficiencies across wind speed classes, or even wind speed and other classifications, it makes it more difficult to trace the results for specific events across different classifications (e.g. precipitation type, temperature, and fall velocity) because the events become lumped into boxes with only outliers shown. For example, looking at Fig. 2a, the two collection efficiencies for rain above 1.3 correspond with very small accumulation values as discussed earlier (i.e. their values approach the gauge measurement uncertainty). Looking at Fig. 2b, these events occur near 2 m s$^{-1}$ and 5 m s$^{-1}$. Fig. 2c shows that one of these events is between -2 °C to 2 °C and one event is above 2 °C. Fig. 2d shows that both of these events have fall velocities above 2.5 m s$^{-1}$. The RMSE values summarized in Tables 3, 5, 6, 8, and 9 provide a useful measure of the scatter, as they capture the spread/scatter between the measurement and transfer function as the transfer functions change continuously with wind speed and temperature or fall velocity.

7. Tables 3 to 9 could be put in an Appendix since that it is showing additional information. One could also do barplots instead of Tables.

**Authors' response:** The authors appreciate the suggestion, but strongly recommend that Tables 3 to 6 remain in results Sect. 3.4 (Assessment of transfer functions: collection efficiency) and Tables 7 to 9 remain in results Sect. 3.5 (Assessment of transfer functions: precipitation accumulation). The results in Table 3 capture the overall transfer function results and demonstrate the improvement in the fall velocity transfer functions relative to current adjustments based on wind speed and temperature. The other Tables demonstrate collection efficiency and precipitation accumulation RMSE by precipitation type, temperature and fall velocity classifications, linking with the results and discussion associated with Figs. 4 and 5. The use of Tables instead of bar plots has the advantage that the specific RMSE values are clearly shown for comparison with future studies.

8. Lines 477-479: The sentence: "While automatic . . . this work" seemed out of place. It may be better in the conclusion?

**Authors' response:** We feel that this statement fits best within the context of the Discussion, where it follows the discussion of the time periods and accumulation thresholds used in this and other work, and establishes boundaries for the scope of this work. We agree that it could also work well in the Conclusions section, but it would be more challenging to establish the same context in that case.

9. Line 505: The sentence: "The HE1 transfer function showed good results for snow, supporting its use for unshielded gauge.". I agree but Figure 3b (as an example) still shows lots of scatter in the collection efficiency for fall speeds associated with snow/solid precipitation (_1-2 m/s). Add a short discussion?

**Authors' response:** This is a good point, and one that we believe is already discussed in the manuscript. Based on the 0.10

collection efficiency RMSE for snow events as identified by the POSS in Table 4, the HE1 transfer function showed good results, as stated in line 505. Looking at the 0.10 collection efficiency RMSE for HE1 at fall velocity values $\leq 1.5$ m/s in Table

6 tells a similar story. However, in line with the reviewer's point, the collection efficiency RMSE for HE1 in Table 6 is higher (0.15) for events with fall velocity values between 1.5 m/s and 2 m/s. This higher RMSE value for HE1 is consistent with that for events classified as mixed precipitation in Table 4. This limitation of HE1 is noted and discussed in lines 516-521 of the manuscript.

10. Lines 537-539: This sentence is not quite right and I think that it is an important point. The references from Kienzle (2008)

and Harder and Pomeroy (2013) should be after the word "instructive" because they developed a method to diagnose the precipitation phase at the surface when the information aloft is not available. Theriault et al. (2012) suggested to use surface temperature but did not develop a method to diagnose the type/phase of precipitation. At the end of the sentence, the authors should refer to a paper that state the importance of the atmospheric conditions aloft to determine the type/phase of precipitation at the surface such as for example Stewart et al. (2015).

Stewart, R. E., J. M. Theriault, and W. Henson, 2015: On the characteristics of and processes producing winter precipitation types near 0_C. Bull. Amer. Meteor. Soc., 96, 623–639, doi:10.1175/BAMS-D-14-00032.1.

**Authors' response:** Thank-you for pointing this out. We will update the references as suggested to improve the clarity of this sentence.

**Unshielded precipitation gauge collection efficiency with wind speed and hydrometeor fall velocity. Part II: experimental results**

**Author Response to J. Kochendorfer (Referee #2)**

General comments

Part II of, "Unshielded precipitation gauge collection efficiency with wind speed and hydrometeor fall velocity" is the experimental companion to the Part I paper, which describes a modelling experiment. Part II tests the transfer function created in Part I, and it goes further to modify this transfer function based on the experimental results. It demonstrates that hydrometeor fall velocity can be used in a practical way to improve the adjustment of unshielded precipitation measurements. These improvements are impressive and significant.

Like Part I, the manuscript is well-written and easy to follow, and it is definitely worth publishing.

**Authors' response:** Thank-you!

Specific comments

Ln. 65 – 67. This is a misinterpretation of those results. In addition to the uncertainty of the adjustment, it overlooks the fact that adjusted measurements increase the magnitude of errors multiplicatively. For example, if the gauge measurement has an inherent uncertainty of 0.1 mm, with CE = 0.5, after adjustment the uncertainty will be doubled along with the measurement. Two single Alter gauges agreeing with each other with an uncertainty of 0.09 mm does not imply that they can be adjusted without increasing the uncertainty. I accept that there is significant room for improvement in our transfer functions, but I find it very difficult to believe that adjusted unshielded measurements will ever be as accurate as well-shielded measurements. I am afraid that someone reading between the lines here might take that to be the suggestion.

**Authors' response:** Dr. Kochendorfer makes a good point here. We will remove the reference to the comparison of replicate configurations of weighing gauges (Ln. 65-67).

Ln. 112. Change, "using similar methodology" to, "using *a* similar methodology" or, "using similar *methods*."

**Authors' response:** Updated to "using a similar methodology".

Ln. 172 and Eq (2). Why was *h* chosen for precipitation, instead of *P*?

**Authors' response:** $h$ was originally chosen to refer to precipitation accumulation as a height in units of mm. $h$ has been revised to $P$ to make the linkage with precipitation clearer and to match the terminology of previous publications. Thank-you.

Ln. 269 – 270. This makes me wonder about the details and physics of the POSS averaging. How is the hydrometeor fall velocity calculated by the POSS when there is mixed precipitation, and/or when there is significant variability in the types of hydrometeors simultaneously present? I am guessing that for the purposes of transfer functions, ideally the fall velocity would be representative of the total mass of water falling, but perhaps it is actually weighted towards the average by volume?

**Authors' response:** The POSS is an X Band (3cm wavelength) radar that measures the Doppler velocity spectrum from which the hydrometeor size distribution is derived. This has been described in detail in previous publications, including its use for precipitation typing; we refer the reviewer to the following publications for the details (Sheppard, 1990; Sheppard and Joe, 1994, 2000, 2008). The advantage of the POSS is that it rapidly measures the Doppler spectrum from a very large volume compared to other disdrometers, which measure individual particles with more limited sampling (e.g. Thies LPM, OTT Parsivel2). For large hydrometeors (say 5 mm), the sample volume is about the size of a small room. Several hundred Doppler/hydrometeor spectra are measured and reported every minute. There is on-ongoing research for snow and mixed precipitation type retrievals. We agree that ideally, the fall velocity would be representative of the total mass of water falling, but the complexities of hydrometeor drag, density, and mass are confounding factors still to be resolved. While the present approach of estimating the event fall velocity from the 30-minute average appears to perform well overall, further study to better characterize the fall velocity distribution and changes over 30-minute time periods could lead to further improvements in the model under specific conditions such as mixed precipitation.

Ln. 289. I apologize in advance, because I hate it when reviewers ask me these types of questions, but how was the threshold fall velocity of 1.93 m s$^{-1}$ selected?

**Authors' response:** The threshold of 1.93 m s$^{-1}$ was determined by varying the fall velocity threshold in 0.01 m s$^{-1}$ increments over the measurement fall velocity range in Table 2. This mean fall velocity threshold provided the lowest RMSE for the HE1 transfer function. A similar approach was used to derive the fall velocity threshold for HE2. We will add this information to the manuscript.

Equation 7b. Given my comments on Part I this should come as no surprise, but I think that defining CE = 0.0 any under conditions is problematic.

**Authors' response:** Dr. Kochendorfer raises an important issue with the definition of the collection efficiency at high wind speeds in the transfer function. The authors recommend revising Eq. 7b, Table 1, and Fig. 4c for HE1 with a minimum collection efficiency of 0.2 and wind speed threshold of 5.75 m s$^{-1}$, following the general approach of Kochendorfer et al.
(2017).

Ln. 299. Clarify that *CEHE2* decreases linearly with wind speed *at a given/fixed hydrometeor fall velocity*.

**Authors' response:** Updated. Thank-you.

Ln. 299 – 300. Explain how this works in practice. How were measurements that occurred when fall velocity was defined as
zero treated? Were they simply removed from the analysis? How is the user of these functions supposed to adjust such
measurements?

**Authors' response:** Over the test period there were no fall velocities of zero reported by the POSS and 30-minute mean fall
velocities were ~1 m s$^{-1}$ or higher. During non-precipitating periods the POSS does not output a fall velocity and these periods
are not included in the 30-minute average. While fall velocities of zero were not encountered during this study, and would not
be expected in general, the *HE2* transfer function is still defined in this case. In the case of zero fall velocity the collection
efficiency decreases with wind speed alone as shown in Eq. 8a. In this case the collection efficiency decrease with wind speed
will be faster than that for conditions where the fall velocity is higher.

Ln. 314 – 315, Figure 4 caption. Typo. I believe that the three occurrences of "*up*" in, "fall velocity *up* categories…" should
be replaced with "*uf*".

**Authors' response:** Updated. Thank-you.

Ln. 352. Why wasn't the same temperature threshold technique used for *KUniversal*? At the risk of personifying a, "get off
my lawn" attitude, I wonder how much of the improved performance of the *KCARE* adjusted measurements were caused by
large errors in measurements that were over-adjusted using *KUniversal* above this temperature threshold? The largest
improvement in RMSE includes some of these measurements, when T is between positive and negative 2 deg C (Table 8), and
I am guessing that at least some of the very poorly measurements were warmer, larger events (Fig. 5b).

**Authors' response:** *KUniversal* was developed from the WMO-SPICE results for eight test sites and is used for comparison
with the present study results from the CARE field test site. Modifications to *KUniversal* using the temperature threshold
technique would need to be assessed based on the entire dataset (all eight sites) and is beyond the scope of this study. *KCARE*
is developed from the CARE dataset for comparison with the site-specific fall velocity transfer functions developed in this
study. Both *KUniversal* and *KCARE* are similar at colder temperatures but differ as the temperature increases. The improvement in the *KCARE* transfer function results are primarily attributed to this more rapid increase in collection efficiency with temperature, reducing the overadjustment of some events and increasing the underadjustment of some events between -5

°C and -2 °C and between -2 °C and 2 °C (as shown in Fig. 5 and Table 8). It is important to note that even the *KCARE* transfer function exhibits increased uncertainties at these warmer temperatures relative to transfer functions using fall velocity, as rain, mixed precipitation, and snow can occur with different collection efficiencies. These differences cannot be distinguished using temperature alone, resulting in increased uncertainties at these temperatures.

Ln. 504. A realistic vertical wind profile, with a zero-slip boundary condition at the Earth's surface, may be important for larger wind shields.

**Authors' response:** Thank-you. This is an important point for studying other shield and gauge combinations in the future.

This note will be added to the manuscript.

Ln. 507 – 509. I agree that it is difficult to accurately adjust measurements at windy sites, but the 'limitation' described here is entirely avoidable. The collection efficiency was defined as zero above 7.19 m s-1 by choice, not by necessity.

**Authors' response:** We will revise the discussion for the HE1 transfer function to include a transfer function minimum collection efficiency of 0.2 for wind speeds above 5.75 m s$^{-1}$ following the general approach of Kochendorfer et al. (2017).

---

## Referee Report (RR1)

General comments:

The revisions that have been made to this manuscript satisfy all of the comments I made on the last versions. With the exception of a few technical issues, the manuscript is ready for publication.

I did not compare the last two papers to this one carefully, but I would hope that by combining them the total length might have decreased more. I will admit that I did not say this in my previous reviews however, so I understand if these suggestions are ignored. Perhaps Section 3.1 can be shortened by focusing on the differences between this approach and past studies, using the appropriate references. I am not sure if all three of these figures are unique or new enough to merit inclusion. In general, as I see it, the main thing that is new in the modeling work is the derivation of a transfer function that includes fall velocity, so keeping this in mind, perhaps there are other modeling sections that can be shortened as well.

Specific comments:

Abstract, ln. 9, 10, 16, 17… The reader hasn't been introduced to HE1 and HE2 yet. These need to be either defined in the abstract, or better yet, different, more generally understood terminology should be used.

Ln. 85. Spain and Norway were omitted from the list of countries with measurement sites.

Ln. 126. How can the ground be frictionless, and at the same time "no-slip" (ln. 775) or "zero-slip" (ln. 988)?

Ln 256. I still find the $z*$ and $u*$ terminology confusing, despite the fact that it has been used (once?) this way by Baghapour et al. (2017). Here is an example of a more common usage, from the AMS Glossary of terms: https://glossary.ametsoc.org/wiki/Friction_velocity Also it seems that the results here are basically the same as Baghapour et al. (2017), so the use of terminology that aids careful comparison may not really be necessary.

Ln. 319 – 323, and Eq. 19. What purpose does the derivation of $U_{wc}$ serve? I don't see how it contributes to the manuscript; I suggest removing this all together, unless I have missed something. The two sentences on ln. 334 – 335 would need to be removed as well.

In Figures 7, 8, 9, and 10 change "overall" on the Y-axis label to "integral."

Ln. 486 – 487. Put $CE_m$, $P_{un}$, and $P_{DFAR}$ within parenthesis.

Ln. 493. It isn't clear why "$CE$" is included in, "Collection efficiency transfer functions $CE$…"

Ln. 763. Change, "and their paths *shows*," to "and their paths *show*."

Ln. 916 – 918. The "nonlinearity in the relationship…" is inadequate. A physical explanation of these $CE$ differences would be preferable.

Ln. 1001. Perhaps change, "vertical" to, "fall" for the sake of consistency in terminology.

---

## Editor Decision (ED1)

**Editor decision hess-2020-553 and hess-2020-554.**

Unshielded precipitation gauge collection efficiency with wind speed and hydrometeor fall velocity. Part I and Part II.

Paper hess-2020-553 and its companion paper hess-2020-554 present a universal collection efficiency transfer function based on wind speed and hydrometeor fall velocity. The transfer function is developed based on CFD modeling experiments in part I and validated based on experimental results in part II.

The three reviewers who evaluated part I express conflicting views, two reviewers doubting the novelty of the paper, while the third reviewer is quite supportive. I believe the author response sufficiently explains the novelty of their contribution, as well as the added value of the function they propose for field application. However, I agree with the recommendation that the two parts of the paper be merged.

There are several reasons to do this. First, combining the two papers has the advantage of making both modelling and experimental results more directly accessible to the reader. Second, there is quite a bit of overlap between part I and II, in the Introduction as well Methods sections (and the overlap would grow given that reviewers recommended more explanation on the CFD modelling and transfer function in part II). Third, the authors can be more critical about what figures and information is essential to present to support the main storyline of the manuscript, especially in part I. Parts of the information provided in Methods and Results and some of the figures that show overlap in content can be moved to Supporting Information, where it is still accessible to the reader. In summary, merging the two parts has the advantage of bringing all the relevant information of this study together, in a much more condensed form.

Furthermore, please revise the manuscript to address the remaining review comments related to Part I and Part II, along the lines you suggested in your author replies.

---

## Author Response (AR2)

**Unshielded precipitation gauge collection efficiency with wind speed and hydrometeor fall velocity.**

**Authors' response:** Thank-you to John Kochendorfer and the anonymous reviewers for providing thoughtful reviews of the original and revised versions of this manuscript and greatly improving the quality of this paper. We have revised the paper to be more concise based on the reviewer suggestions. The list of all relevant changes and point-by-point reviewer responses are included below.

**List of all relevant changes with reference to tracked changes document:**

- Sections 3.1, 3.3, 6.1.1, 6.1.2, and 6.1.4.1 in the manuscript have been revised to present the key information related to the work more succinctly.
- Figs. 2 and 3 removed
- Ln. 8-10. Updated to introduce define HE1 and HE2 transfer function in abstract.
- Ln. 88. Spain added in list.
- Ln. 783. Corrected ln. 775 reference to "slip". "zero-slip" reference refers to the opportunity for future study with large shields using a zero-slip boundary condition at the earth's surface.
- Ln. 319-323, Eq. 19, and ln. 334-336 removed.
- Figs. 5, 6, 7, 8 updated with "integral" replacing "overall"
- Updated CEm, Pun, and PDFAR with parenthesis in text and removed CE equation as it is introduced earlier.
- Updated equation formatting and parenthesis.
- Ln. 763. Changed, "and their paths shows," to "and their paths show."
- Ln. 931-934. Updated with the physical description of the CE differences: "The small differences in collection efficiency across different hydrometeor types with the same fall velocity are attributed to the varying contribution from higher fall velocity hydrometeors, with collection efficiencies approaching 1, in the mass-weighted distribution of hydrometeor fall velocities."
- Ln. 1001. Changed "vertical" to "fall"

**Anonymous Referee #1 Comments:**

The revised manuscript is generally improved. However, the authors seemed to have merged directly Part 1 and 2 without adjusting the Part 1 content to present the key information succinctly. The information added from Part 1 (ex: section 3 and 6.1) should probably be shorten to focus on the key information related to the goal of the study, which is about developing the transfer function that includes the fall speed and wind speed. In the discussion, the modeling and field measurements are separate and I think that some of it could be combined. Therefore, I think that the manuscript needs some minor revisions before publication. The revision is mainly on the organization of the manuscript.

**Authors' response:** Sections 3.1, 3.3, 6.1.1, 6.1.2, and 6.1.4.1 in the manuscript have been revised to present the key information related to the work more succinctly. Figs. 2 and 3 have been removed.

**J. Kochendorfer (Referee #2) Comments**

General comments: The revisions that have been made to this manuscript satisfy all of the comments I made on the last versions. With the exception of a few technical issues, the manuscript is ready for publication. I did not compare the last two papers to this one carefully, but I would hope that by combining them the total length might have decreased more. I will admit that I did not say this in my previous reviews however, so I understand if these suggestions are ignored. Perhaps Section 3.1 can be shortened by focusing on the differences between this approach and past studies, using the appropriate references. I am not sure if all three of these figures are unique or new enough to merit inclusion. In general, as I see it, the main thing that is new in the modeling work is the derivation of a transfer function that includes fall velocity, so keeping this in mind, perhaps there are other modeling sections that can be shortened as well.

**Authors' response:** Sections 3.1, 3.3, 6.1.1, 6.1.2, and 6.1.4.1 in the manuscript have been revised to present the key information related to the work more succinctly. Figs. 2 and 3 have been removed.

Specific comments:

Abstract, ln. 9, 10, 16, 17… The reader hasn't been introduced to HE1 and HE2 yet. These need to be either defined in the abstract, or better yet, different, more generally understood terminology should be used.

**Authors' response:** Updated to introduce define HE1 and HE2 transfer function in abstract.

Ln. 85. Spain and Norway were omitted from the list of countries with measurement sites.

**Authors' response:** Included Spain and Norway in list.

Ln. 126. How can the ground be frictionless, and at the same time "no-slip" (ln. 775) or "zero-slip" (ln. 988)?

**Authors' response:** Corrected ln. 775 reference to "slip". "zero-slip" reference refers to the opportunity for future study with large shields using a zero-slip boundary condition at the earth's surface.

Ln 256. I still find the $z^*$ and $u^*$ terminology confusing, despite the fact that it has been used (once?) this way by Baghapour et al. (2017). Here is an example of a more common usage, from the AMS Glossary of terms: https://glossary.ametsoc.org/wiki/Friction_velocity Also it seems that the results here are basically the same as Baghapour et al. (2017), so the use of terminology that aids careful comparison may not really be necessary.

**Authors' response:** Figure and associated terminology removed.

Ln. 319 – 323, and Eq. 19. What purpose does the derivation of Uwc serve? I don't see how it contributes to the manuscript;

I suggest removing this all together, unless I have missed something. The two sentences on ln. 334 – 335 would need to be removed as well.

**Authors' response:** Ln. 319-323, Eq. 19, and ln. 334-336 removed.

In Figures 7, 8, 9, and 10 change "overall" on the Y-axis label to "integral."

**Authors' response:** Updated

Ln. 486 – 487. Put CEm, Pun, and PDFAR within parenthesis. Ln. 493. It isn't clear why "CE" is included in, "Collection efficiency transfer functions CE…"

**Authors' response:** Updated with CEm, Pun, and PDFAR in parenthesis. Reference to CE is removed as it is introduced earlier.

Ln. 763. Change, "and their paths shows," to "and their paths show."

**Authors' response:** Updated

Ln. 916 – 918. The "nonlinearity in the relationship…" is inadequate. A physical explanation of these CE differences would be preferable.

**Authors' response:** Updated with the physical description of the CE differences: "The small differences in collection efficiency across different hydrometeor types with the same fall velocity are attributed to the contribution from higher fall velocity hydrometeors less coupled to the local airflow, with collection efficiencies approaching 1, in the mass-weighted distribution of hydrometeor fall velocities."

Ln. 1001. Perhaps change, "vertical" to, "fall" for the sake of consistency in terminology

**Authors' response:** Updated

---

## Author Response (AR3)

**Unshielded precipitation gauge collection efficiency with wind speed and hydrometeor fall velocity.**

**Comments to Editor**

Dear Marie-Claire,

Based on your feedback, we have revised the manuscript to be more concise, with details of the method, results, and discussion related to the CFD modelling component moved to the supplementary material. Only aspects necessary to understand the development and application of the CFD transfer function are included in the main paper. The manuscript has also been revised with single sections for the results and discussion, with only critical details of the CFD modelling component included therein. This has improved the flow of the paper and reduced the overall paper length (33 pages including the References section, 7 Figures), in accordance with the recommendations provided.